# Diffusion posterior sampling for simulation-based inference in tall data settings

**Julia Linhart**  *julia.linhart@inria.fr*
*Université Paris-Saclay*
*Inria, CEA*

**Gabriel Victorino Cardoso**  *gabriel.victorino-cardoso@polytechnique.edu*
*CMAP, École Polytechnique*
*Institut Polytechnique de Paris*

**Alexandre Gramfort**  *alexandre.gramfort@inria.fr*
*Université Paris-Saclay*
*Inria, CEA*

**Sylvain Le Corff**  *sylvain.le__corff@sorbonne-universite.fr*
*LPSM, Sorbonne Université*
*UMR CNRS 8001*

**Pedro L. C. Rodrigues**  *pedro.rodrigues@inria.fr*
*Université Grenoble Alpes*
*Inria, CNRS*
*Grenoble INP, LJK*

**Reviewed on OpenReview:** *https://openreview.net/forum?id=cdhfoS6Gyo*

## Abstract

Identifying the parameters of a non-linear model that best explain observed data is a core task across scientific fields. When such models rely on complex simulators, evaluating the likelihood is typically intractable, making traditional inference methods such as MCMC inapplicable. Simulation-based inference (SBI) addresses this by training deep generative models to approximate the posterior distribution over parameters using simulated data. In this work, we consider the *tall data* setting, where *multiple independent observations* provide additional information, allowing sharper posteriors and improved parameter identifiability. Building on the flourishing score-based diffusion literature, F-NPSE (Geffner et al., 2023) estimates the *tall data posterior* by composing individual scores from a neural network trained only for a *single context observation*. This enables more flexible and simulation-efficient inference than alternative approaches for tall datasets in SBI. However, it relies on costly Langevin dynamics during sampling. We propose a new algorithm that eliminates the need for Langevin steps by explicitly approximating the diffusion process of the tall data posterior. Our method retains the advantages of compositional score-based inference while being significantly faster and more stable than F-NPSE. We demonstrate its improved performance on toy problems and standard SBI benchmarks, and showcase its scalability by applying it to a complex real-world model from computational neuroscience.

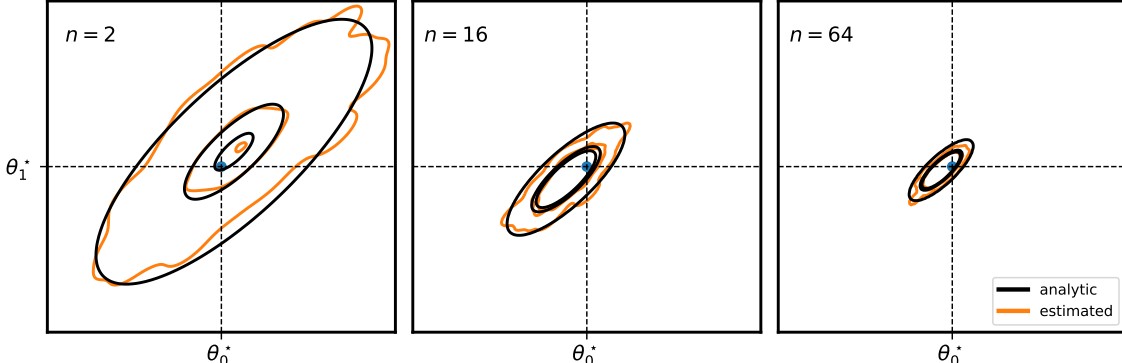

Figure 1: The posterior distribution of a model with a Gaussian simulator and Gaussian prior concentrates around the **true parameter** $\theta^\star$ as the number $n$ of observations $x_i^\star \sim p(x \mid \theta^\star)$ increases. The **analytic** posterior is compared to the posterior **estimated** with our score-based proposal (Algorithm 2: GAUSS).

## 1 Introduction

Inverting non-linear models that describe natural phenomena is a fundamental problem in many scientific domains (Gonçalves et al., 2020; Dax et al., 2023). We adopt a Bayesian perspective, where the goal is to infer the posterior distribution $p(\theta \mid x)$ that relates input parameters $\theta \in \mathbb{R}^m$ to output observations $x \in \mathbb{R}^d$. Sampling from this posterior becomes particularly challenging when the model's outputs are generated via complex simulations (e.g. based on non-linear stochastic differential equations). In such settings, the likelihood $p(x \mid \theta)$ is often intractable, making classical Bayesian approaches such as Markov Chain Monte Carlo (MCMC)(Hastings, 1970) either inapplicable or computationally prohibitive. Simulation-based inference (SBI), also known as likelihood-free inference, circumvents this limitation by relying on model simulations instead of explicit likelihood evaluations. Here, novel deep learning techniques can be used to accurately approximate arbitrarily complex posterior distributions (Cranmer et al., 2020). The standard procedure assumes a prior distribution $\lambda(\theta)$—encoding scientific knowledge about plausible parameter values—and uses the simulator to draw samples from the joint distribution:

$$\Theta_i \sim \lambda(\theta), \quad X_i \sim p(x \mid \Theta_i), \quad (\Theta_i, X_i) \sim p(\theta, x), \quad i = 1, \ldots, N_s,$$

with $N_s$ the simulation budget. From this simulated dataset, a neural network is trained to estimate statistical quantities of interest defined for every $(\theta, x)$. Once trained, it can be evaluated in *any* new observation $x^\star$ to approximate the posterior distribution $p(\theta \mid x^\star)$, a property known as *amortized* inference. Existing SBI methods differ in the target of the neural estimator: Neural Posterior Estimation (NPE) learns the posterior directly (Greenberg et al., 2019), Neural Likelihood Estimation (NLE) approximates the likelihood (Papamakarios et al., 2019), and Neural Ratio Estimation (NRE) estimates the likelihood-to-evidence ratio (Hermans et al., 2020). While NPE provides posterior samples directly via conditional normalizing flows (Papamakarios et al., 2021), NLE and NRE require additional MCMC sampling.

In this work, we consider an extension of the above Bayesian inference framework to the *tall data* setting (Bardenet et al., 2015), where multiple i.i.d. observations $x_{1:n}^\star = (x_1^\star, \ldots, x_n^\star)$ are available.[1] The corresponding *tall data posterior* is expected to provide more precise information about how to invert the simulator-model, with increasingly sharper posterior densities as the number of observations grows (see Fig 1):

$$p(\theta \mid x_{1:n}^\star) \propto \lambda(\theta)^{1-n} \prod_{j=1}^{n} p(\theta \mid x_j^\star) \, . \tag{1}$$

While this framework is crucial for practical applications, the extension of existing SBI algorithms to tall data settings remains challenging and no satisfactory solution has yet been proposed. In Rodrigues et al. (2021),

---

[1]i.i.d.: *independent and identically distributed.* For example, simulations $(x_1^\star, \ldots, x_n^\star) \sim p(x \mid \theta^\star)$ that are generated with the same set of simulator parameters are i.i.d., conditionally on $\theta^\star$.

the authors merge a fixed number of extra observations via a deepset (Zaheer et al., 2017) and fall back to NPE trained on an *augmented dataset* $\mathcal{D}_n = \{(\Theta_i, X_{i,1:n})\}$, with $n$ simulations $X_{i,1:n} = (X_{i,1}, \ldots X_{i,n}) \sim p(x \mid \Theta_i)$ per prior sample $\Theta_i \sim \lambda(\theta)$. While NPE leverages expressive normalizing flows to model the posterior directly, it imposes architectural constraints and requires fixed input dimensions on the neural network, limiting efficient and flexible conditional data modeling. This leads to two major drawbacks: (1) the potentially heavy simulation cost, as the simulation budget scales with the number of observations ($N_s^{\mathrm{aug}} = N_s \times n$) and (2) the lack of flexibility at inference time—when more observations become available, a new model tailored to the new context size must be retrained from scratch. In Hermans et al. (2020), the authors show how the *amortization* of NRE allows to handle the tall data setting *without requiring new simulations or retraining*. By factorizing over the set of multiple observations, they reformulate the "tall data ratio" as the composition of individual ratios from a network trained only for a single context observation. An equivalent extension can be done for NLE (Geffner et al., 2023). However, these approaches still require MCMC sampling to obtain posterior samples, which remains a computational bottleneck and requires careful hyperparameter tuning for convergence guarantees to hold.

Recent developments in score-based generative modeling (SBGM) (Ho et al., 2020; Song et al., 2021b) offer a promising alternative. These methods target the gradient of the log probability—known as the score function—used to reverse a diffusion process that transforms random noise into structured samples from the target distribution. A more detailed overview is provided in Section 2.1. SBGM rivals state-of-the-art generative modeling approaches such as generative adversarial networks (GANs)(Goodfellow et al., 2014) and normalizing flows(Papamakarios et al., 2021) on challenging high-dimensional datasets, without requiring adversarial training or special network architectures. This has led to the adoption and increasing popularity of score-based methods in SBI, introduced as Neural Posterior Score Estimation (NPSE) by Sharrock et al. (2022). NPSE is now among the most flexible SBI algorithms and achieves state-of-the-art performance, especially when paired with transformer architectures (Gloeckler et al., 2024).

Crucially, NPSE directly targets the posterior (unlike NLE/NRE), but also supports compositional inference (unlike NPE). Specifically, from the factorization in Equation (1), the "tall posterior score" can be constructed as a *sum of individual posterior scores*. F-NPSE, recently proposed by Geffner et al. (2023), leverages this idea by composing scores from a network trained on single-observation contexts. Like Hermans et al. (2020) for NRE, it exploits the amortization of NPSE to approximate the tall data posterior without requiring an augmented dataset, and naturally adapts to variable context sizes. However, a major drawback of this approach is that the diffusion process from the composed score is unknown, reintroducing the need of MCMC to sample from the posterior via an annealed Langevin procedure. We discuss this further in Section 2.3.

In this work, we propose a new sampling algorithm that *retains the amortized and compositional benefits of* F-NPSE, *while eliminating the need for Langevin dynamics*. By explicitly approximating the diffusion process of the tall-data posterior, our method enables faster and more stable inference. We demonstrate its superiority over F-NPSE—in both, accuracy and numerical stability—through several numerical experiments: two Gaussian toy models for which all quantities of interest are known analytically, and multiple examples from the SBI benchmark. Finally, we apply our method to invert a complex model from computational neuroscience, demonstrating its scalability to challenging real-world problems.

Section 2 reviews score-based generative models, their application to SBI, and the F-NPSE method. Section 3 introduces the mathematical foundations of our approach and presents the proposed algorithms. Section 4 reports experimental results, and Section 5 concludes with a discussion of findings and future perspectives.

## 2 Background

### 2.1 Score based generative models (SBGM)

The goal of SBGM is to estimate an unknown target data distribution $p_{\mathsf{data}}$ from i.i.d. samples with the help of a forward diffusion process that adds noise to the training data. The main idea is to approximately sample the target distribution by solving the associated *backward diffusion process*. This procedure requires estimating the score functions of the diffused data distributions for different levels of added noise.

**Forward diffusion process (data → noise).** Formally, the idea is to construct a sequence of distributions $\{p_t\}_{t\in[0,T]}$ that defines increasingly noisy versions the target distribution $p_{\mathsf{data}}$, by convolving the data distribution with a known *forward kernel* $q_{t|0}$:

$$p_0 = p_{\mathsf{data}}, \quad p_t(\theta_t) = \int q_{t|0}(\theta_t|\theta_0)p_{\mathsf{data}}(\theta_0)\mathrm{d}\theta_0, \quad \forall t \in [1,T] . \tag{2}$$

In this work, we focus on the variance preserving (VP) framework (Ho et al., 2020; Yang et al., 2023) in which the forward kernel is defined by $q_{t|0}(\theta_t|\theta_0) = \mathcal{N}(\theta_t; \sqrt{\alpha_t}\theta_0, v_t\mathbf{I}_m)$, with $\{\alpha_t\}_{t\in[1:T]} \in [0,1]^T$ is a decreasing sequence of time-dependent scale factors and $v_t = 1 - \alpha_t$ determines the amount of added noise.[2]

**Score estimation.** Following (Vincent, 2011), we can learn the scores $\nabla_{\theta_t} \log p_t(\theta_t)$ of each noisy distribution $p_t$ via a neural network $\mathrm{s}_\phi$, by minimizing the denoising score-matching (DSM) loss

$$\mathcal{L}_{\mathsf{DSM}}(\phi) = \sum_{t=1}^{T} \gamma_t^2 \mathbb{E}_{\Theta_0 \sim p_{\mathsf{data}}, \Theta_t \sim q_{t|0}(\cdot|\Theta_0)} \left[ \|\mathrm{s}_\phi(\Theta_t, t) - \nabla_{\Theta_t} \log q_{t|0}(\Theta_t|\Theta_0)\|^2 \right], \tag{3}$$

where $\gamma_t^2$ is a weighting function. This loss is completely tractable, as the forward kernel can be easily sampled from and it's score function is available in closed form: given a sample from the training set $\Theta_0 \sim p_{\mathsf{data}}$ and $\epsilon_t \sim \mathcal{N}(0, \mathbf{I}_m)$, we have $\Theta_t = \sqrt{\alpha_t}\Theta_0 + \sqrt{v_t}\epsilon_t \sim q_{t|0}$ and $\nabla_{\Theta_t} \log q_{t|0}(\Theta_t|\Theta_0) = -\frac{1}{v_t}(\Theta_t - \sqrt{\alpha_t}\Theta_0)$ .

**Backward diffusion process (noise → data).** Once the score estimator $\mathrm{s}_\phi(\theta_t, t) \approx \nabla_{\theta_t} \log p_t(\theta_t)$ is trained, the goal is to draw *backwards* starting from the noise distribution $\mathcal{N}(0, \mathbf{I}_m)$ approximating $p_T$ to obtain samples approximately distributed according to $p_{\mathsf{data}}$ at time $t = 0$:

$$\Theta_T \sim \mathcal{N}(0, \mathbf{I}_m) \underset{s_\phi(\Theta_T, T)}{\to} \Theta_{T-1} \underset{s_\phi(\Theta_{T-1}, T-1)}{\to} \cdots \underset{s_\phi(\Theta_1, 1)}{\to} \Theta_0 . \tag{4}$$

This represents the "generative" part in SBGM: the score directly drives the reverse dynamics, turning noisy inputs into progressively cleaner samples. Different ways to perform this backward sampling include the use of annealed Langevin dynamics (Song & Ermon, 2019), stochastic differential equations (Song et al., 2021b), or ordinary differential equations (Karras et al., 2022).

In this work, we follow the approach proposed in Song et al. (2021a), which yields the denoising diffusion implicit models (DDIM) sampler. DDIM provides a deterministic update rule to directly map $p_t$ to $p_{t-1}$ using the learned score $s_\phi$ from (3). Formally, this consists in Gaussian transition kernels of the form

$$q_{\phi,t-1|t}(\theta_{t-1}|\theta_t) = \mathcal{N}\left(\theta_{t-1}; \boldsymbol{\mu}_{\phi,t}(\theta_t), \sigma_t^2\mathbf{I}_m\right) , \tag{5}$$

where $\{\sigma_t \in (0, v_{t-1}^{1/2})\}_{t\in[1:T-1]}$ and the mean depends explicitly on the score network:

$$\boldsymbol{\mu}_{\phi,t}(\theta_t) := \frac{1}{\sqrt{\alpha_t}}\left(\theta_t + v_t\mathrm{s}_\phi(\theta_t, t)\right) . \tag{6}$$

Finally, composing these kernels yields the DDIM backward Markov chain:

$$p_{\phi,0:T}(\theta_{0:T}) = p_T(\theta_T) \prod_{t=1}^{T} q_{\phi,t-1|t}(\theta_{t-1}|\theta_t) . \tag{7}$$

Note that this only approximates the true reverse diffusion dynamics: the exact backward kernel involves intractable expectations over $\theta_0$, which DDIM replaces with score-based estimates (as shown in Appendix C). Empirically, DDIM has been shown to produce high-quality samples with fewer steps than alternative stochastic samplers, making it particularly attractive in settings where efficiency and stability are crucial.

---

[2]Under weak conditions ($p_{\mathsf{data}}$ has finite second moment), one can show that $\mathrm{KL}(p_T||\mathcal{N}(0, \mathbf{I}_m)) \to 0$ as $T$ grows.

## 2.2 Neural Posterior Score Estimation (NPSE)

Neural Posterior Score Estimation (NPSE) (Sharrock et al., 2022) adapts score-based generative modeling to simulation-based inference. The goal is now to generate new samples from the posterior $p_{\mathsf{data}} := p(\theta|x)$. This is achieved by extending the score network $s_\phi(\theta_t, x, t)$ to take $x$ as an additional input, trained to approximate the score $\nabla_{\theta_t} \log p_t(\theta_t|x)$. Training follows the same denoising score matching principle used in unconditional settings, but now minimizes the loss in expectation over both $\theta$ and $x$:

$$\mathcal{L}_{\mathsf{NPSE}}(\phi) = \sum_{t=1}^{T} \gamma_t^2 \mathbb{E}_{(\Theta_0, X) \sim p(\theta, x), \Theta_t \sim q_{t|0}(\cdot|\Theta_0, X)} \left[ \|s_\phi(\Theta_t, X, t) - \nabla_{\Theta_t} \log q_{t|0}(\Theta_t|\Theta_0, X)\|^2 \right]. \tag{8}$$

The key insight is that this only requires to draw training pairs from the joint distribution $p(\theta, x) = p(\theta)p(x|\theta)$, which is readily available in SBI via simulation. Convergence of this objective to the true conditional score function has been shown by Batzolis et al. (2021). Once trained, NPSE enables amortized inference by generating samples from the posterior $p(\theta|x^\star)$ for any new observation $x^\star$ by plugging the learned score $s_\phi$ into any score-based sampler (e.g. DDIM).

## 2.3 Factorized Neural Posterior Score Estimation (F-NPSE)

The F-NPSE method proposed in Geffner et al. (2023) defines a sequence of distributions

$$\nabla_{\theta_t} \log \varrho_t(\theta_t \mid x_{1:n}^\star) = (1 - n)(1 - t)\nabla_{\theta_t} \log \lambda(\theta_t) + \sum_{j=1}^{n} s_\phi(\theta_t, x_j, t), \tag{9}$$

which, at $t = 0$, coincides with the score of the tall data posterior from Equation (1). Note, however, that this sequence does not correspond to the true diffusion process, as defined in (2), i.e. $\varrho_t(\theta \mid x_{0:n}^\star) \neq p_t(\theta \mid x_{0:n}^\star)$, for $t > 0$. Its appeal lies in the compositional structure of the score: it enables inference from a single amortized score network (here $s_\phi$), trained on individual observations, eliminating the need for augmented datasets or retraining. F-NPSE has shown to outperform competing SBI methods for *tall data* settings (based on NPE, NLE, or NRE), achieving the best trade-off between sample efficiency and error accumulation as $n$ grows.

The main drawback is sampling: since the composed score does not correspond to any known forward diffusion, deterministic samplers such as DDIM cannot be applied. F-NPSE therefore relies on annealed Langevin dynamics[3], which is sensitive to hyperparameters and can require many iterations for convergence. This makes Langevin the bottleneck of F-NPSE, and motivates the need for alternatives that enable faster and more stable sampling.

# 3 Diffusion Posterior Sampling for tall data

In this section, we propose a new algorithm that approximately samples the tall data posterior, using only individual scores obtained for each observation from a previously trained amortized NPSE. The novelty of our method lies in the tractable computation of the scores associated with the diffusion process of the factorized tall posterior, thereby eliminating the need of costly and unstable Langevin steps used in F-NPSE.

- (Section 3.1) First, we derive a formula for the score of the diffused factorized tall data posterior.

- (Section 3.2) We then show how we can compute this score, by introducing a new approximation of the considered backward diffusion process.

- (Section 3.3) We propose two different algorithms to efficiently implement this approximation.

The resulting approximate score can then directly be plugged into deterministic score-based samplers, such as DDIM, to infer the tall data posterior.

---

[3]Langevin dynamics provide an MCMC procedure to approximate the backward diffusion process and sequentially generate samples for each $\varrho_{t-1}$ from $\varrho_t$ (Song & Ermon, 2019). It only requires access to the score function, not closed-form expressions of the backward kernels as in deterministic samplers like DDIM.

### 3.1 Exact computation of the tall data posterior score

Let $x_{1:n}^\star = (x_1^\star, \ldots, x_n^\star)$ be i.i.d. observations. Our goal is to sample from the tall data posterior $p(\theta \mid x_{1:n}^\star)$ via the DDIM backward Markov chain defined in Equation (34), while only relying on a score estimate $s_\phi(\theta_t, x, t)$ of $\nabla_{\theta_t} \log p_t(\theta_t \mid x)$, i.e. of the diffused posterior *for a single context observation* $x$. To do so, we need a closed-form expression of the diffused tall data posterior score $\nabla_{\theta_t} \log p_t(\theta_t \mid x_{1:n}^\star)$ that writes as a function of the individual posterior scores $\nabla_{\theta_t} \log p_t(\theta_t \mid x_j^\star)$, for every $j \in [1, n]$.

Using the factorized expression from (1) in the definition of the diffused tall posterior from (2), we can write

$$
\begin{aligned}
p_t(\theta_t \mid x_{1:n}^\star) &= \int p(\theta_0 \mid x_{1:n}^\star) q_{t|0}(\theta_t|\theta_0) \mathrm{d}\theta_0 \\
&\propto \int \left( \lambda(\theta_0)^{1-n} \prod_{j=1}^n p(\theta_0 \mid x_j^\star) \right) q_{t|0}(\theta_t|\theta_0) \mathrm{d}\theta_0 \,.
\end{aligned}
\tag{10}
$$

Given the diffused prior $p_t^\lambda(\theta_t) = \int \lambda(\theta_0) q_{t|0}(\theta_t|\theta_0) \mathrm{d}\theta_0$ and the diffused individual posteriors $p_t(\theta_t \mid x_j^\star)$, we now introduce the following backward transition kernels obtained via Bayes' rule:

$$
q_{0|t}^\lambda(\theta_0|\theta_t) = \frac{\lambda(\theta_0) q_{t|0}(\theta_t|\theta_0)}{p_t^\lambda(\theta_t)} \quad \text{and} \quad q_{0|t}(\theta_0|\theta_t, x) = \frac{p(\theta_0 \mid x) q_{t|0}(\theta_t|\theta_0)}{p_t(\theta_t \mid x)} \,.
\tag{11}
$$

Rearranging terms in Equation (10) and using (11) yields

$$
\begin{aligned}
p_t(\theta_t \mid x_{1:n}^\star) &\propto \int \left( \lambda(\theta_0) q_{t|0}(\theta_t|\theta_0) \right)^{1-n} \prod_{j=1}^n p(\theta_0 \mid x_j^\star) q_{t|0}(\theta_t|\theta_0) \mathrm{d}\theta_0 \\
&= L_\lambda(\theta_t, x_{1:n}^\star) p_t^\lambda(\theta_t)^{1-n} \prod_{j=1}^n p_t(\theta_t \mid x_j^\star) \,,
\end{aligned}
$$

with $L_\lambda(\theta_t, x_{1:n}^\star) = \int q_{0|t}^\lambda(\theta_0|\theta_t)^{1-n} \prod_{j=1}^n q_{0|t}(\theta_0|\theta_t, x_j^\star) \mathrm{d}\theta_0$. The corresponding score writes

$$
\nabla_{\theta_t} \log p_t(\theta_t|x_{1:n}^\star) = (1-n) \nabla_{\theta_t} \log p_t^\lambda(\theta_t) + \sum_{j=1}^n \nabla_{\theta_t} \log p_t(\theta_t|x_j^\star) + \nabla_{\theta_t} \log L_\lambda(\theta_t, x_{1:n}^\star) \,.
\tag{12}
$$

The first two terms in the above equation are tractable: the prior score can be computed analytically in most cases[4] and the single posterior scores are approximated by evaluating the learned score model $s_\phi(\theta_t, x, t)$ at every $x_j^\star$. This leaves us with the last term: the score of $L_\lambda(\theta_t, x_{1:n}^\star)$, which involves the backward kernels of the prior and each individual posterior. It is intractable and needs to be approximated. Note that this term is missing in the score formula (9) from F-NPSE and is the reason why Langevin corrector steps are required. More intuition on the influence of this correction term is given in the following section and Appendix B.

### 3.2 Second order approximation of the backward diffusion process

In the previous section, we derived a closed-form expression of the tall data posterior score. We now show how to compute it efficiently. The difficulty lies in the last term of (12), the score of $L_\lambda(\theta, x_{1:n}^\star)$. It involves integrating over all backward kernels, and cannot be computed in closed form. To handle this, we approximate these kernels using Gaussian distributions, following the Tweedie framework (Boys et al., 2023):

$$
\hat{q}_{0|t}^\lambda(\theta_0|\theta_t) = \mathcal{N}(\theta_0; \boldsymbol{\mu}_{t,\lambda}(\theta_t), \Sigma_{t,\lambda}(\theta_t)) \quad \text{and} \quad \hat{q}_{0|t}(\theta_0|\theta_t, x_j^\star) = \mathcal{N}(\theta_0; \boldsymbol{\mu}_{t,j}(\theta_t), \Sigma_{t,j}(\theta_t)) \,,
\tag{13}
$$

where the means and covariance matrices are the ones of the backward processes respectively associated with the diffused prior $p_t^\lambda$ and each diffused individual posterior $p_t(\theta_t \mid x_j^\star)$. Specifically, Boys et al. (2023) show

---

[4]See Appendix D for the Gaussian and Uniform cases. If not analytically computable, the prior score can be learned via the classifier-free guidance approach, at the same time as the posterior score, as shown in Appendix L.1.

that $\boldsymbol{\mu}_t(\theta_t) = \mathbb{E}\left[\theta_0 \mid \theta_t\right]$ and $\Sigma_t(\theta_t) = Cov(\theta_0 \mid \theta_t)$ are functions of the score of $p_t$ and its derivatives, which are computationally tractable. Hence, this choice has two advantages: (i) it preserves the local mean/variance structure of the true backward diffusion process, and (ii) it yields a tractable formula for the correction term.

In what follows, Lemma 3.1 shows that, under this approximation, the correction term $L_\lambda$ simplifies to a product of Gaussian distributions. After taking logarithms, this product becomes a tractable sum that integrates seamlessly into the score expression of Equation (12). Further simplifications then yield our final expression of the approximate tall posterior score formula, stated in Lemma 3.2, which forms the basis of an efficient and stable implementation. Full proofs are postponed to Appendix A.

**Lemma 3.1.** *Let $\Lambda(\theta) = \sum_{j=1}^{n} \Sigma_{t,j}^{-1}(\theta) + (1-n)\Sigma_{t,\lambda}^{-1}(\theta)$ and assume it is positive definite. The approximate log-correction term is defined using (13) and can be written as a linear combination of Gaussian log-factors:*[5]

$$\log \hat{L}_\lambda(\theta_t, x_{1:n}^\star) := \log \int \hat{q}_{0|t}^\lambda(\theta_0|\theta_t)^{1-n} \prod_{j=1}^{n} \hat{q}_{0|t}(\theta_0|\theta_t, x_j^\star) \mathrm{d}\theta_0 \tag{14}$$

$$= \sum_{j=1}^{n} \zeta_j(\theta_t) + (1-n)\zeta_\lambda(\theta_t) - \zeta_{\mathsf{all}}(\theta_t), \tag{15}$$

*where $\zeta_k = \zeta(\boldsymbol{\mu}_{t,k}, \Sigma_{t,k}) = -\frac{1}{2}\left(m\log 2\pi - \log|\Sigma_{t,k}^{-1}| + \boldsymbol{\mu}_{t,k}^\top \Sigma_{t,k}^{-1}\boldsymbol{\mu}_{t,k}\right)$, for $k \in \{j = 1, \ldots, n; \lambda\}$ and $\zeta_{\mathsf{all}} = \zeta(\Lambda^{-1}\boldsymbol{\eta}, \Lambda^{-1})$ with $\boldsymbol{\eta} = \sum_{j=1}^{n} \Sigma_{t,j}^{-1}\boldsymbol{\mu}_{t,j} + (1-n)\Sigma_{t,\lambda}^{-1}\boldsymbol{\mu}_{t,\lambda}$.*

**Lemma 3.2.** *Under the same assumptions as in Lemma 3.1, and using the approximate log-correction term from (18), the tall posterior score in (12) can be approximated as*

$$\nabla_{\theta_t} \log p_t(\theta_t \mid x_{1:n}^\star) \approx \Lambda(\theta_t)^{-1} \left( \sum_{j=1}^{n} \Sigma_{t,j}^{-1}(\theta_t) \nabla_{\theta_t} \log p_t(\theta_t \mid x_j^\star) + (1-n)\Sigma_{t,\lambda}^{-1}(\theta_t) \nabla_{\theta_t} \log p_t^\lambda(\theta_t) \right) + F, \tag{16}$$

*where $F = F(\theta_t, x_{1:n}^\star) = 0$ if for all $1 \le j \le n$, $\nabla_{\theta_t}\Sigma_{t,j}(\theta_t) = 0$ and $\nabla_{\theta_t}\Sigma_{\lambda,t}(\theta_t) = 0$.*

Formula (16) defines our approximation of the tall posterior score and depends solely on the individual prior and posterior scores (which are known analytically or estimated via NPSE), and $F$ (explicited in the proof in Appendix A). By enforcing constant covariance matrices $\Sigma_{t,\lambda}$ and $\Sigma_{t,j}$, the residual $F$-term vanishes. This gives us a tractable and practical formula, which reduces to a precision-weighted version of the score formula (9) from F-NPSE, enabling the use of deterministic samplers such as DDIM.

**Remark.** An alternative deterministic sampler is proposed in Geffner et al. (2023) (Appendix D), which also avoids Langevin dynamics by composing Gaussian reverse transitions $q_{t-1|t}$. However, it imposes a shared isotropic variance, whereas our approximation preserves the covariance structure of every single posterior and the prior. In particular, our approach is exact when individual posteriors and the prior are Gaussian, while theirs is not unless all covariances are isotropic. A detailed theoretical comparison is provided in Appendix M.1, highlighting the advantage of our proposal in faithfully approximating the true reverse dynamics.

### 3.3 Algorithms

We now turn to the practical side: computing the covariance matrices $\Sigma_{t,j}(\theta_t)$ in a way that ensures they remain constant. We present two different strategies, which lead to two algorithms implementing our approximate tall posterior score from (16): `JAC` (Algorithm 1) and `GAUSS` (Algorithm 2). For some prior choices (e.g. Gaussian) $\Sigma_{t,\lambda}$ is by construction constant and can be computed analytically (see Appendix D), otherwise, we use the same strategy as for $\Sigma_{t,j}$, represented by the `prior_fn` function in both algorithms.

---

[5]The Gaussian log-factors (denoted by the zeta functions $\zeta_j, \zeta_\lambda, \zeta_{\mathsf{all}}$) represent the (approximate) Gaussian contributions of the backward diffusion for each single posterior and the prior, and their global contribution.

**Jacobian approximation (`JAC`, Algorithm 1).** Following Boys et al. (2023), it can be shown that $\Sigma_{t,j}(\theta_t) = \frac{v_t}{\sqrt{\alpha_t}}\nabla_{\theta_t}\boldsymbol{\mu}_{t,j}(\theta_t)$, with $\boldsymbol{\mu}_{t,j}(\theta_t)$ defined via the posterior score as in (6). $\Sigma_{t,j}$ can therefore be approximated by taking the Jacobian of the learned score $s_\phi(\theta_t, x_j^\star, t)$ (in orange). As in (Boys et al., 2023), we do not propagate gradients through $\Sigma_{t,j}$, rendering $F(\theta_t, x_{1:n}^\star) = 0$. But this approach has two main drawbacks. First, we need to calculate a $m \times m$ matrix which is prohibitive for large $m$. Second, we have to take the derivative w.r.t. the inputs of the score neural network, which is known to be unstable.

**Gaussian approximation (`GAUSS`, Algorithm 2).** As an alternative, we consider a constant Gaussian approximation of the covariance matrix, which automatically gives $F(\theta_t, x_{1:n}^\star) = 0$. Indeed, in the case where $p(\theta_t|x) = \mathcal{N}(\mu_0, \Sigma_0)$, we have that $\Sigma_t = (\Sigma_0^{-1} + \frac{\alpha_t}{v_t}\mathbf{I}_m)^{-1}$, as derived in Appendix D. Note that choosing $\Sigma_0 = \mathbf{I}_m$ results in the approximation proposed by Song et al. (2023). The idea behind `GAUSS` is to use this formula as an approximation of the real covariance $\Sigma_{t,j}$. To do so, we first estimate $\Sigma_{0,j}$ for each $x_j^\star$, by running DDIM with a small number of iterations ($\approx 100$) for each $j$ and computing the empirical covariance matrix of the resulting samples (represented in blue).

---

**Algorithm 1** `JAC`

**Input:** $\theta_t$, $x_{1:n}^\star$, $t$, prior_fn
**Output:** $s_{1:n}$
$\Sigma_{t,\lambda}^{-1}, s_\lambda \leftarrow$ prior_fn$(\theta_t, t)$
**for** $j \leftarrow 1$ to $n$ **do**
    $s_j \leftarrow s_\phi(\theta_t, x_j^\star, t)$
    $\hat{\Sigma}_{t,j}^{-1} \leftarrow \frac{\alpha_t}{v_t}\left(\mathbf{I}_m + v_t\nabla_{\theta_t}s_\phi(\theta_t, x_j^\star, t)\right)^{-1}$
**end for**
$\Lambda \leftarrow (1-n)\Sigma_{t,\lambda}^{-1} + \sum_{j=1}^n \hat{\Sigma}_{t,j}^{-1}$
$\tilde{s}_{1:n} \leftarrow (1-n)\Sigma_{t,\lambda}^{-1}s_\lambda + \sum_{j=1}^n \hat{\Sigma}_{t,j}^{-1}s_j$
$s_{1:n} \leftarrow$ LinSolve$(\Lambda, \tilde{s}_{1:n})$

**Algorithm 2** `GAUSS`

**Input:** $\theta_t$, $x_{1:n}^\star$, $t$, $\hat{\Sigma}_{1:n}$, prior_fn
**Output:** $s_{1:n}$
$\Sigma_{t,\lambda}^{-1}, s_\lambda \leftarrow$ prior_fn$(\theta_t, t)$
**for** $j \leftarrow 1$ to $n$ **do**
    $s_j \leftarrow s_\phi(\theta_t, x_j^\star, t)$
    $\hat{\Sigma}_{t,j}^{-1} \leftarrow \hat{\Sigma}_j^{-1} + \frac{\alpha_t}{v_t}\mathbf{I}_m$
**end for**
$\Lambda \leftarrow (1-n)\Sigma_{t,\lambda}^{-1} + \sum_{j=1}^n \hat{\Sigma}_{t,j}^{-1}$
$\tilde{s}_{1:n} \leftarrow (1-n)\Sigma_{t,\lambda}^{-1}s_\lambda + \sum_{j=1}^n \hat{\Sigma}_{t,j}^{-1}s_j$
$s_{1:n} \leftarrow$ LinSolve$(\Lambda, \tilde{s}_{1:n})$

---

The approximate score obtained via `JAC` or `GAUSS` can now directly be plugged into deterministic score-based samplers, such as DDIM, to infer the tall posterior. This should enable faster and more stable inference compared to Langevin sampling. We verify this statement with experimental results in the following section.

## 4 Experiments

We investigate the performance of `JAC` and `GAUSS` compared to F-NPSE, referred to as `LANGEVIN`, on different tasks with increasing difficulty:

- (Section 4.1) two Gaussian toy models for which all quantities of interest are known analytically

- (Section 4.2) various SBI benchmark examples, where the score has to be learned via NPSE

- (Section 4.3) the Jansen and Rit Neural Mass Model, a challenging real-world example

For `JAC` and `GAUSS`, we infer the tall posterior by plugging the scores obtained via Algorithms 1 and 2 into the DDIM sampler defined in Section 2.1. For F-NPSE, we use the unadjusted Langevin algorithm (ULA) from (Geffner et al., 2023) with $L = 5$ Langevin steps per time step and $\tau = 0.5$. In all cases, we use a uniform time schedule $\{t_i = i/T\}_{i=1}^T$. We evaluate the quality of the inferred tall posteriors using distance-based metrics, comparing them, when possible, to samples from the true tall posterior. Each subsection details task-specific evaluation setups. Further implementation details can be found in Appendix E and the code reproducing all experiments is available at `https://github.com/JuliaLinhart/diffusions-for-sbi`.

**Remark.** We additionally compare with the deterministic sampler from Geffner et al. (2023), mentioned in the Remark from Section 3.2. Full empirical results are provided in Appendix M.2 and complement the theoretical comparison from Appendix M.1.

## 4.1 Gaussian toy models

We consider two toy examples for which the analytic form of the posterior and corresponding score functions are known: a multivariate `Gaussian` $p(x \mid \theta) = \mathcal{N}(x; \theta, (1-\rho)\mathbf{I}_m + \rho\mathbf{1}_m)$ with correlation factor $\rho = 0.8$, and a Gaussian Mixture Model (`GMM`) $p(x \mid \theta) = 0.5\,\mathcal{N}(x; \theta, 2.25\Sigma) + 0.5\,\mathcal{N}(x; \theta, \Sigma/9)$, where $\Sigma$ is a diagonal matrix with values increasing linearly from 0.6 to 1.4, as in Geffner et al. (2023). Both examples are carried out with a Gaussian prior $\lambda(\theta) = \mathcal{N}(\theta; 0, \mathbf{I}_m)$ with known score. See Appendix D for all analytical formulas.

The following experiments evaluate the speed and robustness to noise of each tall posterior sampling algorithm, for increasing number of observations $n \in [1, 100]$ and parameter dimensions $m \in \{2, 4, 8, 10, 16, 32\}$. We quantify accuracy using the sliced Wasserstein (sW) distance between estimated and true tall posterior samples, over 5 random seeds. Each seed corresponds to a different parameter $\theta^\star \sim \lambda(\theta)$, used to simulate the conditioning observations $x_{1:n}^\star$ for the tall posterior $p(\theta \mid x_{1:n}^\star)$. The true tall posterior is known analytically for `Gaussian` and sampled via Metropolis Adjusted Langevin (MALA) (Roberts & Tweedie, 1996) for `GMM`.

We model the noise by considering the score estimator $\tilde{s}_\phi(\theta_t, x, t) = \nabla_{\theta_t} \log p_t(\theta_t|x) + \epsilon\sqrt{v_t}\; r_\phi(\theta_t, x, \alpha_t)$, where $\nabla_{\theta_t} \log p_t(\theta_t|x)$ is the known posterior score and $r_\phi$ is a randomly initialized neural net with outputs in range $[-1, 1]$. This construction leads to a controlled error of $\epsilon \geq 0$ for the *noise predictor* $-\tilde{s}_\phi(\theta_t, x, t)/\sqrt{v_t}$, which is what we actually optimize when training a score model (see Appendix E). The input $\theta_t$ denotes the forward-diffused parameter, $\theta_t = \sqrt{\alpha_t}\,\theta + \sqrt{1-\alpha_t}\,\varepsilon$, with $\theta \sim \lambda(\theta)$ and $\varepsilon \sim \mathcal{N}(0, I)$.

**Runtimes.** Table 1 displays the total running time and sW for each algorithm on the `Gaussian` example. For the same number of time steps $T$, our algorithm yields smaller sW than the Langevin sampler while accounting for $L = 5$ times less neural network evaluations (one per Langevin steps). Table 4 in Appendix G shows similar results for `GMM`. The average speed-up of our algorithms over all considered number of time steps $T$ and different choices of $\epsilon$ and $n$ are shown in Table 2. It shows approximately the same values for $m = 2, 4, 8, 10, 16, 32$, meaning that the speed-up is not impacted by the data dimension. `GAUSS` is the most efficient algorithm, being 2.5 times faster than the Langevin sampler. `JAC` is approximately 1.8 times faster.

Based on Table 1, we consider an *equivalent time setting* with 400 and 1000 steps for `JAC` / `LANGEVIN` and `GAUSS` respectively. This setting will be used in all subsequent experiments.

| Algorithm | T steps | $\Delta t$ (s) | sW |
|---|---|---|---|
| GAUSS | 50 | 0.45 +/- 0.00 | 0.17 +/- 0.08 |
| JAC | 50 | 0.41 +/- 0.00 | 3.14 +/- 4.12 |
| LANGEVIN | 50 | 0.83 +/- 0.00 | nan +/- nan |
| GAUSS | 150 | 0.90 +/- 0.00 | 0.17 +/- 0.06 |
| JAC | 150 | 1.22 +/- 0.00 | 1.57 +/- 2.33 |
| LANGEVIN | 150 | 2.50 +/- 0.01 | 0.65 +/- 0.42 |
| GAUSS | 400 | 2.04 +/- 0.00 | 0.20 +/- 0.11 |
| JAC | 400 | 3.26 +/- 0.01 | 0.85 +/- 1.20 |
| LANGEVIN | 400 | 6.65 +/- 0.02 | 0.65 +/- 0.43 |
| GAUSS | 1000 | 4.77 +/- 0.01 | 0.22 +/- 0.10 |
| JAC | 1000 | 8.18 +/- 0.03 | 0.25 +/- 0.09 |
| LANGEVIN | 1000 | 16.65 +/- 0.03 | 0.65 +/- 0.42 |

Table 1: Sliced Wasserstein (sW) and runtime $\Delta t$ for T time steps for the `Gaussian` example with $m = 10$, $n = 32$ and $\epsilon = 10^{-2}$. Mean and std over 5 different seeds.

| $m$ | Speed up `GAUSS` | Speed up `JAC` |
|---|---|---|
| 2 | $0.39 \pm 0.01$ | $0.56 \pm 0.00$ |
| 4 | $0.39 \pm 0.01$ | $0.55 \pm 0.00$ |
| 8 | $0.39 \pm 0.01$ | $0.56 \pm 0.00$ |
| 10 | $0.38 \pm 0.01$ | $0.52 \pm 0.00$ |
| 16 | $0.38 \pm 0.01$ | $0.54 \pm 0.00$ |
| 32 | $0.37 \pm 0.01$ | $0.52 \pm 0.00$ |

Table 2: Ratio between the runtime for `GAUSS` and `JAC` w.r.t. `LANGEVIN` for the `Gaussian` example in different dimensions $m$. Averaged over the number of time steps $T \in \{50, 150, 400, 1000\}$, different noise levels $\epsilon \in \{0, 10^{-3}, 10^{-2}, 10^{-1}\}$ and the number of observations $n \in [1, 100]$. Mean and std over 5 different seeds.

**Robustness to noise.** Figure 2 portrays the effect of the perturbation $\epsilon$ in the posterior approximation for each algorithm, across $n \in [1, 100]$ observations. In the `Gaussian` case, `GAUSS` performs best in all settings — as expected, since the second-order approximations in our method are exact in this case. For `GMM` however, the approximation from Section 3.2 is not exact, i.e. the backward kernel from (11) is not Gaussian. We use this to analyze the effect of our approximation while controlling the score estimation error. `GAUSS` remains competitive with `LANGEVIN`, outperforming it for small $n$ and matching it for $n > 90$. In both models, `JAC` is extremely accurate in the noise-free setting ($\epsilon = 0$), but quickly becomes unstable as noise increases.

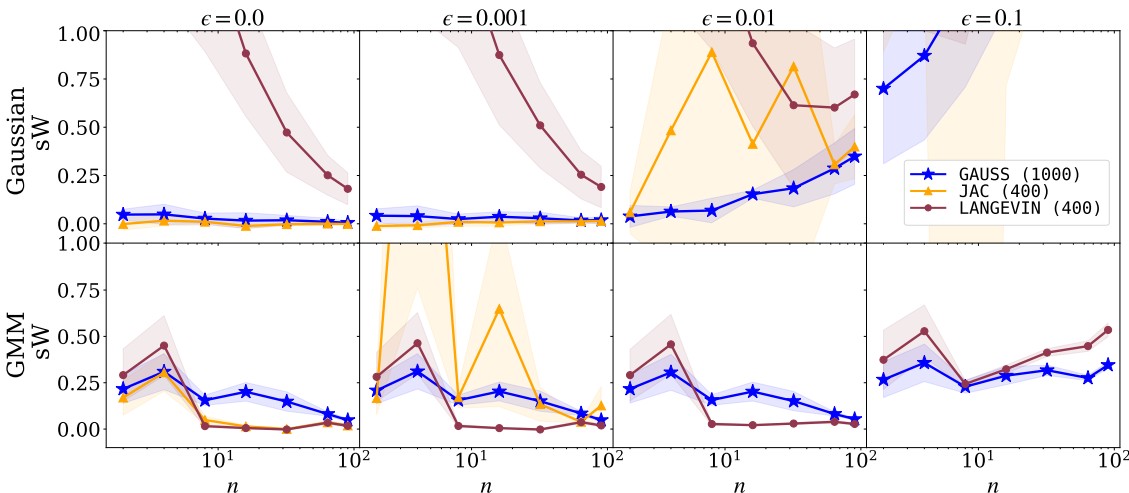

Figure 2: Sliced Wasserstein (sW) distance as a function of $n$ and for increasing noise levels $\epsilon$. Results are shown for both Gaussian toy examples with $m = 10$. Mean and std over 5 different seeds.

We include in Figures 8 and 9 in Appendix G a complete analysis for the `Gaussian` example, across all dimensions $m$. They confirm the results of Figure 2, highlighting the precision of `JAC` in the non-perturbed case and its instability otherwise. They also show the superior robustness of `GAUSS` in high dimensions.

Overall, the above experiments show that our proposal outperforms the Langevin sampler from F-NPSE in terms of speed and robustness to noise. Furthermore, they suggest that `GAUSS` offers a good trade-off between precision and robustness, hence a better algorithm choice than `JAC` in SBI settings, where the posterior score is unknown and has to be learned. We investigate the validity of this statement in the following section.

## 4.2 Benchmark SBI examples

We consider three examples from the popular SBI benchmark (Lueckmann et al., 2021), with different $\theta$- and $x$-space dimensions (resp. $m$ and $d$), and different posterior structures:

- `SLCP` ($m = 5$, $d = 8$): Uniform prior and Gaussian simulator, whose mean and covariance are non-linear functions of the input parameters $\theta$. Multi-modal posterior.

- `SIR` ($m = 2$, $d = 10$): Log-Normal prior and simulator based on a set of differential equations that outputs samples from a Binomial distribution. Uni-modal posterior.

- `Lotka-Volterra` ($m = 4$, $d = 20$): Log-Normal prior and simulator based on a set of differential equations that outputs samples from a Log-Normal distribution. Uni-modal posterior.

The score corresponding to each prior is analytically computable, as done in Appendix D. However, contrarily to the toy models from Section 4.1, the analytical posterior score is not available and has to be learned via score-matching. The goal is to evaluate the robustness of our sampling algorithms in this learning setting. We train a simple MLP on $N_{\text{train}}$ samples over $5\,000$ epochs using the Adam optimizer (see Appendix E). According to the *equivalent time setting* from Section 4.1, we use $T = 1000$ steps for `GAUSS` and $T = 400$ steps for `JAC` and `LANGEVIN`. We also consider *clipped* versions, ensuring that the samples at every step stay within the high probability region of a standard Gaussian (truncated to $[-3, 3]$). The goal is to stabilize `JAC` and `LANGEVIN`, that were found less robust than `GAUSS`. This may however slow sampling and introduce bias.

To assess the performance of each sampling algorithm in this learning setting, we evaluate their accuracy for increasing $n \in [1, 8, 14, 22, 30]$ under varying $N_{\text{train}} \in [10^3, 3.10^3, 10^4, 3.10^4]$. Our empirical evaluation samples 25 ground-truth parameters $\theta^\star \sim \lambda(\theta)$ from the prior, used to simulate the conditionning observations $x_j^\star \sim p(x \mid \theta^\star)$ for $j = 1, \dots, n$ for the tall posterior $p(\theta \mid x_{1:n}^\star)$. For all three tasks, reference samples from

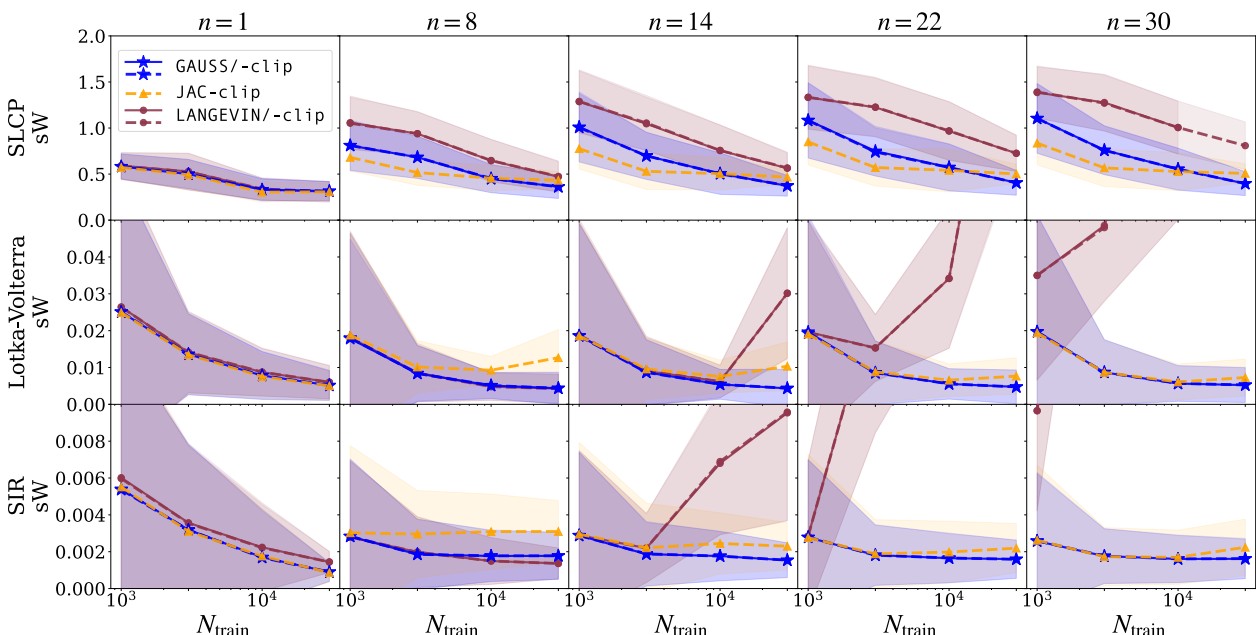

Figure 3: Sliced Wasserstein (sW) distance as a function of $N_{\text{train}}$ and for increasing $n$, between samples obtained by **GAUSS**, **JAC** and **LANGEVIN**, and the true tall posterior $p(\theta \mid x^{\star}_{1,n})$. Mean and std over 25 seeds.

the true tall posterior are obtained via MCMC using `numpyro` (Phan et al., 2019), and used to compute distance-based evaluation metrics. Outliers with values above the 99th percentile (or `NaN`) are excluded.

Figure 3 portrays the sliced Wasserstein (sW) distance for each task as a function of the size $N_{\text{train}}$ of the training set for the score model. Larger values of $N_{\text{train}}$ are expected to yield better score estimates and thus more accurate posterior approximations. This is represented by a decreasing tendency of the sW. Overall, we observe that **GAUSS** outperforms all other algorithms. It yields consistently lower distance values and scales to high $n$ values, compared to **LANGEVIN/-clip** that diverges for $n \geq 14$ for the `Lotka-Volterra` and `SIR` examples. Note that **JAC** diverges as soon as $n > 1$, which is why we didn't include it in the plots. On the other hand, **JAC-clip** is more or less equivalent to **GAUSS**, with slightly better results for `SLCP`.

We complement these results with two additional metrics: Maximum Mean Discrepancy (MMD) and Classifier Two-Sample Test (C2ST), both from the `sbibm` package. Their relevance is motivated by the different posterior structures of the benchmark examples (see Figure 10 in Appendix H): `SLCP` is multimodal, `Lotka-Volterra` is high-dimensional but unimodal, and `SIR` is low-dimensional and smooth. Each metric captures different aspects of the posterior approximation, making them complementary tools for assessing statistical validity. This is summarized in Table 3; full numerical results are in Appendix H. They highlight that, across all metrics and all tasks, our methods significantly outperforms the Langevin sampler.

| Metric | What it measures | Strengths and limitations | Best method for $n > 1$ |
|---|---|---|---|
| **sW** | Geometric alignment of distributions based on sliced 1D projections | Sensitive to multi-modality (`SLCP`); less effective in high-dimensions (`LV`) and in the absence of geometric structure (`SIR`) | **GAUSS** (all tasks), **JAC-clip** (`SLCP`) |
| **MMD** | Moment-based global differences using kernel methods | Effective in detecting smooth deviations and higher order correlations; known to fail in multi-modal settings (`SLCP`) | **GAUSS** (`SLCP`, `LV`), **JAC-clip** (`SLCP`, `SIR`) |
| **C2ST** | Classification accuracy between true and approximate samples | Fast and general validation tool, capturing any inconsistency; can be "too discriminative" (`LV`, `SLCP`) and is harder to interpret | **GAUSS** (`LV`), **JAC-clip** (`SLCP`, `SIR`) |

Table 3: Metric description and results summary. Each metric provides complementary information on the statistical validity of the approximate posterior and is more or less appropriate for the considered benchmark task. The last column displays which method performs best in the tall data setting ($n > 1$).

The results of this section show that our methods outperform the `LANGEVIN` baseline, particularly in scaling to large observation contexts, even in a setting where the posterior score is learned from data. While `GAUSS` is overall the most accurate, the strong performance of `JAC-clip` suggests it is a viable alternative when stability can be ensured. However, a notable trend across tasks is the performance degradation as $n$ increases—as highlighted by in figures of Appendix H. Indeed, the compositional approach introduces a key limitation: approximation errors accumulate as we sum over $n$ evaluations of the learned score model to obtain the tall posterior score. We explore a potential solution to this issue in Appendix L.2 via partially factorized methods, offering a trade-off between simulation cost and number of network evaluations.

Finally, note that we chose to limit our comparison to the F-NPSE baseline, as Geffner et al. (2023) already demonstrated its strong performance relative to other SBI methods. This supports our motivation to improve upon F-NPSE directly, and our results show that the proposed approach is indeed promising. We include results for additional benchmark tasks in Appendix I, confirming the trends observed in this section.

### 4.3  Inverting a non-linear model from computational neuroscience

In this Section, we illustrate the benefit of the tall data setting for challenging real-world applications and consider the Bayesian inversion of the Jansen and Rit Neural Mass Model (JRNMM) (Jansen & Rit, 1995). The output $x(t)$ of this simulator is a time series obtained by taking as input a set of four parameters $\theta = (C, \mu, \sigma, g)$. Appendix J.1 gives a full description of the JRNMM. Importantly, there exists a coupling-effect of parameters $g$ and $(\mu, \sigma)$ on the amplitude of the output signal $x(t)$, meaning that the same observed signal could be generated for different pairs of $g$ and $(\mu, \sigma)$. This indeterminacy makes the inference problem ill-posed (Rodrigues et al., 2021). We now demonstrate how a *tall data* posterior can give more precise information on how to invert the simulator, considering two cases:

- **3D JRNMM:** a simplified setting in which we fix $g = 0$ to lift the indeterminacy on $(\mu, \sigma)$. The tall posterior is expected to concentrate around the true parameter values.

- **4D JRNMM:** the full JRNMM, with coupled parameters $g$ and $(\mu, \sigma)$. The tall posterior should reveal the resulting indeterminacy on $(\mu, \sigma)$.

In each case, we estimate the tall data posterior of the JRNMM using `GAUSS`, `JAC` and `LANGEVIN`, according to the *equivalent time setting* from Section 4.1. The posterior score was estimated via NPSE using a MLP trained on $N_{\text{train}} = 50\,000$ samples from the joint distribution, with a uniform prior placed over the range of physiologically meaningful values of the simulator parameters. See Appendices E and J.2 for further details.

We first evaluate the accuracy of each tall posterior sampling algorithm. This time, no samples from the true posterior are available and previously considered metrics cannot be computed. We therefore use the *local* Classifier-Two-Sample Test ($\ell$-C2ST) (Linhart et al., 2023) and the default implementation from the `sbi` Python package (Álvaro Tejero-Cantero et al., 2020). $\ell$-C2ST compares conditional distributions by training a classifier on a separate calibration set from the joint distribution. The validation results are given and explained in detail in Appendix J.3. They show that `GAUSS` is the only method that passes the test in both, the 3D and 4D cases, confirming the reliability of `GAUSS` in this challenging setting and motivating its use in the remainder of this section.

We now investigate the ability of our tall posterior to concentrate around the true parameters $\theta^\star$ used to simulate the observations $x_{1:n}^\star$ as $n$ increases. This is quantified with the MMD between the posterior marginals and the Dirac distribution $\delta_{\theta^\star}$ at the true parameters $\theta^\star$, computed on $10\,000$ samples obtained with `GAUSS`. Figure 4 shows the results for the simplified 3D JRNMM case. We observe as expected a progressive concentration around $\theta^\star$, with sharper posterior marginals and decreasing MMD. Figure 5 displays the results for the full 4D JRNMM. Here, the tall posterior takes a "sharpened banana" shape: we observe a progressive convergence of the inferred posterior mean towards $\theta^\star$ (black dots and dashed lines), a sharpening around $(C, g)$ and a sustained dispersion along the dimensions of $(\mu, \sigma)$, which explains the non-decreasing MMD. This gives evidence about the indeterminacy caused by the coupling between $g$ and $(\mu, \sigma)$, which is consistent with the results from (Rodrigues et al., 2021).

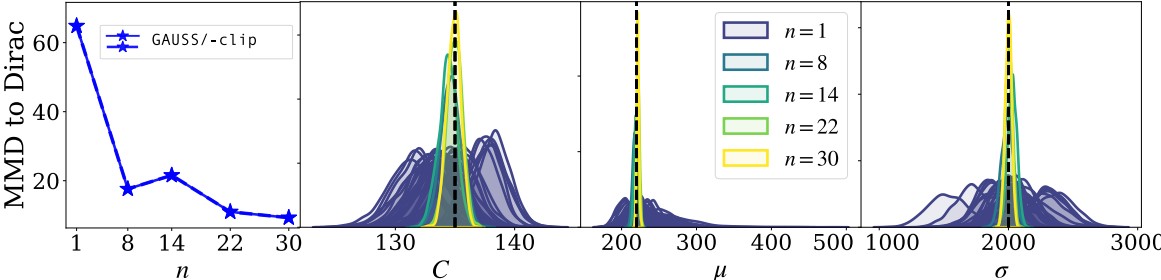

Figure 4: Inference on the 3D JRNMM (fixed $g = 0$) with **GAUSS**. (*Left*): MMD between the marginals of the approximate posterior and the Dirac of the true parameters $\theta^\star$ (black dashed lines). (*Right*): Histograms of the 1D marginals of the inferred posterior for 30 single observations ($n = 1$) and sets $x^\star_{1:n}$ of increasing size.

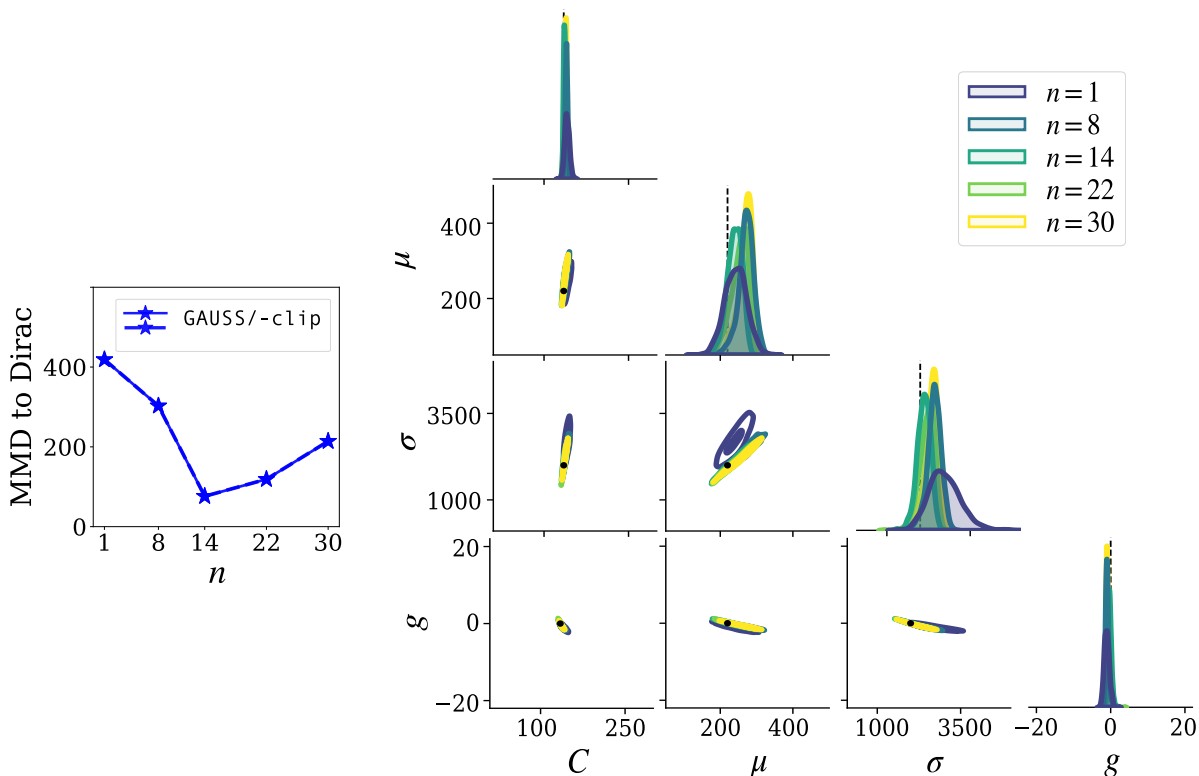

Figure 5: Inference on the 4D JRNMM with **GAUSS**. (*Left*): MMD between the marginals of the approximate posterior and the Dirac of the true parameters $\theta^\star$ (black dots and dashed lines). (*Right*): Histograms of the 1D and 2D marginals of the inferred posterior for observation sets $x^\star_{1:n}$ of increasing size.

## 5    Conclusion

We presented a new sampling approach for tall posterior inference in simulation-based settings, eliminating the need for Langevin dynamics as used in F-NPSE (Geffner et al., 2023), while retaining its amortized and compositional benefits. By explicitly approximating the diffusion process of the factorized tall posterior, our method enables faster and more stable inference through established score-based samplers such as DDIM.

Across toy models and benchmark tasks, our method consistently outperforms the Langevin baseline in terms of speed, accuracy, and scalability with increasing observation set sizes. In particular, **GAUSS** emerges as a robust and consistent choice, while **JAC** achieves highly accurate results when score estimates are reliable.

These findings validate our central motivation: improving compositional inference by modeling the diffusion dynamics of the tall posterior directly. We further demonstrated that `GAUSS` scales effectively to real-world problems, as illustrated by our neuroscience application. This example underscores a key benefit of the tall data setting: aggregating observations can uncover parameter dependencies that are hidden at the single-observation level. This insight aligns with recent hierarchical extensions of amortized inference (e.g. HNPE), suggesting that structured modeling across observations can improve parameter identifiability.

Looking ahead, several research directions emerge. `JAC` has strong potential, but requires better stabilization. Clipping samples improves robustness at the cost of slower, biased inference. Recent work by Gloeckler et al. (2025) suggests promising alternatives. Another interesting direction is the refinement of our second-order approximation. For example, one could try to combine our fixed-covariance strategy with data-driven covariance estimation methods like those from (Rissanen et al., 2025), to better approximate local curvature without sacrificing stability. Finally, while partially factorized methods help mitigate error accumulation—the main limitation of compositional inference methods—further complementary strategies are needed to make our method more scalable, beyond hundreds, to thousands of context observations. Arruda et al. (2025) suggest one way to do that.

Together, our results demonstrate the power of score-based methods for amortized and compositional inference. Beyond tall posteriors, we believe this framework opens new possibilities for modular, hierarchical, and other structured inference tasks across scientific domains.

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

# A    Proofs

## A.1    Proof of Lemma 3.1

**Lemma 3.1.** *Let* $\Lambda(\theta) = \sum_{j=1}^{n} \Sigma_{t,j}^{-1}(\theta) + (1-n)\Sigma_{t,\lambda}^{-1}(\theta)$ *and assume it is positive definite. The approximate log-correction term is defined using (13) and can be written as a linear combination of Gaussian log-factors:*

$$\log \hat{L}_\lambda(\theta_t, x_{1:n}^\star) := \log \int \hat{q}_{0|t}^\lambda(\theta_0|\theta_t)^{1-n} \prod_{j=1}^{n} \hat{q}_{0|t}(\theta_0|\theta_t, x_j^\star) \mathrm{d}\theta_0 \tag{17}$$

$$= \sum_{j=1}^{n} \zeta_j(\theta_t) + (1-n)\zeta_\lambda(\theta_t) - \zeta_{\mathsf{all}}(\theta_t), \tag{18}$$

*where* $\zeta_k = \zeta(\boldsymbol{\mu}_{t,k}, \Sigma_{t,k}) = -\frac{1}{2}\left(m\log 2\pi - \log|\Sigma_{t,k}^{-1}| + \boldsymbol{\mu}_{t,k}^\top \Sigma_{t,k}^{-1}\boldsymbol{\mu}_{t,k}\right)$, *for* $k \in \{j = 1,\ldots,n; \lambda\}$ *and* $\zeta_{\mathsf{all}} = \zeta(\Lambda^{-1}\boldsymbol{\eta}, \Lambda^{-1})$ *with* $\boldsymbol{\eta} = \sum_{j=1}^{n} \Sigma_{t,j}^{-1}\boldsymbol{\mu}_{t,j} + (1-n)\Sigma_{t,\lambda}^{-1}\boldsymbol{\mu}_{t,\lambda}$.

**Proof.** Let $(\boldsymbol{\mu}_k)_{1\le k \le K} \in (\mathbb{R}^m)^K$ and $(\Sigma_k)_{1\le k \le K}$ be covariance matrices in $\mathbb{R}^{m\times m}$. Denote by $p_k$ the Gaussian pdf with mean $\boldsymbol{\mu}_k$ and covariance matrix $\Sigma_k$. Note that

$$p_k : \theta_0 \mapsto \exp\left(\tilde{\zeta}_k + (\Sigma_k^{-1}\boldsymbol{\mu}_k)^\top \theta_0 - \frac{1}{2}\theta_0^\top \Sigma_k^{-1}\theta_0\right),$$

where

$$\tilde{\zeta}_k = -\frac{1}{2}\left(m\log 2\pi - \log|\Sigma_k^{-1}| + \boldsymbol{\mu}_k^\top \Sigma_k^{-1}\boldsymbol{\mu}_k\right).$$

Therefore,

$$\prod_{k=1}^{K} p_k(\theta_0) = \exp\left(\sum_{k=1}^{K} \tilde{\zeta}_k - \tilde{\zeta}_{\mathsf{all}}\right) \exp\left(\tilde{\zeta}_{\mathsf{all}} + \tilde{\boldsymbol{\eta}}^\top \theta_0 - \frac{1}{2}\theta_0^\top \tilde{\Lambda}\theta_0\right)$$

$$= \exp\left(\sum_{k=1}^{K} \tilde{\zeta}_k - \tilde{\zeta}_{\mathsf{all}}\right) \mathcal{N}(\theta_0; \tilde{\Lambda}^{-1}\tilde{\boldsymbol{\eta}}, \tilde{\Lambda}^{-1}),$$

with $\tilde{\boldsymbol{\eta}} = \sum_{k=1}^{K} \Sigma_k^{-1}\boldsymbol{\mu}_k$, $\tilde{\Lambda} = \sum_{k=1}^{K} \Sigma_k^{-1}$, and $\tilde{\zeta}_{\mathsf{all}} = -(m\log 2\pi - \log|\tilde{\Lambda}| + \tilde{\boldsymbol{\eta}}^\top \tilde{\Lambda}^{-1}\tilde{\boldsymbol{\eta}})/2$. We can apply this result to Equation (17) with

$$\boldsymbol{\eta}(\theta) = \sum_{j=1}^{n} \Sigma_{t,j}^{-1}(\theta)\boldsymbol{\mu}_{t,j}(\theta) + (1-n)\Sigma_{t,\lambda}^{-1}(\theta)\boldsymbol{\mu}_{t,\lambda}(\theta),$$

$$\Lambda(\theta) = \sum_{j=1}^{n} \Sigma_{t,j}^{-1}(\theta) + (1-n)\Sigma_{t,\lambda}^{-1}(\theta),$$

since by assumption $\Lambda(\theta)$ is definite positive. This provides the following reformulation of Equation (17):

$$\log \hat{L}_\lambda(\theta_t, x_{1:n}^\star) = \log \int \exp\left(\sum_{j=1}^{n} \zeta_j(\theta_t) + (1-n)\zeta_\lambda(\theta_t) - \zeta_{\mathsf{all}}(\theta_t)\right) \mathcal{N}(\theta_0; \Lambda^{-1}(\theta_t)\boldsymbol{\eta}(\theta_t), \Lambda^{-1}(\theta_t)) \mathrm{d}\theta_0.$$

Finally, because the sum in the above formula does not depend on the integrator variable $\theta_0$, we can take it out of the integral and separate the expression of $\hat{L}_\lambda(\theta, x_{1:n}^\star)$ into two terms:

$$\log \hat{L}_\lambda(\theta_t, x_{1:n}^\star) = \underbrace{\log \int \mathcal{N}(\theta_0; \Lambda^{-1}(\theta_t)\eta(\theta_t), \Lambda^{-1}(\theta_t)) \mathrm{d}\theta_0}_{=0} + \left(\sum_{j=1}^{n} \zeta_j(\theta_t) + (1-n)\zeta_\lambda(\theta_t) - \zeta_{all}(\theta_t)\right) \tag{19}$$

The first term is zero as the integral of the Gaussian p.d.f. over the whole space is equal to 1. This leaves us with the second term and concludes the proof.

**Positive Definiteness of $\Lambda(\theta)$ .** The approximation from Lemma 3.1 is only valid if the matrix $\Lambda(\theta)$ is symmetric positive definite (SPD), i.e. is a valid covariance matrix. In what follows, we investigate what conditions on the prior distribution will ensure such property. Using the partial ordering defined by the convex cone of SPD matrices (Bhatia, 2006), it follows that:

$$\Lambda(\theta) \succ 0 \quad \Longleftrightarrow \quad \sum_{j=1}^{n} \Sigma_{t,j}^{-1}(\theta) + (1-n)\Sigma_{t,\lambda}^{-1}(\theta) \succ 0 \, , \tag{20}$$

$$\Longleftrightarrow \quad \sum_{j=1}^{n} \Sigma_{t,j}^{-1}(\theta) \succ (n-1)\Sigma_{t,\lambda}^{-1}(\theta) \, , \tag{21}$$

$$\Longleftrightarrow \quad \Sigma_{t,\lambda}(\theta) \succ \left( \frac{1}{(n-1)} \sum_{j=1}^{n} \Sigma_{t,j}^{-1}(\theta) \right)^{-1} \, . \tag{22}$$

Note that as $n$ increases, the right-hand side of the inequality in Equation 22 converges to the harmonic mean of the $\Sigma_{t,j}(\theta)$, which helps building an intuition for the correct choice of $\Sigma_{t,\lambda}(\theta)$, the covariance of the backward diffusion kernel associated to the prior distribution $\lambda(\theta)$. For instance, a sufficient choice for $\Lambda(\theta)$ to be SPD, would be to have a $\Sigma_{t,\lambda}(\theta)$ whose associated ellipsoid[6] covers the ellipsoids generated by the covariance matrices $\Sigma_{t,j}(\theta)$, associated to the posteriors $p(\theta \mid x_j^\star)$ for every single observation $x_j^\star$ considered in the tall data inference task. Intuitively, this corresponds to choosing a prior that is broader than the posterior distribution, which is normally the case in Bayesian approaches.

### A.2 Proof of Lemma 3.2

**Lemma 3.2.** *Under the same assumptions as in Lemma 3.1, and using the approximate log-correction term from (18), the tall posterior score in (12) can be approximated as*

$$\nabla_{\theta_t} \log p_t(\theta_t \mid x_{1:n}^\star) \approx \Lambda(\theta_t)^{-1} \left( \sum_{j=1}^{n} \Sigma_{t,j}^{-1}(\theta_t)\nabla_{\theta_t} \log p_t(\theta_t \mid x_j^\star) + (1-n)\Sigma_{t,\lambda}^{-1}(\theta_t)\nabla_{\theta_t} \log p_t^\lambda(\theta_t) \right) + F \, ,$$

*where $F = F(\theta_t, x_{1:n}^\star) = 0$ if $\nabla_{\theta_t}\Sigma_{t,j}(\theta_t) = 0$ for all $1 \leq j \leq n$ and $\nabla_{\theta_t}\Sigma_{\lambda,t}(\theta_t) = 0$.*

***Proof.*** Remember that the full tall posterior score writes

$$\nabla_{\theta_t} \log p_t(\theta \mid x_{1:n}^\star) = (1-n)\nabla_{\theta_t} \log p_t^\lambda(\theta_t) + \sum_{j=1}^{n} \nabla_{\theta_t} \log p_t(\theta_t \mid x_j^\star) + \nabla_{\theta_t} \log L_\lambda(\theta_t, x_{1:n}^\star) \, , \tag{23}$$

We can replace the log-correction term $\log L_\lambda$ by its estimator $\log \hat{L}_\lambda$ from 17 in Lemma 3.1. Now let's compute that term explicitly. Taking the gradient directly gives us:

$$\nabla_{\theta_t} \log \hat{L}_\lambda(\theta_t, x_{1:n}^\star) = \sum_{j=1}^{n} \nabla_{\theta_t}\zeta_j(\theta_t) + (1-n)\nabla_{\theta_t}\zeta_\lambda(\theta_t) - \nabla_{\theta_t}\zeta_{\text{all}}(\theta_t) \, , \tag{24}$$

where we recall that $\zeta(\boldsymbol{\mu}, \Sigma^{-1}) = -\left( m \log 2\pi - \log |\Sigma^{-1}| + \mu^\top \Sigma^{-1} \mu \right)/2.$

---

[6]The ellipsoid $\mathcal{E}_A$ associated with SPD matrix $A$ is defined as $\mathcal{E}_A = \{\mathbf{x} \ : \ \mathbf{x}^\top A^{-1}\mathbf{x} < 1\}.$

First, we compute the terms $\nabla_\theta \zeta_j(\theta)$. The chain rule gives us

$$\nabla_\theta \zeta_j(\theta) = \nabla_\theta \zeta(\boldsymbol{\mu}_{t,j}(\theta), \Sigma_{t,j}^{-1}(\theta))$$
$$= \nabla_\mu \zeta(\boldsymbol{\mu}_{t,j}(\theta), \Sigma_{t,j}^{-1}(\theta))^\top \nabla_\theta \boldsymbol{\mu}_{t,j}(\theta) \; + \; \nabla_{\Sigma^{-1}} \zeta(\boldsymbol{\mu}_{t,j}(\theta), \Sigma_{t,j}^{-1}(\theta)) \nabla_\theta \Sigma_{t,j}^{-1}(\theta) \,,$$

where

$$\nabla_\mu \zeta(\boldsymbol{\mu}, \Sigma^{-1}) = -\Sigma^{-1}\mu \,, \quad \nabla_\Sigma^{-1} \zeta(\boldsymbol{\mu}, \Sigma^{-1}) = \frac{1}{2}\left(\Sigma - \mu\mu^\top\right) \,.$$

Therefore, we obtain

$$\nabla_\theta \zeta_j(\theta) = -\boldsymbol{\mu}_{t,j}(\theta)^\top \Sigma_{t,j}^{-1}(\theta)^\top \nabla_\theta \boldsymbol{\mu}_{t,j}(\theta) + \frac{1}{2}\left(\Sigma_{t,j}(\theta) - \boldsymbol{\mu}_{t,j}(\theta)\boldsymbol{\mu}_{t,j}(\theta)^\top\right)\nabla_\theta \Sigma_{t,j}^{-1}(\theta) \,.$$

Note that for $q_{t|0}(\theta_t|\theta_0) = \mathcal{N}(\theta_t; \sqrt{\alpha_t}\theta_0, \upsilon_t \mathbf{I}_m)$ [7] we have that $\nabla_{\theta_t}\boldsymbol{\mu}_{t,j}(\theta_t) = (\sqrt{\alpha_t}/\upsilon_t)\Sigma_{t,j}(\theta_t)$ (Boys et al., 2023), which leads to

$$\nabla_{\theta_t}\zeta_j(\theta_t) = -\frac{\sqrt{\alpha_t}}{\upsilon_t}\boldsymbol{\mu}_{t,j}(\theta_t) + \frac{1}{2}\left(\Sigma_{t,j}(\theta_t) - \boldsymbol{\mu}_{t,j}(\theta_t)\boldsymbol{\mu}_{t,j}(\theta_t)^\top\right)\nabla_{\theta_t}\Sigma_{t,j}^{-1}(\theta_t)$$
$$= -\upsilon_t^{-1}\theta - \nabla_{\theta_t}\log p_t(\theta_t \mid x_j^\star) + \frac{1}{2}\left(\Sigma_{t,j}(\theta_t) - \boldsymbol{\mu}_{t,j}(\theta_t)\boldsymbol{\mu}_{t,j}(\theta_t)^\top\right)\nabla_{\theta_t}\Sigma_{t,j}^{-1}(\theta_t) \,. \tag{25}$$

Formula (25) also applies to $\nabla_{\theta_t}\zeta_\lambda(\theta_t)$ (with $p_t^\lambda, \boldsymbol{\mu}_{t,\lambda}, \Sigma_{t,\lambda}$). Replacing these expressions in (24), yields

$$\nabla_\theta \log \hat{L}_\lambda(\theta_t, x_{1:n}^\star) = -(1-n)\nabla_{\theta_t}\log p_t^\lambda(\theta_t) - \sum_{j=1}^n \nabla_{\theta_t}\log p_t(\theta_t \mid x_j^\star) - \upsilon_t^{-1}\theta_t - \nabla_{\theta_t}\zeta_{\text{all}}(\theta_t)$$
$$+ \frac{1}{2}\left[\sum_{j=1}^n \left(\Sigma_{t,j}(\theta_t) - \boldsymbol{\mu}_{t,j}(\theta_t)\boldsymbol{\mu}_{t,j}(\theta_t)^\top\right)\nabla_{\theta_t}\Sigma_{t,j}^{-1}(\theta_t)\right]$$
$$+ \frac{1-n}{2}\left(\Sigma_{t,\lambda}(\theta_t) - \boldsymbol{\mu}_{t,\lambda}(\theta_t)\boldsymbol{\mu}_{t,\lambda}(\theta_t)^\top\right)\nabla_{\theta_t}\Sigma_{t,\lambda}^{-1}(\theta_t, x_j^\star) \,.$$

Replacing the above formula in (23), the first two terms compensate each other, which gives the following approximation for the score:

$$\nabla_{\theta_t}\log p_t(\theta_t \mid x_{1:n}^\star) \approx -\upsilon_t^{-1}\theta_t - \nabla_{\theta_t}\zeta_{\text{all}}(\theta_t)$$
$$+ \frac{1}{2}\left[\sum_{j=1}^n \left(\Sigma_{t,j}(\theta_t) - \boldsymbol{\mu}_{t,j}(\theta_t)\boldsymbol{\mu}_{t,j}(\theta_t)^\top\right)\nabla_{\theta_t}\Sigma_{t,j}^{-1}(\theta_t)\right] \tag{26}$$
$$+ \frac{1-n}{2}\left(\Sigma_{t,\lambda}(\theta_t) - \boldsymbol{\mu}_{t,\lambda}(\theta_t)\boldsymbol{\mu}_{t,\lambda}(\theta_t)^\top\right)\nabla_{\theta_t}\Sigma_{t,\lambda}^{-1}(\theta_t, x_j^\star) \,.$$

We now compute $\nabla_\theta \zeta_{\text{all}}(\theta)$. By noting that $\zeta_{\text{all}} = \zeta(\Lambda(\theta)^{-1}\boldsymbol{\eta}(\theta), \Lambda(\theta))$, we can follow the same steps as before and obtain

$$\nabla_\theta \zeta_{\text{all}}(\theta) = -\nabla_\theta(\Lambda(\theta)^{-1}\boldsymbol{\eta}(\theta))\boldsymbol{\eta}(\theta) + \frac{1}{2}\left[\mathbf{I}_m - \Lambda(\theta)^{-1}\boldsymbol{\eta}(\theta)\boldsymbol{\eta}(\theta)^\top\right]\Lambda(\theta)^{-1}\nabla_\theta\Lambda(\theta)$$
$$= -\Lambda(\theta)^{-1}\nabla_\theta\boldsymbol{\eta}(\theta)\boldsymbol{\eta}(\theta) - \boldsymbol{\eta}(\theta)^\top\nabla_\theta\Lambda(\theta)^{-1}\boldsymbol{\eta}(\theta) + \frac{1}{2}\left[\mathbf{I}_m - \Lambda(\theta)^{-1}\boldsymbol{\eta}(\theta)\boldsymbol{\eta}(\theta)^\top\right]\Lambda(\theta)^{-1}\nabla_\theta\Lambda(\theta) \,. \tag{27}$$

---

[7]Variance preserving (VP) framework introduced in Section 2.1.

Note now that

$$\nabla_\theta \boldsymbol{\eta}(\theta) = \sum_{j=1}^n \Sigma_{t,j}^{-1}(\theta)\nabla_\theta \boldsymbol{\mu}_{t,j}(\theta) + \boldsymbol{\mu}_{t,j}(\theta)^\top \nabla_\theta \Sigma_{t,j}^{-1}(\theta) + (1-n)\left(\Sigma_{t,\lambda}^{-1}(\theta)\nabla_\theta \boldsymbol{\mu}_{t,\lambda}(\theta) + \boldsymbol{\mu}_{t,\lambda}(\theta)^\top \nabla_\theta \Sigma_{t,\lambda}^{-1}(\theta)\right)$$

$$= \frac{\sqrt{\alpha_t}}{\upsilon_t}\mathbf{I}_m + \sum_{j=1}^n \boldsymbol{\mu}_{t,j}(\theta)^\top \nabla_\theta \Sigma_{t,j}^{-1}(\theta) + (1-n)\boldsymbol{\mu}_{t,\lambda}(\theta)^\top \nabla_\theta \Sigma_{t,\lambda}^{-1}(\theta),$$

and that

$$\sqrt{\alpha_t}\boldsymbol{\eta}(\theta) = \sum_{j=1}^n \Sigma_{t,j}^{-1}(\theta)\left(\theta + \upsilon_t \nabla_\theta \log p_t(\theta \mid x_j^\star)\right) + (1-n)\Sigma_{t,\lambda}^{-1}(\theta)\left(\theta + \upsilon_t \nabla_\theta \log p_t^\lambda(\theta)\right)$$

$$= \Lambda(\theta)\theta + \upsilon_t \left[\sum_{j=1}^n \Sigma_{t,j}^{-1}(\theta)\nabla_\theta \log p_t(\theta \mid x_j^\star) + (1-n)\Sigma_{t,\lambda}^{-1}(\theta)\nabla_\theta \log p_t^\lambda(\theta)\right]. \tag{28}$$

We can use the above formulas to replace $\nabla_\theta \boldsymbol{\eta}(\theta)\boldsymbol{\eta}(\theta)$ in the term $\Lambda(\theta)^{-1}\nabla_\theta \boldsymbol{\eta}(\theta)\boldsymbol{\eta}(\theta)$ of Equation (27). The first term of this new expression is $-\upsilon^{-1}\theta$, which (setting $\theta = \theta_t$) compensates with the first term in the score formula from (26) and finally leaves us with the wanted score approximation:

$$\nabla_{\theta_t} \log p_t(\theta_t \mid x_{1:n}^\star) \approx \Lambda(\theta_t)^{-1}\left(\sum_{j=1}^n \Sigma_{t,j}^{-1}(\theta_t)\nabla_{\theta_t} \log p_t(\theta_t \mid x_j^\star) + (1-n)\Sigma_{t,\lambda}^{-1}(\theta_t)\nabla_{\theta_t} \log p_t^\lambda(\theta_t)\right) + F(\theta_t, x_{1:n}^\star),$$

where

$$F(\theta, x_{1:n}^\star) = \boldsymbol{\eta}(\theta)^\top \nabla\Lambda(\theta)^{-1}\boldsymbol{\eta}(\theta) - \frac{1}{2}\left[\mathbf{I}_m - \Lambda(\theta)^{-1}\boldsymbol{\eta}(\theta)\boldsymbol{\eta}(\theta)^\top\right]\Lambda(\theta)^{-1}\nabla_\theta \Lambda(\theta)$$

$$+ \frac{1}{2}\left[\sum_{j=1}^n \left(\Sigma_{t,j}(\theta) - \boldsymbol{\mu}_{t,j}(\theta)\boldsymbol{\mu}_{t,j}(\theta)^\top\right)\nabla_\theta \Sigma_{t,j}^{-1}(\theta)\right]$$

$$+ \frac{1-n}{2}\left(\Sigma_{t,\lambda}(\theta) - \boldsymbol{\mu}_{t,\lambda}(\theta)\boldsymbol{\mu}_{t,\lambda}(\theta)^\top\right)\nabla_\theta \Sigma_{t,\lambda}^{-1}(\theta, x_j^\star).$$

$F(\theta, x_{1:n}^\star)$ contains all the terms that depend on the gradients of the covariance matrices $\Sigma_{t,j}(\theta)$ and $\Sigma_{t,\lambda}(\theta)$ and is 0 if they are considered constant (see definition of $\Lambda(\theta)$ in Lemma 3.1).

## B    Influence of the correction term

In this Appendix we provide some intuition on the influence of the correction term $\nabla_{\theta_t} \log L_\lambda(\theta_t, x_{1:n}^\star)$ in the backward diffusion process. Following the definition of the backward kernels in equation (11):

- For $t \to 1$: the forward kernel and the diffused data distribution approach the noise distribution: $p_t(\theta_t \mid x) \underset{t \to 1}{\to} \mathcal{N}(\theta_t; 0, \mathbf{I}_m)$ and $q_{t|0}(\theta_t \mid \theta_0) \underset{t \to 1}{\to} \mathcal{N}(\theta_t; 0, \mathbf{I}_m)$. The backward kernel is thus equivalent to the target data distribution:

$$p_{0|t}(\theta_0 \mid \theta_t, x) = \frac{p(\theta_0 \mid x) q_{t|0}(\theta_t \mid \theta_0)}{p_t(\theta_t \mid x)} \underset{t \to 1}{\sim} p(\theta_0 \mid x). \tag{29}$$

  Therefore the backward kernels vary very little with $\theta$, and because they define $\log L_\lambda(\theta_t, x_{1:n}^\star)$ (see eq. (17)), its gradient is close to zero. In other words, $\nabla_{\theta_t} \log L_\lambda(\theta_t, x_{1:n}^\star)$ has no significant impact at the beginning of the backward diffusion (aka. sampling or generative process).

- For $t \to 0$: the denominator is the diffused distribution that gets close to the target data distribution $p_t(\theta_t \mid x) \underset{t \to 0}{\to} p(\theta_0 \mid x)$. Therefore the backward kernel is approximately

$$p_{0|t}(\theta_0 \mid \theta_t, x) = \frac{p(\theta_0 \mid x) q_{t|0}(\theta_t \mid \theta_0)}{p_t(\theta_t \mid x)} \underset{t \to 0}{\sim} q_{t|0}(\theta_t \mid \theta_0) \underset{t \to 0}{\to} \delta_{\theta_0}(\theta_t). \tag{30}$$

  Here, the dependence on $\theta_t$ is convergence to a Dirac function. This means that the gradient of $\log L_\lambda(\theta_t, x_{1:n}^\star)$ will increase during the sampling process and finally explode when $t$ approaches 0. The correction term therefore plays an important role as we approach the target tall data posterior distribution, at the end of the sampling process.

## C  Denoising Diffusion Implicit Models (DDIM)

In this section we derive the DDIM backward Markov chain mentioned in Section 2.

DDIM introduces a set of inference distributions defined for $t \in [1:T]$ as

$$q^{\sigma}_{t-1|t,0}(\theta_t|\theta_0,\theta_{t+1}) = \mathcal{N}\left(\theta_{t-1};\boldsymbol{\mu}_t(\theta_0,\theta_t),\sigma_t^2\mathbf{I}_m\right) ,\tag{31}$$

with $\boldsymbol{\mu}_t(\theta_0,\theta_t) = \sqrt{\alpha_{t-1}}\theta_0 + (\upsilon_{t-1}-\sigma_t^2)^{1/2}(\theta_t - \sqrt{\alpha_t}\theta_0)/\upsilon_t^{1/2}$ and $\sigma = \{\sigma_t \in (0,\upsilon_{t-1}^{1/2})\}_{t\in[1:T-1]}$. Note that $\boldsymbol{\mu}_t$ is chosen so that $q^{\sigma}_{t|0}(\theta_t|\theta_0) = q_{t|0}(\theta_t \mid \theta_0) = \mathcal{N}(\theta_t;\sqrt{\alpha_t}\theta_0,\upsilon_t\mathbf{I}_m)$ (Song et al., 2021a, Lemma 1, Appendix B). This property allows us to write $p_{t-1}(\theta_{t-1}) = \iint q^{\sigma}_{t-1|t,0}(\theta_{t-1}|\theta_t,\theta_0)q_{0|t}(\theta_0|\theta_t)p_t(\theta_t)\mathrm{d}\theta_0\mathrm{d}\theta_t$.

Even though Equation (31) suggests a way of passing from $p_t$ to $p_{t-1}$, it involves an intractable kernel $\int q^{\sigma}_{t-1|t,0}(\theta_{t-1}|\theta_t,\theta_0)q_{0|t}(\theta_0|\theta_t)\mathrm{d}\theta_0$. DDIM approximates the marginal distribution with

$$\hat{p}_{t-1}(\theta_{t-1}) = \int q^{\sigma}_{t-1|t,0}(\theta_{t-1}|\theta_t,\boldsymbol{\mu}_t(\theta_t))p_t(\theta_t)\mathrm{d}\theta_t ,\tag{32}$$

where $\boldsymbol{\mu}_t(\theta_t) = \mathbb{E}_{\Theta_0 \sim q^{\sigma}_{0|t}}[\Theta_0]$, and verifies $\sqrt{\alpha_t}\boldsymbol{\mu}_t(\theta_t) - \theta_t = \upsilon_t\mathbb{E}_{\Theta_0 \sim q^{\sigma}_{0|t}}\left[\nabla \log q_{t|0}(\theta_t|\Theta_0)\right]$. Using the learned score from Equation 3, we can now define the following approximation of $\boldsymbol{\mu}_t(\theta_t)$:

$$\boldsymbol{\mu}_{\phi,t}(\theta_t) := \frac{1}{\sqrt{\alpha_t}}\left(\theta_t + \upsilon_t\mathrm{s}_\phi(\theta_t,t)\right).\tag{33}$$

We finally obtain the following backward Markov chain for DDIM:

$$p_{\phi,0:T}(\theta_{0:T}) = p_T(\theta_T)\prod_{t=1}^{T} q_{\phi,t-1|t}(\theta_{t-1}|\theta_t) ,\tag{34}$$

where $p_T(\theta_T) = p_T(\theta_T)$, $q_{\phi,t-1|t}(\theta_{t-1}|\theta_t) = q^{\sigma}_{t-1|t,0}(\theta_{t-1}|\theta_t,\boldsymbol{\mu}_{\phi,t}(\theta_t))$ and $q_{\phi,0|1}(\theta_0|\theta_1) = \mathcal{N}(\theta_0;\boldsymbol{\mu}_{\phi,1}(\theta_1),\sigma_0^2\mathbf{I}_m)$, with $\sigma_0 > 0$ a free parameter.

# D Analytical formulas for score and related quantities

## D.1 Gaussian case

The considered Bayesian Inference task is to estimate the mean $\theta \in \mathbb{R}^m$ of a Gaussian simulator model $p(x \mid \theta) = \mathcal{N}(x; \theta, \Sigma)$, given a Gaussian prior $\lambda(\theta) = \mathcal{N}(\theta; \boldsymbol{\mu}_\lambda, \Sigma_\lambda)$. For a single observation $x^\star$, the true posterior is also a Gaussian obtained using Bayes formula, as the product of two Gaussian distributions:

$$p(\theta \mid x^\star) = \mathcal{N}(\theta; \boldsymbol{\mu}_{\mathrm{post}}(x^\star), \Sigma_{\mathrm{post}}) \tag{35}$$

with $\boldsymbol{\mu}_{\mathrm{post}}(x^\star) = \Sigma_{\mathrm{post}}(\Sigma^{-1} x^\star + \Sigma_\lambda^{-1} \boldsymbol{\mu}_\lambda)$ and $\Sigma_{\mathrm{post}} = (\Sigma^{-1} + \Sigma_\lambda^{-1})^{-1}$. Note that in this case, the full posterior can be written as

$$p(\theta \mid x_{1:n}^\star) = \mathcal{N}(\theta; \boldsymbol{\mu}_{\mathrm{post}}(x_{1:n}^\star), \Sigma_{\mathrm{post},n}) \tag{36}$$

with $\Sigma_{\mathrm{post},n} = (n\Sigma^{-1} + \Sigma_\lambda^{-1})^{-1}$ and $\boldsymbol{\mu}_{\mathrm{post}}(x_{1:n}^\star) = \Sigma_{\mathrm{post},n}(\sum_{j=1}^n \Sigma^{-1} x_j^\star + \Sigma_\lambda^{-1} \boldsymbol{\mu}_\lambda)$.

Assume that $q_{t|0}(\theta_t \mid \theta_0) = \mathcal{N}(\theta; \sqrt{\alpha_t} \theta_0, v_t \mathbf{I}_m)$; see Section 2. Using standard results (e.g. Equation 2.115 in Bishop, 2006), we can derive the analytic formula of the diffused prior

$$\begin{aligned} p_t^\lambda(\theta) &= \int \lambda(\theta_0) q_{t|0}(\theta_t \mid \theta_0) \mathrm{d}\theta_0 \\ &= \mathcal{N}(\theta_t; \sqrt{\alpha_t} \boldsymbol{\mu}_\lambda, \alpha_t \Sigma_\lambda + v_t \mathbf{I}_m) \end{aligned} \tag{37}$$

and of the diffused posterior

$$\begin{aligned} p_t(\theta_t \mid x^\star) &= \int p(\theta_0 \mid x^\star) q_{t|0}(\theta_t \mid \theta_0) \mathrm{d}\theta_0 \\ &= \mathcal{N}(\theta_t; \sqrt{\alpha_t} \boldsymbol{\mu}_{\mathrm{post}}(x^\star), \alpha_t \Sigma_{\mathrm{post}} + v_t \mathbf{I}_m). \end{aligned} \tag{38}$$

The corresponding Fisher scores are

$$\begin{aligned} \nabla_{\theta_t} \log p_t^\lambda(\theta_t) &= -(\alpha_t \Sigma_\lambda + v_t \mathbf{I}_m)^{-1}(\theta_t - \sqrt{\alpha_t} \boldsymbol{\mu}_\lambda), \tag{39} \\ \nabla_{\theta_t} \log p_t(\theta_t \mid x^\star) &= -(\alpha_t \Sigma_{\mathrm{post}} + v_t \mathbf{I}_m)^{-1}(\theta_t - \sqrt{\alpha_t} \boldsymbol{\mu}_{\mathrm{post}}(x^\star)). \tag{40} \end{aligned}$$

Replacing the score from (40) in the formulas from (Boys et al., 2023), we get the following expressions for the mean and covariance matrix of the backward kernel $q_{0|t}(\theta_0 \mid \theta_t, x^\star)$:

$$\begin{aligned} \Sigma_t(\theta_t, x^\star) &= \frac{v_t}{\alpha_t} \left(1 - v_t \left(\alpha_t \Sigma_{\mathrm{post}} + v_t \mathbf{I}_m\right)^{-1}\right) = \Sigma_t, \quad \text{(constant)} \\ \boldsymbol{\mu}_t(\theta_t, x^\star) &= \frac{1}{\sqrt{\alpha_t}} \left(1 - v_t \left(\alpha_t \Sigma_{\mathrm{post}} + v_t \mathbf{I}_m\right)^{-1}\right) \theta_t + v_t \left(\alpha_t \Sigma_{\mathrm{post}} + v_t \mathbf{I}_m\right)^{-1} \boldsymbol{\mu}_{\mathrm{post}}(x^\star) \\ &= \frac{\sqrt{\alpha_t}}{v_t} \Sigma_t \theta_t + v_t \left(\alpha_t \Sigma_{\mathrm{post}} + v_t \mathbf{I}_m\right)^{-1} \boldsymbol{\mu}_{\mathrm{post}}(x^\star). \end{aligned}$$

The same goes for the prior backward diffusion kernel $q_{0|t}^\lambda(\theta_0 \mid \theta)$:

$$\begin{aligned} \Sigma_{t,\lambda}(\theta_t) &= \frac{v_t}{\alpha_t} \left(1 - v_t \left(\alpha_t \Sigma_\lambda + v_t \mathbf{I}_m\right)^{-1}\right) = \Sigma_{t,\lambda}, \quad \text{(constant)} \\ \boldsymbol{\mu}_{t,\lambda}(\theta_t) &= \frac{1}{\sqrt{\alpha_t}} \left(1 - v_t \left(\alpha_t \Sigma_\lambda + v_t \mathbf{I}_m\right)^{-1}\right) \theta_t + v_t \left(\alpha_t \Sigma_\lambda + v_t \mathbf{I}_m\right)^{-1} \boldsymbol{\mu}_\lambda \\ &= \frac{\sqrt{\alpha_t}}{v_t} \Sigma_{t,\lambda} \theta_t + v_t \left(\alpha_t \Sigma_\lambda + v_t \mathbf{I}_m\right)^{-1} \boldsymbol{\mu}_\lambda. \end{aligned}$$

### D.2  Mixture of Gaussians case

In the case of the Mixture of Gaussians, the prior is $\lambda = \mathcal{N}(0, \mathbf{I}_m)$, the simulator is given by $p(x|\theta) = \frac{1}{2}\mathcal{N}(x; \theta, \Sigma_1) + \frac{1}{2}\mathcal{N}(x; \theta, 1/9\Sigma_2)$ where $\Sigma_1 = 2.25\Sigma$, $\Sigma_2 = 1/9\Sigma$ and $\Sigma$ is a diagonal matrix with values increasing linearly between 0.6 and 1.4. In this case, the posterior density writes

$$p(\theta|x) = \omega_1(x)\mathcal{N}(\theta; \boldsymbol{\mu}_1, \Sigma_{1,p}) + \omega_2(x)\mathcal{N}(\theta; \boldsymbol{\mu}_2, \Sigma_{2,p}) \tag{41}$$

where for $i = 1, 2$, $\Sigma_{i,p} = \left(\Sigma_i^{-1} + \mathbf{I}_m\right)^{-1}$, $\mu_i = \Sigma_{i,p}\Sigma_i^{-1}x$, $\tilde{\omega}_i(x) = \mathcal{N}(x; \theta, \Sigma_i + \mathbf{I}_m)$ and $\omega_i = \tilde{\omega}_i/\sum_{j=1}^{2}\tilde{\omega}_j$.

Therefore, the diffused marginals are

$$
\begin{aligned}
p_t(\theta_t|x) = \ &\omega_1(x)\mathcal{N}(\theta_t; \alpha_t^{1/2}\mu_1, \alpha_t\Sigma_{1,p} + (1-\alpha_t)\mathbf{I}_m) \\
&+ \omega_2(x)\mathcal{N}(\theta_t; \alpha_t^{1/2}\mu_2, \alpha_t\Sigma_{2,p} + (1-\alpha_t)\mathbf{I}_m),
\end{aligned}
\tag{42}
$$

from which the score is

$$
\begin{aligned}
\nabla_{\theta_t} \log p_t(\theta_t|x) = \ &-\omega_1(x)(\alpha_t\Sigma_{1,p} + (1-\alpha_t)\mathbf{I}_m)^{-1}(\theta_t - \alpha_t^{1/2}\mu_1) \\
&- \omega_2(x)(\alpha_t\Sigma_{2,p} + (1-\alpha_t)\mathbf{I}_m)^{-1}(\theta_t - \alpha_t^{1/2}\mu_2).
\end{aligned}
\tag{43}
$$

### D.3  Score of the diffused Log-Normal prior

The Log-Normal distribution can easily be transformed into a Gaussian distribution:

$$\Theta \sim \text{LogNormal}(m, s) \Rightarrow \log\Theta \sim \mathcal{N}(m, s).$$

We can therefore directly apply the Gaussian case from the previous section.

### D.4  Score of the diffused Uniform prior

Consider the case where the prior is a Uniform distribution $\lambda(\theta) = \mathcal{U}(\theta; a, b)$, with $(a, b) \in \mathbb{R}^m \times \mathbb{R}^m$, the lower and upper bounds respectively. Assume that $q_{t|0}(\theta_t \mid \theta_0) = \mathcal{N}(\theta_t; \sqrt{\alpha_t}\theta_0, v_t\mathbf{I}_m)$; see Section 2 with $v_t = 1 - \alpha_t$. We get the following analytic formula for the diffused prior:

$$
\begin{aligned}
p_t^\lambda(\theta_t) \ &= \ \int \lambda(\theta_0)q_{t|0}(\theta_t \mid \theta_0)\mathrm{d}\theta_0 &\tag{44} \\
&= \ \frac{1}{\prod_{i=1}^{m}(b_i - a_i)} \int_{[a_1,b_1]\times\cdots\times[a_m,b_m]} \mathcal{N}\left(\theta_t; \sqrt{\alpha_t}\theta_0, v_t\mathbf{I}_m\right) &\tag{45} \\
&= \ \frac{1}{\sqrt{\alpha_t}\prod_{i=1}^{m}(b_i - a_i)} \prod_{i=1}^{m}\left(\Phi\left(\sqrt{\alpha_t}b_i; \theta_{t,i}, v_t\right) - \Phi\left(\sqrt{\alpha_t}a_i; \theta_{t,i}, v_t\right)\right), &\tag{46}
\end{aligned}
$$

where $\theta_{t,i}$ the $i$th coordinate of $\theta_t$ and $\Phi(.; \mu, \sigma^2)$ is the c.d.f. of a univariate Gaussian with mean $\mu$ and variance $\sigma^2$. The score of the above quantity is then simply obtained by computing the score of each one-dimensional element in the above product. For $i \in [1, m]$, the $i$th coordinate of $\nabla_{\theta_t} \log p_t^\lambda(\theta_t)$ can be written as $\nabla_{\theta_{t,i}} \log f(\theta_{t,i}) = \frac{\nabla_{\theta_{t,i}} f(\theta_{t,i})}{f(\theta_{t,i})}$ with

$$
\begin{aligned}
f(\theta_{t,i}) \ &= \ \left(\Phi\left(\sqrt{\alpha_t}b_i; \theta_{t,i}, v_t\right) - \Phi\left(\sqrt{\alpha_t}a_i; \theta_{t,i}, v_t\right)\right) &\tag{47} \\
\nabla_{\theta_{t,i}} f(\theta_{t,i}) \ &= \ -\frac{1}{\sqrt{\alpha_t}v_t}\left(\mathcal{N}\left(\sqrt{\alpha_t}b_i; \theta_{t,i}, v_t\right) - \mathcal{N}\left(\sqrt{\alpha_t}a_i; \theta_{t,i}, v_t\right)\right). &\tag{48}
\end{aligned}
$$

# E   Experimental Setup

**Code and compute resources.** All experiments are implemented with Python (Python Software Foundation, 2017) combined with PyTorch (Paszke et al., 2019). The code to reproduce all numerical experiments is provided in the following repository: `https://github.com/JuliaLinhart/diffusions-for-sbi`. This includes specific information about the code environment and installation requirements, mentioned in the "readme.md" file. In this url, we also provide pregenerated data and precomputed results that are necessary to quickly generate all figures of the paper without extensive computation time.

**Methods.** We choose F-NPSE (Geffner et al., 2023) as a baseline for comparisons, which uses unadjusted Langevin dynamics with $L = 5$ Langevin steps and a step size of $\delta_t = \tau(1-\alpha_t)/\sqrt{\alpha_t}$ with $\tau = 0.5$, to approximately sample from the tall posterior. We do not compare to other SBI methods based on NPE/NLE/NRE, as the score-based framework has shown to demonstrate clear superior performance (Sharrock et al., 2022; Geffner et al., 2023; Gloeckler et al., 2024). Of course, this choice reduces the range of comparisons of our experimental section, but allows us to focus solely on the best alternative method from current literature.

For `GAUSS` and `JAC`, we infer the tall posterior by plugging the approximate tall posterior scores obtained via Algorithms 1 and 2 into the DDIM sampler defined in Section 2.1. Specifically, we sample from the backward Markov chain from Equation (34), with $\sigma_t^2 = \eta^2(1-\alpha_{t-1})/\upsilon_t(1-\alpha_t/\alpha_{t-1})$ where $\eta = 0.2, 0.5, 0.8, 1$ for a number of steps $T = 50, 150, 400, 1000$ respectively. We use a uniform scheduling $\{t_i = i/T\}_{i=1}^T$.

**The score network.** Except for the two toy models considered in Section 4.1, the posterior scores have to be learned via NPSE. The same score model architecture and training hyper parameters are used for all tasks. Our implementation of the score model $s_\phi(\theta, x, t)$ is an MLP with layer normalization and 3 hidden layers of 256 hidden features. It takes as input the variables $(\theta, x, t)$ and outputs a vector of the same size as $\theta \in \mathbb{R}^m$. Before being passed to the score network, a positional embedding is computed for $t \in [0, 1]$ as per: $(\cos(ti\pi), \sin(ti\pi))_{1 \le i \le f}$, where we chose $f = 3$ for the number of frequencies. Optionally, $\theta$ and $x$ can each be passed to an embedding network. Note that for the benchmark experiment in Section 4.2, no embedding networks were used, as we chose to use the same score model architecture across all tasks, each with different data dimensions. For the more complex neuroscience application in Section 4.3, $\theta$ and $x$ were each embedded with a MLP with 3 hidden layers of 64 hidden features and an output dimension of 32.

For more stability, following Song et al. (2021a) and Ho et al. (2020), we actually learn the *noise distribution* $\epsilon_\phi$, which is related to the score function via $\epsilon_\phi(\theta, x, t) = -\sqrt{\upsilon_t} \, s_\phi(\theta, x, t)$, with $\upsilon_t = 1 - \alpha_t$ the variance of the VP forward kernel, defined in Section 2.1. Given a training dataset with $N = N_{\text{train}}$ samples $(\Theta_{0,i}, X_i) \sim p(\theta, x)$, the empirical loss function for the "noise predictor" NPSE writes:

$$\mathcal{L}_{\text{NPSE}}^N(\phi) = \frac{1}{N} \sum_{i=1}^N \|\epsilon_\phi(\Theta_{t,i}, X_i, t_i) - Z_i\|^2, \quad \text{with} \quad \Theta_{t,i} = \sqrt{\alpha_{t_i}}\Theta_{0,i} + \upsilon_{t_i} \, Z_i, \tag{49}$$

where $Z_i \sim \mathcal{N}(0, \mathbf{I}_m)$ and $t_i \sim \mathcal{U}(0, 1)$. Training is done using the Adam optimizer over $5\,000$ epochs and early stopping on $20\%$ of the training data. Note that early stopping requires to be done with care, since the loss function is stochastic. For each example and training set $N_{\text{train}}$, we trained several models for different learning rates and batch sizes, and selected the one that corresponds to the best trade-off between smallest validation loss and latest stopping epoch, as detailed in Appendix F. The training dataset is always normalized to zero mean and standard variance (for variables $\theta$ and $x$). The same transformation has to be applied to the observations $x_{1:n}^\star$ and the prior score function $\nabla_\theta \log p_t^\lambda(\theta)$ during sampling. After sampling, the inverse transformation is applied to the obtained samples of the approximate tall posterior distribution.

**Validation.** To evaluate the accuracy of the sampling algorithms, we compute the distance between $10^4$ samples of the estimated and true posteriors considering three different metrics: the sliced Wasserstein (sW) distance from the `POT` python package with $10^3$ projections, the Maximum Mean Discrepancy (MMD) and the Classifier-Two-Sample Test (C2ST), both from the `sbibm` package, with default parameters.

The true posterior samples are obtained via MCMC using `numpyro` with `jax`, except for the Gaussian Mixture Toy Model from Section 4.1, for which we consider the Metropolis-Adjusted Langevin Algorithm (MALA); see (Roberts & Tweedie, 1996).

For the JRNMM in Section 4.3, no samples from the true posterior are available and the above metrics cannot be computed. We therefore use the *local* Classifier-Two-Sample Test ($\ell$-C2ST) (Linhart et al., 2023) and the implementation provided in the `sbi` toolbox, with default parameters.

## F   Loss functions, training strategy and model selection

In this section we explain our model selection strategy, namely the choices of learning rate and batch size used for the training of the score models in the experiments from Sections 4.2 and 4.3. As detailed in Appendix E, the model architecture (i.e. number of MLP layers, hidden features, etc.) is fixed and training is done using the Adam optimizer over $5\,000$ epochs and early stopping using 20% of the training data to compute the validation loss. Note that early stopping requires to be done with care, since the loss function is stochastic. For each example, we therefore trained several models for different learning rates and/or batch sizes, and selected the one that corresponds to the best trade-off between smallest validation loss and latest stopping epoch.

Figure 6 displays the train and validation losses for all SBI benchmarks examples and training set sizes $N_{\text{train}}$. For small $N_{\text{train}}$, over fitting behavior can be detected, which is particularly visible for the SLCP example. In these cases, a higher batch size of 256 is chosen, since it yields more stable loss functions, with delayed over fitting and thus also delayed early stopping. The learning rate is then chosen to correspond to the smallest validation loss when the gap is clear (e.g. for SLCP). In cases were no obvious overfitting is detected, the best validation loss in each setting is similar and the learning rate can then be chosen to correspond to the latest early stopping epoch (to mitigate variations due to the stochasticity of the loss function and ensure longer training).

The same strategy was then used to select the best model for the experiments on the JRNMM, for which the loss functions are displayed in Figure 7. Here, the score models were trained on a large training set of size $N_{\text{train}} = 50\,000$. We therefore directly chose a larger batch size of 256 for more stability, leading to equal early stopping times in each setting. The chosen models then correspond to the smallest validation loss, here obtained for a learning rate of $1e^{-4}$.

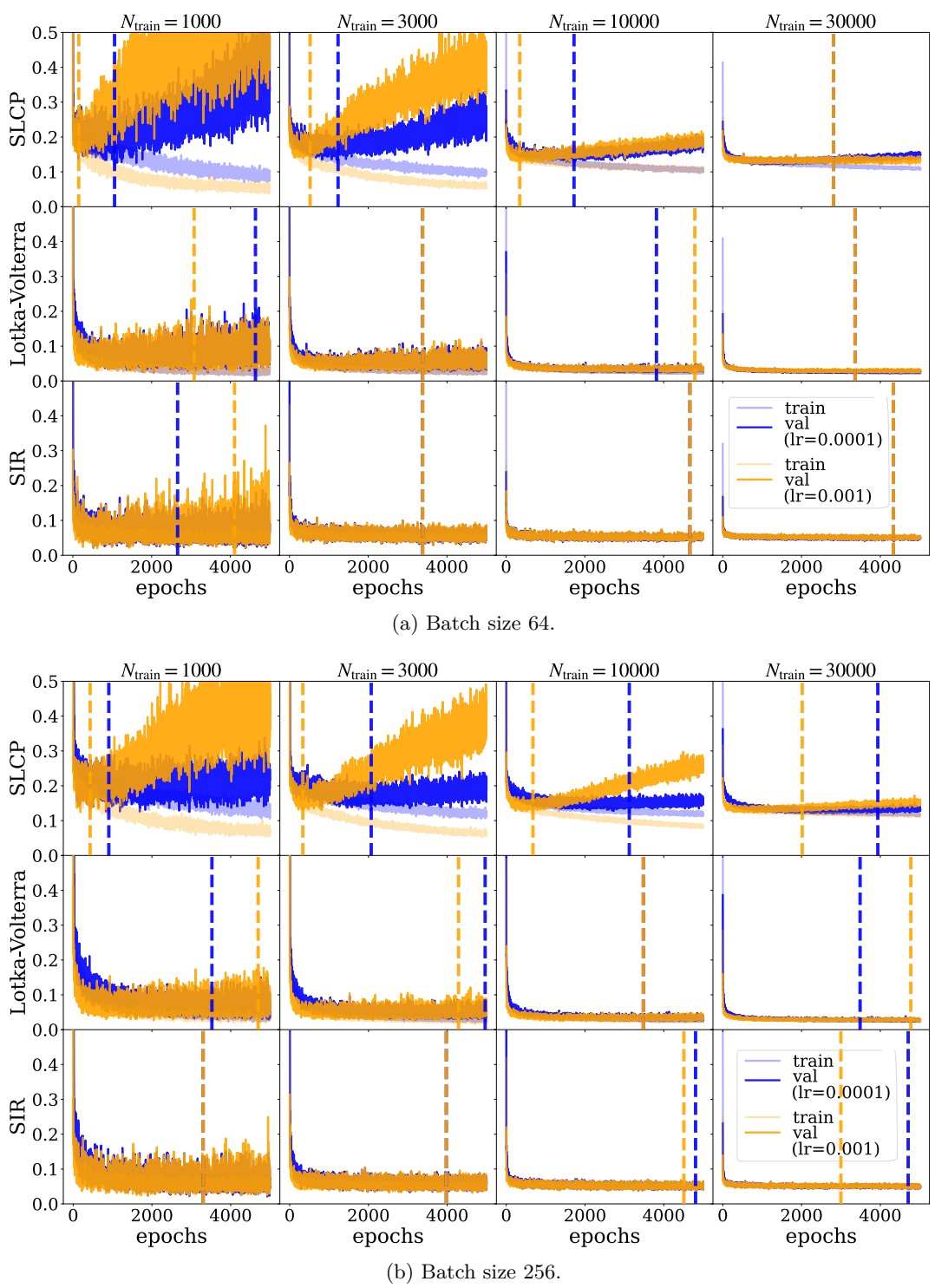

(a) Batch size 64.

(b) Batch size 256.

Figure 6: Train and validation losses for the SBI benchmark examples obtained for score networks trained over $5\,000$ epochs with the Adam optimzer and learning rates $\mathrm{lr} = 1e^{-3}$ (orange) and $\mathrm{lr} = 1e^{-4}$ (blue). The two Figures respectively corresponds to a batch size of 64 and 256. The validation loss was computed on a held-out validation set of 20% of the training dataset of size $N_{\mathrm{train}} \in [10^3, 3.10^3, 10^4, 3.10^4]$. Dashed lines indicate the epoch at early stopping time. A higher batch size of 256 is chosen, since it yields more stable loss functions, with delayed over fitting and thus also delayed early stopping (particularly visible for the SLCP example). The learning rate is then chosen to correspond to the smallest validation loss or, in cases were no obvious overfitting is detected, to the latest early stopping epoch (to mitigate small variations due to the stochasticity of the loss function and ensure longer training).

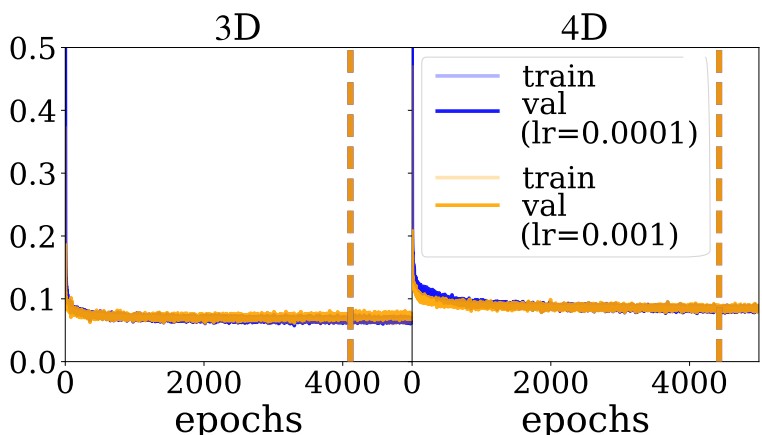

Figure 7: Loss functions for the Jansen and Rit Neural Mass Model obtained for score networks trained over 5 000 epochs with the Adam optimzer, a batch size of 256 and learning rates lr $= 1e^{-3}$ (orange) and lr $= 1e^{-4}$ (blue). The validation loss was computed on a held-out validation set of 20% of the training dataset of size $N_{\mathrm{train}} = 50\ 000$. Dashed lines indicate the epoch at early stopping time. As early stopping in both settings occurs at the same time, the chosen model corresponds to the smallest validation loss, here obtained for a learning rate of $1e^{-4}$.

# G    Additional results for the toy models

Table 4 displays the results obtained for the `GMM` example, which are comparable to the ones from Table 1, used to define the equivalent time setting in Section 4.1.

| Algorithm | N steps | $\Delta t$ (s) | $sW$ |
|-----------|---------|----------------|------|
| GAUSS | 50 | 0.99 +/- 0.00 | 0.30 +/- 0.05 |
| JAC | 50 | 0.89 +/- 0.00 | 82.73 +/- 67.41 |
| Langevin | 50 | 1.77 +/- 0.01 | 0.24 +/- 0.01 |
| GAUSS | 150 | 1.83 +/- 0.01 | 0.31 +/- 0.04 |
| JAC | 150 | 2.66 +/- 0.00 | 5.89 +/- 1.07 |
| Langevin | 150 | 5.28 +/- 0.01 | 0.35 +/- 0.02 |
| GAUSS | 400 | 3.91 +/- 0.01 | 0.31 +/- 0.04 |
| JAC | 400 | 7.13 +/- 0.02 | 3.41 +/- 1.04 |
| Langevin | 400 | 14.06 +/- 0.04 | 0.43 +/- 0.02 |
| GAUSS | 1000 | 8.85 +/- 0.03 | 0.34 +/- 0.03 |
| JAC | 1000 | 17.69 +/- 0.07 | 7.33 +/- 5.41 |
| Langevin | 1000 | 35.08 +/- 0.15 | 0.47 +/- 0.02 |

Table 4: Sliced Wasserstein (sW) distance and total sampling time $\Delta t$ for the `GMM` toy problem with $m = 10$, $n = 32$ and $\epsilon = 10^{-2}$ and number T of sampling steps. Mean and std over 5 different seeds.

To complete the robustness analysis, we include in Figures 8 and 9 results obtained for the `Gaussian` example across all considered dimensions $m \in [2, 4, 8, 10, 16, 32]$, number of observations $n$, and perturbations $\epsilon$. Figure 8 confirms the results of Figure 2 and additionally shows that **JAC** is less robust in higher dimensions than **GAUSS**: for $m > 4$, it yields sW values outside the plot boundaries (and **LANGEVIN** already for $m > 2$). Figure 9 shows the same results but compares the results obtained for the different noise levels $\epsilon$, for each algorithm separately. For one, it highlights the precision of **JAC** in the non-perturbed case ($\epsilon = 0$) and its instability otherwise. On the other hand, it emphasizes the superior robustness of **GAUSS** to noise, especially for medium noise level ($\epsilon = 0.01$) and in high dimensions.

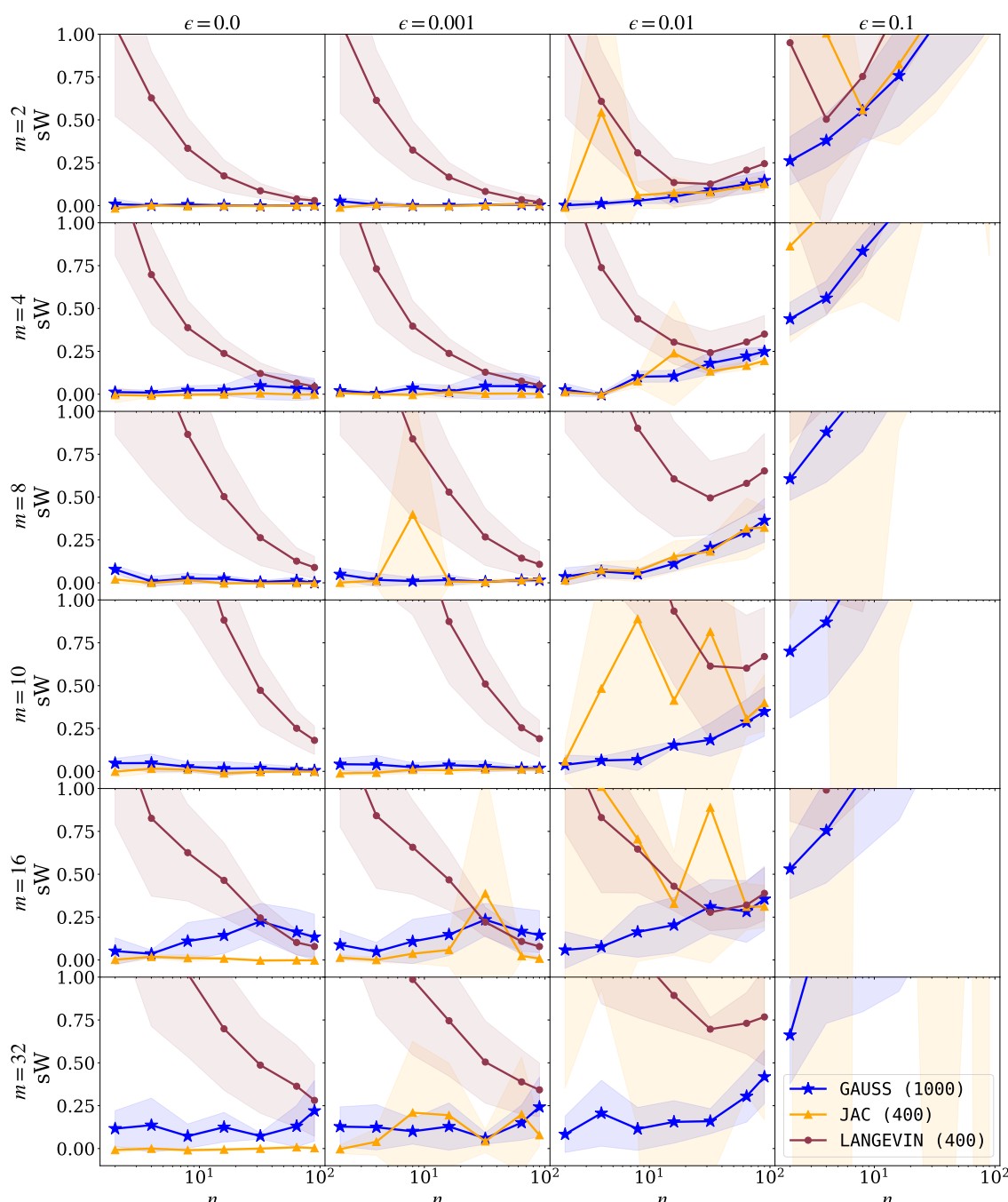

Figure 8: Sliced Wasserstein (sW) distance as a function of the number of observations $n$ for each algorithm (GAUSS, JAC and LANGEVIN) and with different levels of $\epsilon$. Results are shown for the Gaussian example in several dimensions $m \in [2, 4, 8, 10, 16, 32]$. Mean and std over 5 different seeds.

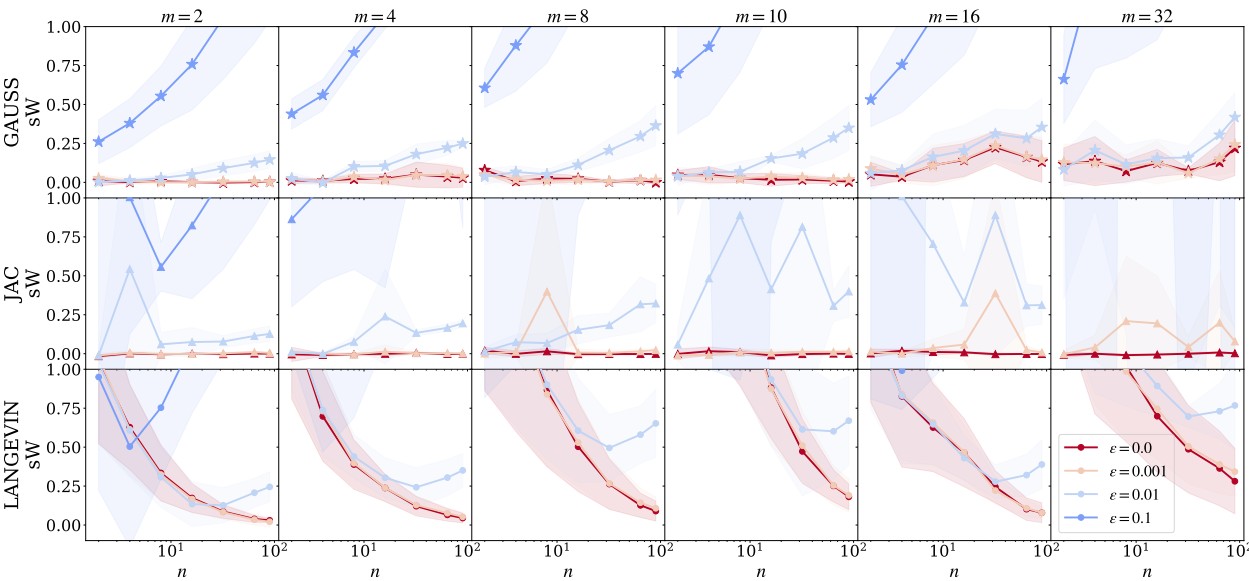

Figure 9: Sliced Wasserstein (sW) distance as a function of the number of observations $n$ for different levels of $\epsilon$, for each algorithm (GAUSS, JAC and LANGEVIN). Results are shown for the Gaussian example in several dimensions $m \in [2, 4, 8, 10, 16, 32]$. Mean and std over 5 different seeds.

# H    Additional results for SBI benchmarks

We here include more detailed results for the three SBI benchmark tasks from Section 4.2. As a complement to the sliced Wassersetin (sW), we report results obtained for the Maximum Mean Discrepancy (MMD) and the Classifier-Two-Sample Test (C2ST) metrics from the `sbibm` package, respectively in Appendix H.2 and H.3. The results for sW are also included in Appendix H.1.

Tables 5, 6 and 7 summarize the results obtained for all metrics, for the three benchmark tasks respectively. In **bold** are displayed the smallest values for the considered metric and number of observations $n$. This highlights that the approximate posterior samples obtained with our methods (`GAUSS` and `JAC`) are closest to the true posterior samples, in particular for $n > 1$. In red we mark the highest value for each metric across all methods and number of i.i.d. observations $n$. This highlights that `LANGEVIN` performs significantly worse, especially as the number of observations $n$ increases ($n = 30$), and this for all considered metrics.

The results in this section highlight an interesting point: increasing distance values (i.e. degradation of performance) for higher $n$, which is particularly well shown for the MMD and C2ST metric in Figures 14 and 16 respectively. This can be explained by the accumulation of approximation errors, as we sum over $n$ evaluations of the score model to obtain the tall posterior score for $n$ observations. We refer the reader to Appendix L.2, which investigates a possible solution to this issue: partially factorized score-based posterior sampling algorithms, such as PF-NPSE proposed by (Geffner et al., 2023).

The tables also point out the differences between the metrics, each providing complementary information about the statistical validity of the approximate posterior. For instance, sW is known to capture well the geometric features of the underlying distribution but struggles in higher dimensions. MMD allows to analyze correlations in higher dimensions, but is known to fail for example in multi-modal settings. C2ST is a measure that captures all kinds of inconsistencies, but this means that it can quickly yield very high values and can be "too discriminative". It also means that C2ST is less informative if the goal is to analyze specific statistical features. The three benchmark examples each have different posterior structures, as shown in Figure 10 which enables to illustrate these differences in a more detailed analysis of the results from Tables 5, 6 and 7.

**SLCP - Table 5.** This is a hard example for which the posterior has 4 modes in a 5-dimensional parameter space. The general difficulty of this task is shown with C2ST, with high scores across all methods and number of observations $n$ (C2ST= 0.7 for $n = 1$ for all methods and increases rapidly for $n > 1$). Here MMD is uninformative (low scores everywhere), while *sW effectively captures the approximation errors due to the multi-modal structure of the posterior.*

**Lotka-Volterra (LV)- Table 6.** The posterior has a single mode and lives in a 4-dimensional parameter space. The difficulty of this task is to be accurate in every dimension and to concentrate around the true parameters as $n$ increases. Again, C2ST is high for all methods and all $n$. *MMD completes the analysis by effectively capturing the accumulation of errors as $n$ increases.* This time *sW is uninformative* (small scores everywhere) as there is no specific geometric structure like multi-modality in the SLCP example.

**SIR - Table 7.** The posterior has only one mode and lives in a 2-dimensional parameter space. This task is simpler as shown by the C2ST scores that start low ( 0.6 for $n = 1$). Like in LV, *MMD reflects the accumulation error as $n$ increases and sW is uninformative given the lack of difficult geometry in the posterior distribution.*

| | sW | | | MMD | | | C2ST | | |
|---|---|---|---|---|---|---|---|---|---|
| $n$ | 1 | 14 | 30 | 1 | 14 | 30 | 1 | 14 | 30 |
| **GAUSS** | 0.31 +/- 0.10 | **0.37** +/- **0.11** | **0.40** +/- **0.13** | 0.02 +/- 0.02 | **0.02** +/- **0.02** | **0.02** +/- **0.01** | 0.70 +/- 0.08 | 0.92 +/- 0.06 | 0.95 +/- 0.05 |
| **JAC-clip** | 0.31 +/- 0.09 | 0.47 +/- 0.08 | 0.51 +/- 0.10 | 0.02 +/- 0.02 | **0.02** +/- **0.01** | **0.02** +/- **0.01** | 0.70 +/- 0.07 | **0.89** +/- **0.07** | **0.94** +/- **0.06** |
| **LANGEVIN-clip** | 0.31 +/- 0.11 | 0.56 +/- 0.17 | 0.81 +/- 0.26 | 0.02 +/- 0.02 | 0.05 +/- 0.05 | 0.10 +/- 0.07 | 0.70 +/- 0.08 | 0.92 +/- 0.06 | 0.97 +/- 0.03 |

Table 5: **Results for SLCP** for $N_{\mathrm{train}} = 3.10^4$. Mean and std over 25 parameters $\theta^\star \sim \lambda(\theta)$.

| | sW | | | MMD | | | C2ST | | |
|---|---|---|---|---|---|---|---|---|---|
| $n$ | 1 | 14 | 30 | 1 | 14 | 30 | 1 | 14 | 30 |
| `GAUSS` | 0.005 +/- 0.004 | **0.004** +/- **0.005** | **0.005** +/- **0.005** | 0.05 +/- 0.07 | **0.20** +/- **0.23** | **0.31** +/- **0.21** | 0.77 +/- 0.10 | **0.93** +/- **0.04** | **0.96** +/- **0.03** |
| `JAC-clip` | 0.005 +/- 0.004 | 0.01 +/- 0.007 | 0.007 +/- 0.005 | 0.05 +/- 0.07 | 0.26 +/- 0.47 | 0.40 +/- 0.38 | **0.76** +/- **0.10** | 0.94 +/- 0.07 | 0.97 +/- 0.04 |
| `LANGEVIN` | 0.006 +/- 0.005 | 0.03 +/- 0.02 | 0.38 +/- 0.18 | 0.15 +/- 0.10 | 0.61 +/- 0.23 | 0.63 +/- 0.12 | 0.85 +/- 0.07 | 0.99 +/- 0.02 | 1.00 +/- 0.00 |

Table 6: **Results for LV** for $N_{\mathrm{train}} = 3.10^4$. Mean and std over 25 parameters $\theta^\star \sim \lambda(\theta)$.

| | sW | | | MMD | | | C2ST | | |
|---|---|---|---|---|---|---|---|---|---|
| $n$ | 1 | 14 | 30 | 1 | 14 | 30 | 1 | 14 | 30 |
| `GAUSS` | 0.001 +/- 0.001 | **0.002** +/- **0.001** | **0.002** +/- **0.001** | 0.007 +/- 0.007 | 0.23 +/- 0.16 | 0.38 +/- 0.25 | 0.57 +/- 0.03 | 0.78 +/- 0.07 | 0.86 +/- 0.07 |
| `JAC-clip` | 0.001 +/- 0.001 | **0.002** +/- **0.001** | **0.002** +/- **0.001** | 0.006 +/- 0.005 | **0.08** +/- **0.09** | **0.21** +/- **0.09** | 0.58 +/- 0.03 | **0.75** +/- **0.07** | **0.84** +/- **0.06** |
| `LANGEVIN` | 0.001 +/- 0.001 | 0.01 +/- 0.006 | 0.07 +/- 0.02 | 0.018 +/- 0.018 | 0.56 +/- 0.25 | 0.73 +/- 0.03 | 0.63 +/- 0.04 | 0.93 +/- 0.12 | 1.00 +/- 0.00 |

Table 7: **Results for SIR** for $N_{\mathrm{train}} = 3.10^4$. Mean and std over 25 parameters $\theta^\star \sim \lambda(\theta)$.

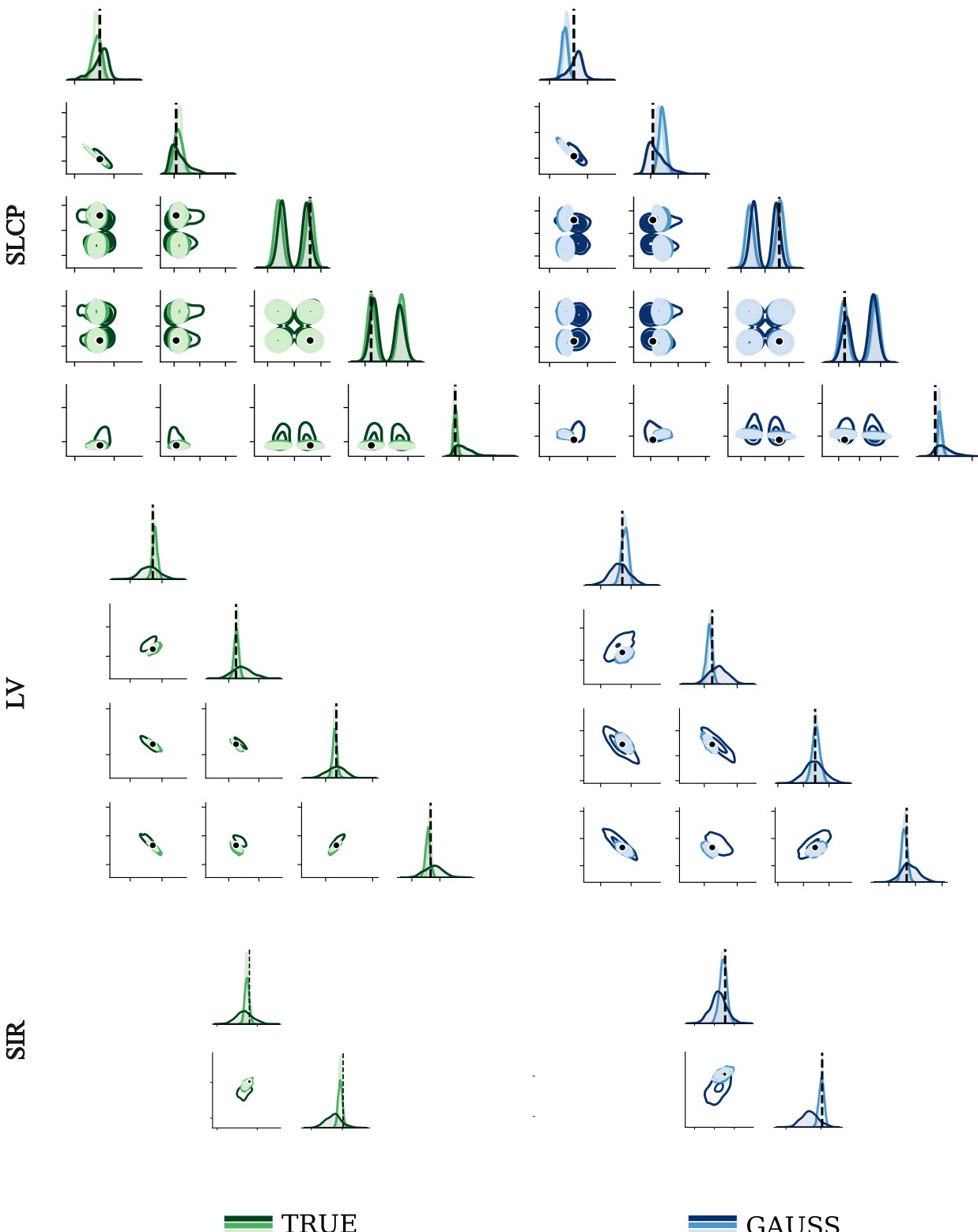

Figure 10: **Pairplots for SLCP, Lotka Volterra (LV) and SIR.** 1D and 2D marginals for the true (green, left) vs. approximate tall posterior samples obtained with GAUSS (blue, right) for a score model trained on $N_{\text{train}} = 3.10^4$ samples and for $n = 1, 14, 30$ (dark to light colors) i.i.d. observations simulated for a given parameter (black dots and dashed lines).

## H.1 Sliced Wasserstein distance

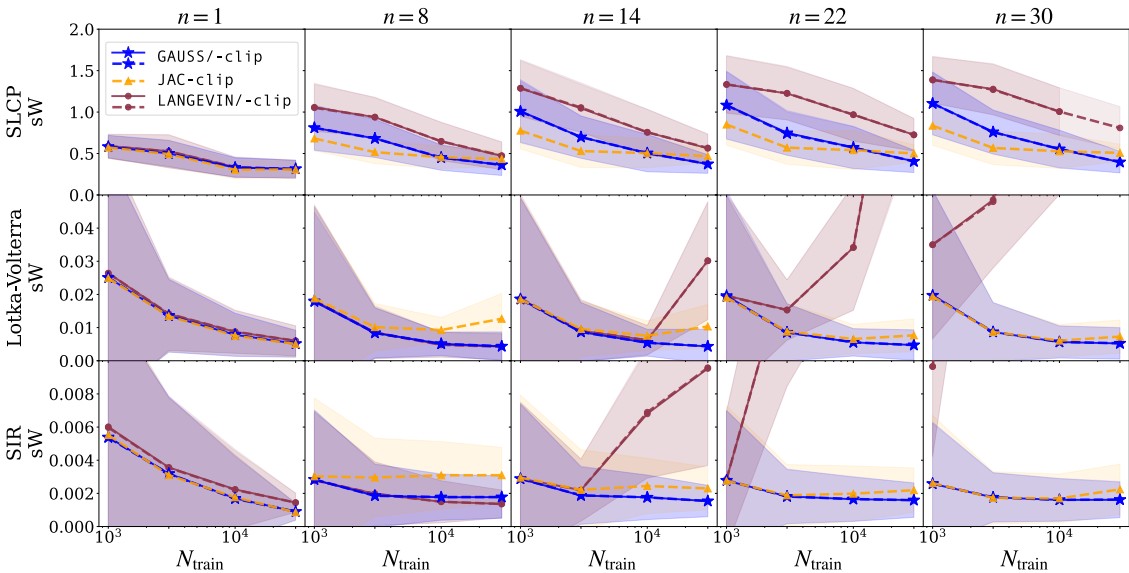

Figure 11: Sliced Wasserstein (sW) as a function of $N_{\text{train}} \in [10^3, 3.10^3, 10^4, 3.10^4]$ between the samples obtained by each algorithm and the true tall posterior distribution $p(\theta \mid x_{1,n}^\star)$ (for $n \in [1, 8, 14, 22, 30]$). Mean and std over 25 different parameters $\theta^\star \sim \lambda(\theta)$.

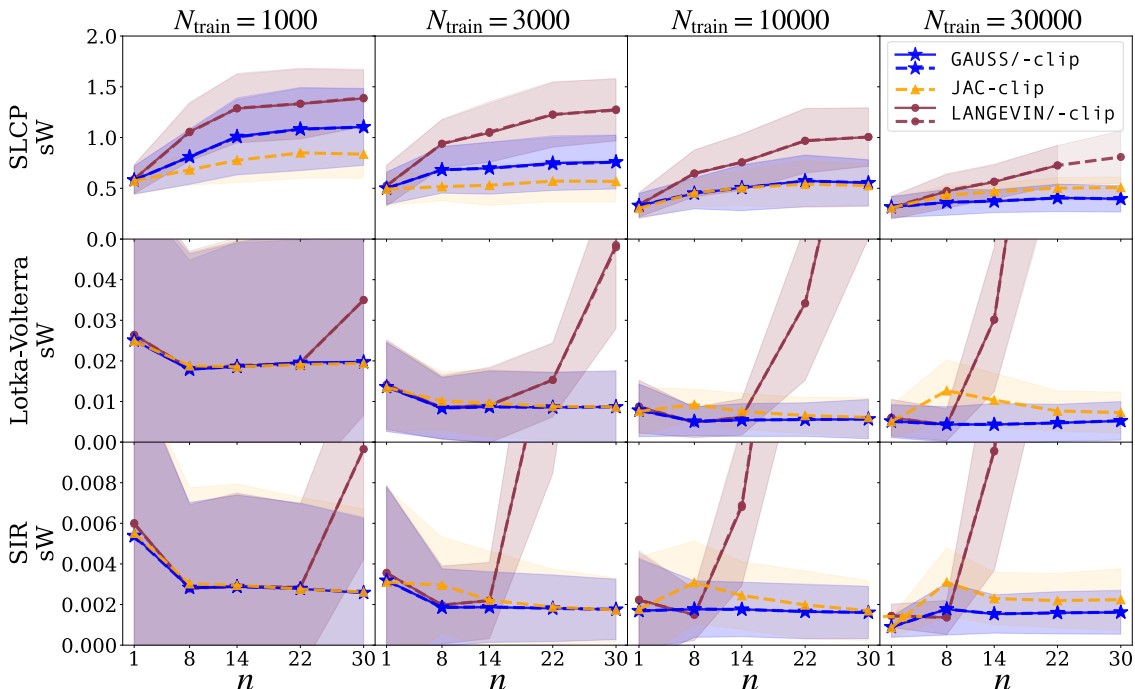

Figure 12: Sliced Wasserstein (sW) a function of $n \in [1, 8, 14, 22, 30]$ between the samples obtained by each algorithm and the true tall posterior distribution $p(\theta \mid x_{1,n}^\star)$ (for $N_{\text{train}} \in [10^3, 3.10^3, 10^4, 3.10^4]$). Mean and std over 25 different parameters $\theta^\star \sim \lambda(\theta)$.

## H.2 Maximum Mean Discrepancy

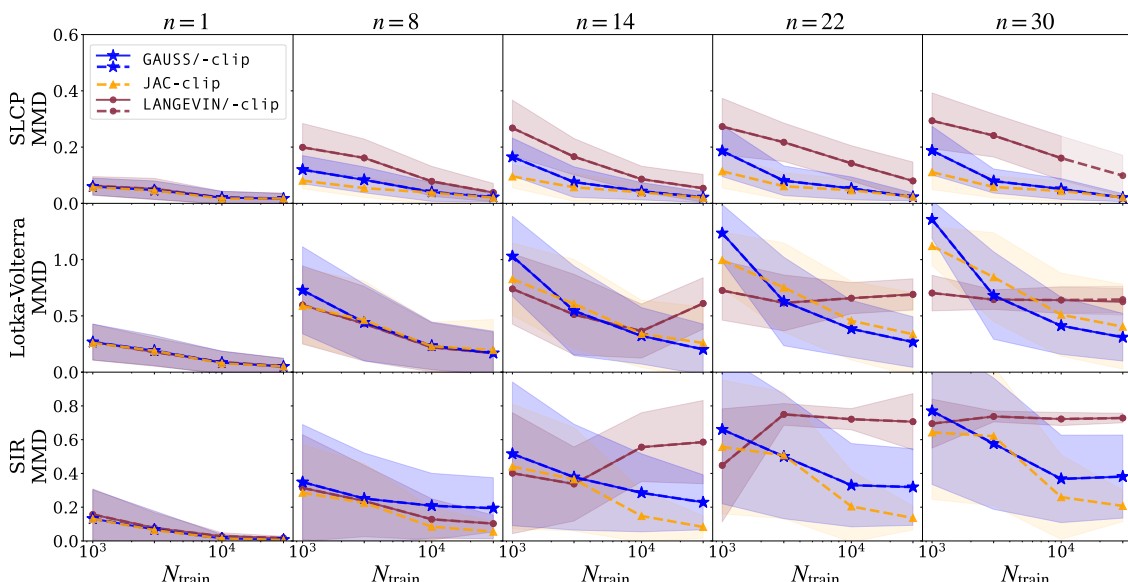

Figure 13: MMD as a function of $N_{\text{train}} \in [10^3, 3.10^3, 10^4, 3.10^4]$ between the samples obtained by each algorithm and the true tall posterior distribution $p(\theta \mid x_{1,n}^\star)$ (for $n \in [1, 8, 14, 22, 30]$). Mean and std over 25 different parameters $\theta^\star \sim \lambda(\theta)$.

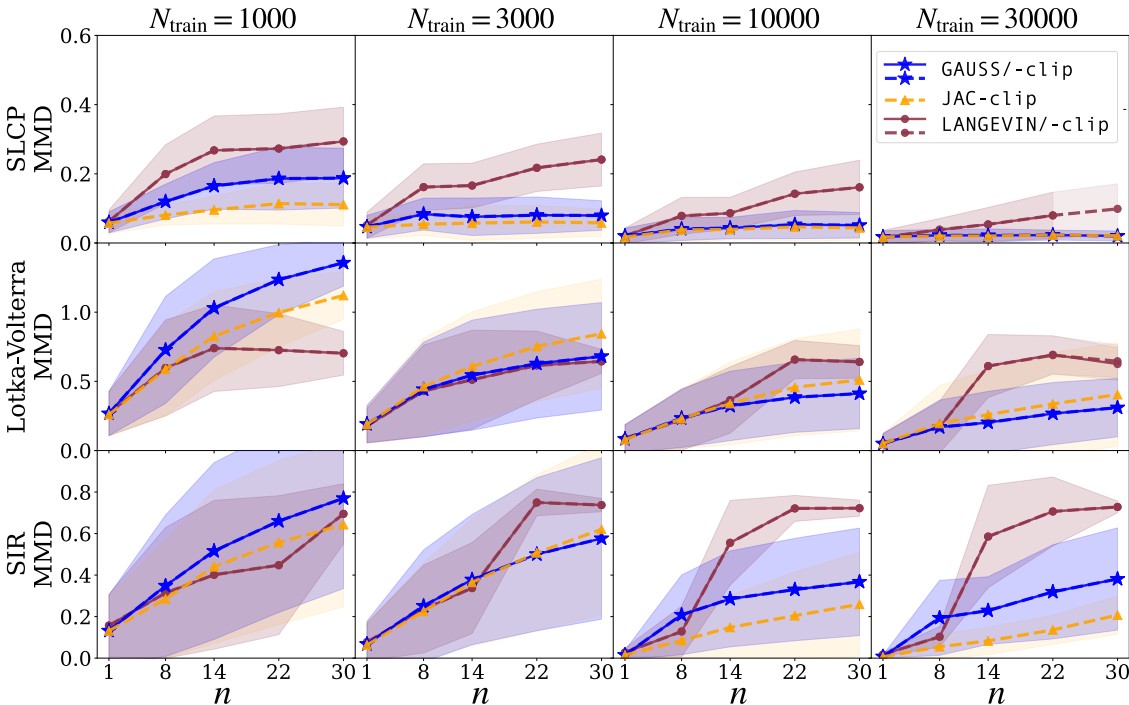

Figure 14: MMD as a function of $n \in [1, 8, 14, 22, 30]$ between the samples obtained by each algorithm and the true tall posterior distribution $p(\theta \mid x_{1,n}^\star)$ (for $N_{\text{train}} \in [10^3, 3.10^3, 10^4, 3.10^4]$). Mean and std over 25 different parameters $\theta^\star \sim \lambda(\theta)$.

## H.3 Classifier-Two-Sample Test

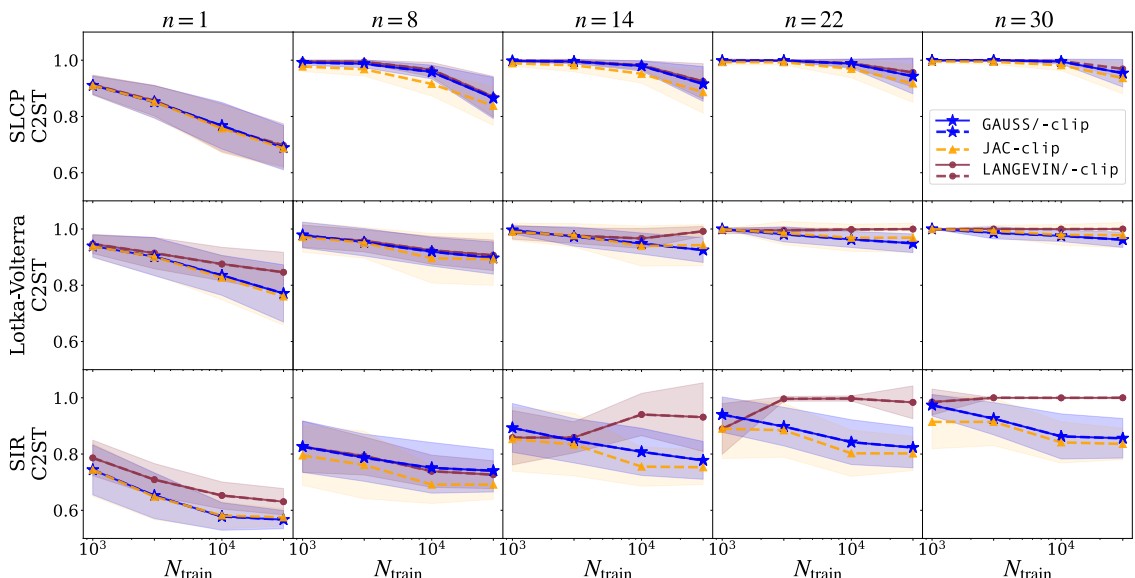

Figure 15: C2ST as a function of $N_{\text{train}} \in [10^3, 3.10^3, 10^4, 3.10^4]$ between between the samples obtained by each algorithm and the true tall posterior distribution $p(\theta \mid x^\star_{1,n})$ (for $n \in [1, 8, 14, 22, 30]$). Mean and std over 25 different parameters $\theta^\star \sim \lambda(\theta)$.

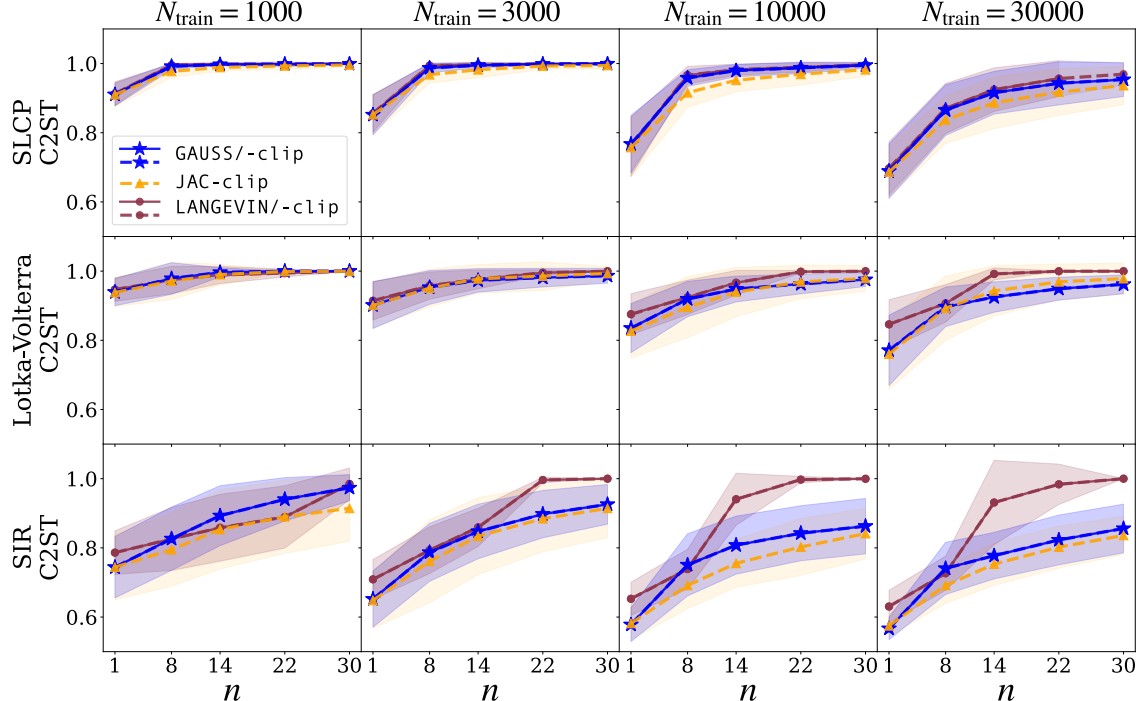

Figure 16: C2ST as a function of $n \in [1, 8, 14, 22, 30]$ between the samples obtained by each algorithm and the true tall posterior distribution $p(\theta \mid x^\star_{1,n})$ (for $N_{\text{train}} \in [10^3, 3.10^3, 10^4, 3.10^4]$) and the Dirac of the true parameters $\theta^\star$ used to simulate the observations $x^\star_{1,n}$. Mean and std over 25 different parameters $\theta^\star \sim \lambda(\theta)$.

# I  Results for additional tasks from the SBI benchmark

To complete our empirical study, we added results for additional examples from the SBI benchmark (Lueck-mann et al., 2021): `Gaussian Linear`, `GMM`, `GMM (uniform)`[8], `B-GLM/ (raw)`[9] and `Two Moons`. These new results allows us to compare the performance of our proposal on other challenging situations, such as when scaling to highly structured (e.g. multimodal) posteriors and high-dimensional observation spaces. Note that these examples go a step further as compared to the experiments carried out by Geffner et al. (2023), including non-Gaussian priors[10]. Figures 17, 18 and 19 respectively report the sW, MMD and C2ST as a function of $N_{\text{train}}$. They all show that our algorithm outperforms the Langevin sampler, with smaller distance values everywhere but for the `GMM (uniform)` example. Interestingly, the certainly very discriminative C2ST metric, shows that all three algorithms fail to infer the tall posterior for the `Two Moons` and `B-GLM/ (raw)` examples.

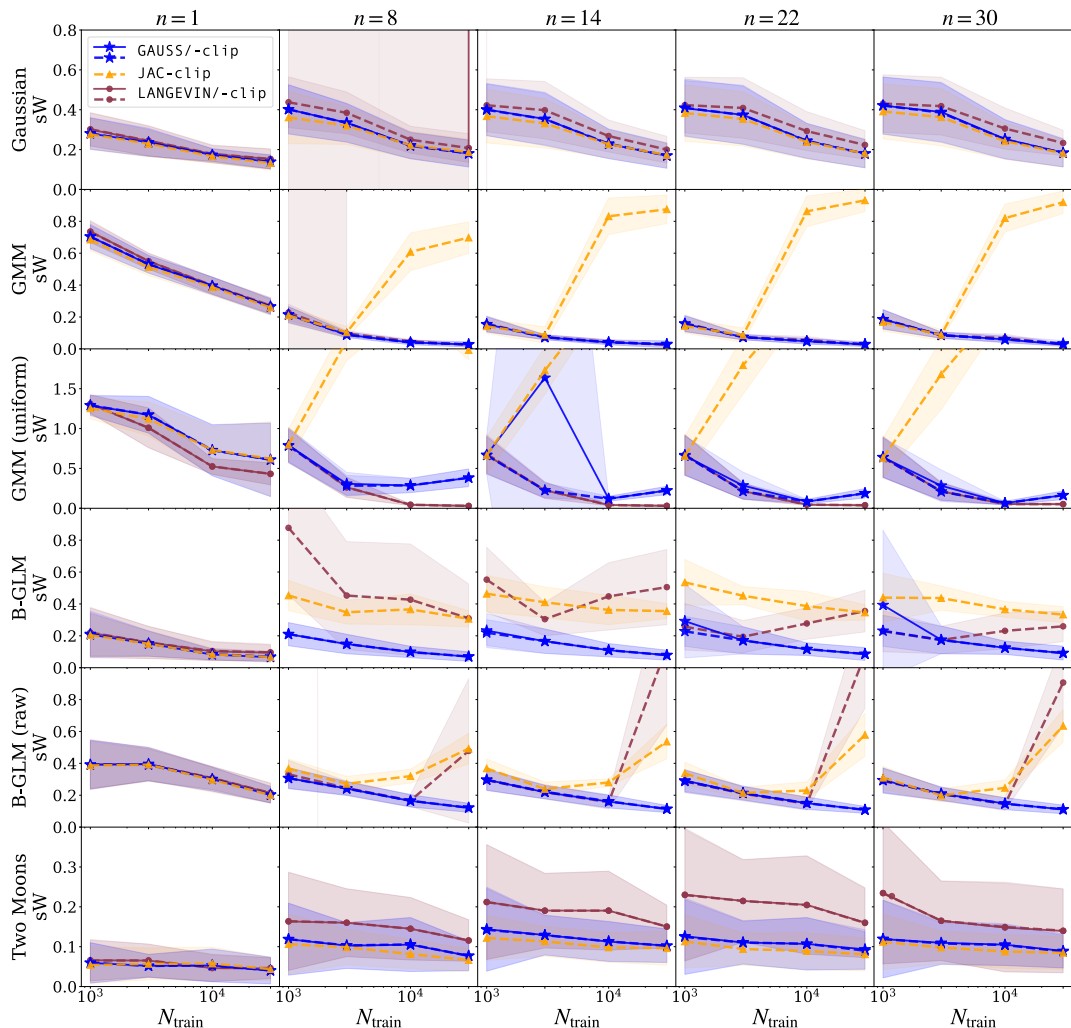

Figure 17: Results for additional benchmark examples. Sliced Wasserstein (sW) distance as a function of $N_{\text{train}} \in [10^3, 3.10^3, 10^4, 3.10^4]$ between between the samples obtained by each algorithm and the true tall posterior distribution $p(\theta \mid x^{\star}_{1,n})$ (for $n \in [1, 8, 14, 22, 30]$). Mean and std over 25 different parameters $\theta^{\star} \sim \lambda(\theta)$.

---

[8]same as `GMM` but with a Uniform prior.
[9]Bernoulli GLM with summary statics / high dimensional raw data.
[10]Note that this was already the case for `SLCP` with Uniform prior.

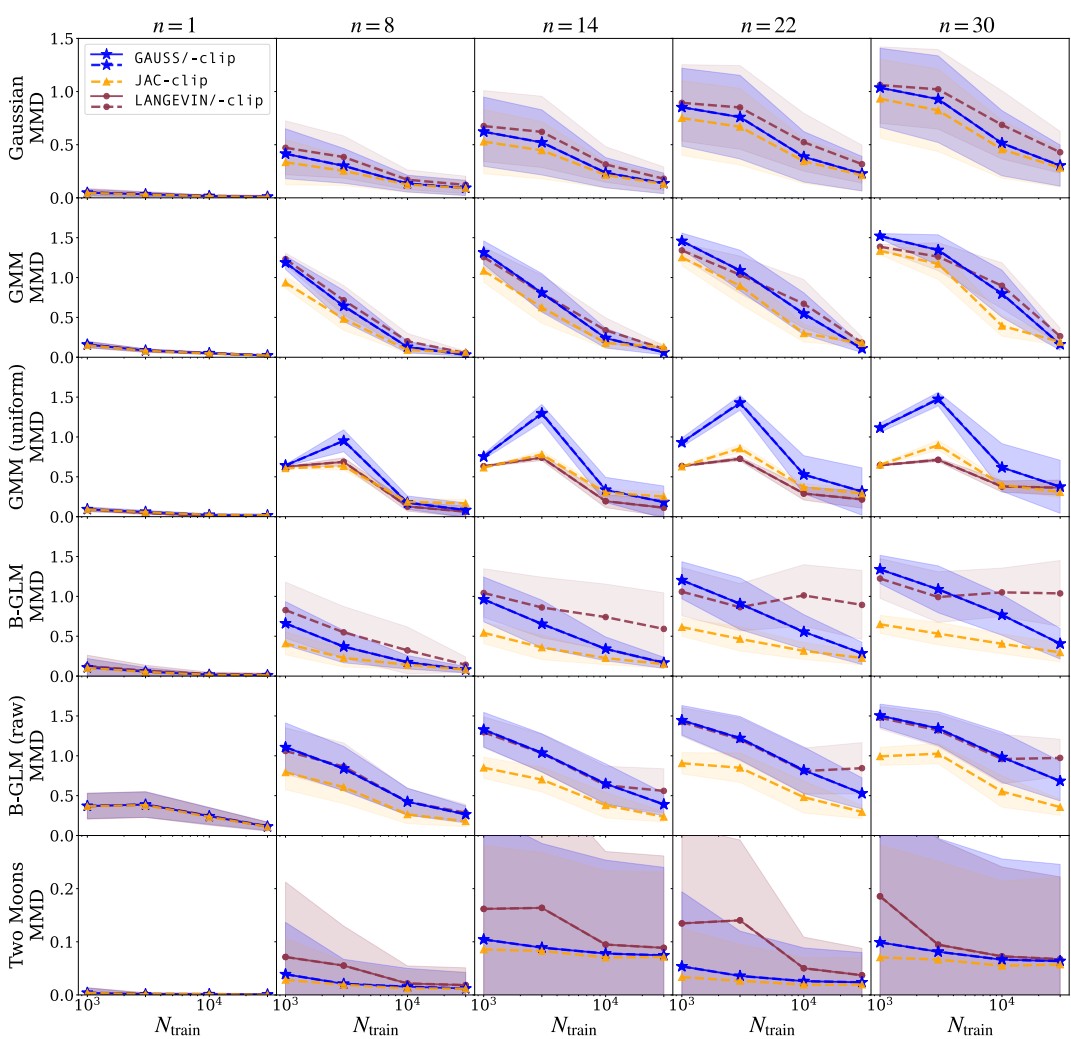

Figure 18: Results for additional benchmark examples. MMD as a function of $N_{\text{train}} \in [10^3, 3.10^3, 10^4, 3.10^4]$ between between the samples obtained by each algorithm and the true tall posterior distribution $p(\theta \mid x^\star_{1,n})$ (for $n \in [1, 8, 14, 22, 30]$). Mean and std over 25 different parameters $\theta^\star \sim \lambda(\theta)$.

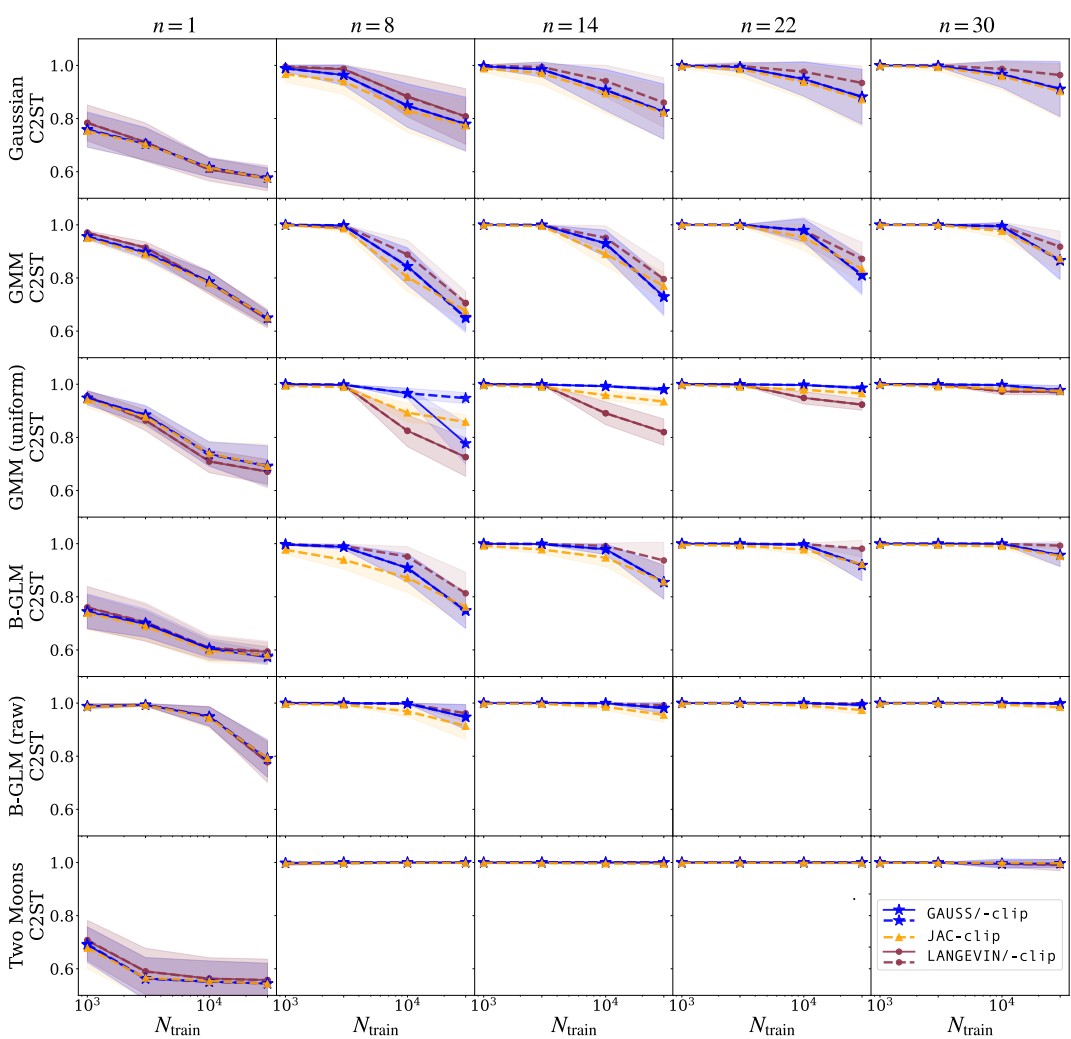

Figure 19: Results for additional benchmark examples. C2ST as a function of $N_{\text{train}} \in [10^3, 3.10^3, 10^4, 3.10^4]$ between between the samples obtained by each algorithm and the true tall posterior distribution $p(\theta \mid x_{1,n}^\star)$ (for $n \in [1, 8, 14, 22, 30]$). Mean and std over 25 different parameters $\theta^\star \sim \lambda(\theta)$.

## J  The Jansen and Rit Neural Mass Model (JRNMM)

### J.1  The JRNMM as a system of stochastic differential equations

The JRNMM (Jansen & Rit, 1995) serves as an illustrative example from computational neuroscience in Section 4.3. Specifically, we consider the implementation by Buckwar et al. (2019) of the stochastic version of the JRNMM presented in (Ableidinger et al., 2017). In our experiments this model is simply presented as a black-box simulator with four input parameters $\theta = (C, \mu, \sigma, g)$ that outputs a signal $x(t) = 10^{g/10}$, which represent the EEG measurement of a brain signal $s(t)$. We will now give insights into the inner workings of this simulator, which are behind the generation of $s(t)$, following the description provided in (Rodrigues et al., 2021). The stochastic JRNMM models the interaction between excitatory and inhibitory neuronal populations in the cortical column, described by a system of three coupled non-linear stochastic differential equations, that can be written as a six-dimensional first-order stochastic differential system:

$$
\begin{aligned}
\dot{X}_0(t) &= X_3(t) \\
\dot{X}_1(t) &= X_4(t) \\
\dot{X}_2(t) &= X_5(t) \\
\dot{X}_3(t) &= \left(Aa\left(\mu_3 + \mathrm{Sigm}\left(X_1(t) - X_2(t)\right) - 2aX_3(t) - a^2 X_0(t)\right) + \sigma_3 \dot{W}_3(t)\right. \\
\dot{X}_4(t) &= \left(Aa\left(\mu_4 + C_2\,\mathrm{Sigm}\left(C_1 X_0(t)\right) - 2aX_4(t) - a^2 X_1(t)\right) + \sigma_4 \dot{W}_4(t)\right. \\
\dot{X}_5(t) &= \left(Bb\left(\mu_5 + C_4\,\mathrm{Sigm}\left(C_3 X_0(t)\right) - 2bX_4(t) - b^2 X_2(t)\right) + \sigma_5 \dot{W}_5(t)\right.
\end{aligned}
\tag{50}
$$

The observed EEG measurement then corresponds to $x(t) = 10^{g/10}(X_1(t) - X_2(t))$, where $g$ is a gain factor expressed in decibels. According to Jansen & Rit (1995), most of the parameters in 50 are expected to be almost constant across different individuals and different experimental conditions, except for $(C_1, C_2, C_3, C_4)$, that represent connectivity, and $\mu_4, \sigma_4$, the statistical parameters of the input signal from neighboring cortical columns. Based on this assumption, Buckwar et al. (2019) propose a simplified implementation of the model defined by a reduced set of four parameters $\theta = (C, \mu, \sigma, g)$ where $\mu = \mu_4$, $\sigma = \sigma_4$ and all connectivity parameters are related via $C_1 = C$, $C_2 = 0.8\,C$, $C_3 = 0.25$, $C_4 = 0.25$. All other parameters in 50 are fixed at their "constant" value.

### J.2  Inference for the JRNMM

This section details some high level choices to perform inference on the JRNMM that is used to illustrate our proposal on a real-world example. They are exactly the same as used in the experiments from the original HNPE paper (Rodrigues et al., 2021).

**Prior distribution.** To avoid any bias due to misspecification issues, inference is often performed with simple and rather uninformative prior. This ensures that the parameter space is sufficiently explored to provide reliable results. In this work, the prior distribution is chosen to be Uniform over the range of scientifically plausible parameter values (Rodrigues et al. (2021), Buckwar et al. (2019), Ableidinger et al. (2017), Jansen & Rit (1995)):

$$
C \sim \mathcal{U}(10, 250), \ \mu \sim \mathcal{U}(50, 500), \ \sigma \sim \mathcal{U}(0, 5000), \ g \sim \mathcal{U}(-20, +20) \ .
\tag{51}
$$

**Summary Statistics.** In traditional SBI methods, it is standard to use summary statistics to improve the inference quality by reducing the dimensionality while describing sufficiently well the statistical features of the data. When using neural density estimators, it is possible to learn these summary statistics with specified embedding networks, such as LSTMS for time series data (Rodrigues & Gramfort, 2020). However, for comparison purposes and enhanced clarity of our contributions, we stick to the summary statistics traditionally used for neural time series data (Buckwar et al., 2019). They consist of the logarithm of the power spectral density (PSD) of each observed time series. The PSD is evaluated in 33 frequency bins between zero and $64\,\mathrm{Hz}$ (half of the sampling rate). This leads to a setting with 4 parameters to estimate given observations defined in a 33-dimensional space.

### J.3 Validation results with $\ell$-C2ST

In this section, we report results on the validity of the inferred posterior for the simplified version (3D) and the full (4D) Jansen and Rit Neural Mass Model (JRNMM).

In both cases, we evaluate the accuracy of each posterior sampling algorithms (**GAUSS**, **JAC** and **LANGEVIN**) w.r.t. the true tall posterior, using the *local* Classifier-Two-sample Test ($\ell$-C2ST) diagnostic proposed by Linhart et al. (2023) and implemented in the `sbi` toolbox. $\ell$-C2ST is a frequentist hypothesis test that evaluates the null hypothesis of sample equality between two conditional distributions by training a classifier on data from the respective joint distributions. The test statistic takes values in $[0, 0.25]$ and is defined by the mean squared error (MSE) between the predicted class probabilities and one half, i.e. the chance level where the classifier is unable to distinguish between the two data classes. Higher MSE values indicate more distributional differences.

Figures 20a and 20b respectively show the $\ell$-C2ST results for the 3D and 4D JRNMM cases, obtained using the `MLPClassifier` from `scikit-learn`, with default parameters from the `sbi` implementation and trained using $10\,000$ samples $(\Theta_i, X_{i,1:n})$ from the joint distributions corresponding to the estimated and true tall data posteriors. We report the mean and std over 5 different seeds used for the initialization of the classifier. On the left, we plot the $\ell$-C2ST statistics, computed on $10\,000$ samples of the approximate posterior obtained by each sampling algorithms for a given observation set $x^\star_{1:n} \sim p(x \mid \theta^\star)$. On the right, we display the associated p-values computed by comparing the obtained statistics to the null hypothesis, estimated over 100 trials. In both cases, the **GAUSS** algorithm consistently yields test statistics close to 0 and p-values above the significance level set at $\alpha = 0.05$, which means that there is no evidence that the estimated posterior is not statistically consistent with the true tall posterior at $x^\star_{1:n}$. This is not the case for **JAC**, whose un-clipped version yields MSE values outside the plot limits and **LANGEVIN**, for which the test is more easily rejected in the 3D case (large std on pvalues that overlap with the significance level) and very clearly rejected in the 4D case for $n > 14$ (p-value below the significance level).

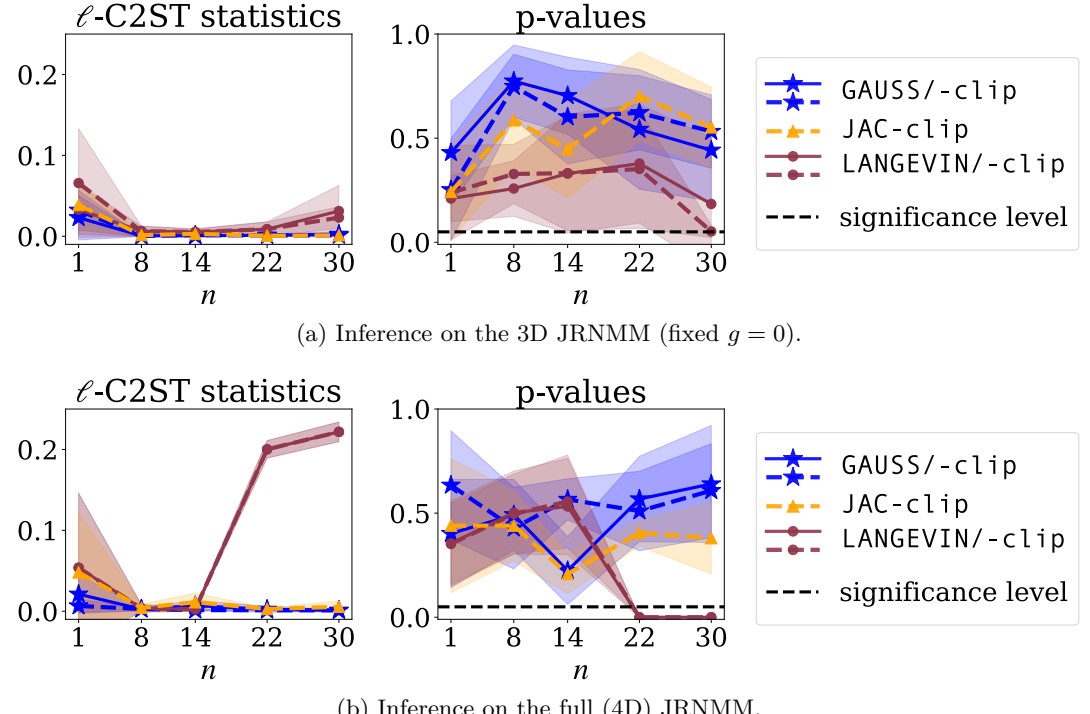

(a) Inference on the 3D JRNMM (fixed $g = 0$).

(b) Inference on the full (4D) JRNMM.

Figure 20: Accuracy of the sampling algorithms w.r.t. the true tall posterior. We show $\ell$-C2ST statistics (left) and corresponding p-values (right) computed for samples obtained with **GAUSS**, **JAC** and **LANGEVIN** at $x^\star_{1:n}$ for $n \in [1, 8, 14, 22, 30]$. Mean and std over 5 seeds.

# K  Limitations of Langevin sampling

**Sensibility to the quality of the score model.** Our results in Section 4.1 analyze the robustness of the different sampling algorithms and essentially show that the Langevin algorithm is very sensible to noisy score networks.

**Step-size choice.** We found that different step sizes can generate very different results for a given score model. Figure 21 shows a comparison between `LANGEVIN` — the unadjusted Langevin algorithm (ULA) from (Geffner et al., 2023) used in all our experiments from Section 4 — and `tamed ULA`, an additional implementation with *tamed* step sizes from (Brosse et al., 2017) as a means to stabilize ULA. We can see that `LANGEVIN` quickly diverges as $n$ increases. A possible explanation could be a learning rate that is too large for settings with big $n$ (i.e. not enough steps are done). We can see that the stabilization tools from the tamed version yield a more stable ULA algorithm, but does not provide a satisfying solution either (lower sW, but still diverges). Fundamentally, there exists a setting where the Langevin algorithm will work (small enough learning rate, run for a long enough time), but this setting is extremely dependent of the problem at hand. This is precisely the strength of our algorithm as compared to ULA: we do not need to sample several times for each marginal $p_t$ at each time step $t$. Note that unfortunately, the code for (Geffner et al., 2023) is not available, so our results are based in a best-effort attempt to reproduce the proposed algorithm.

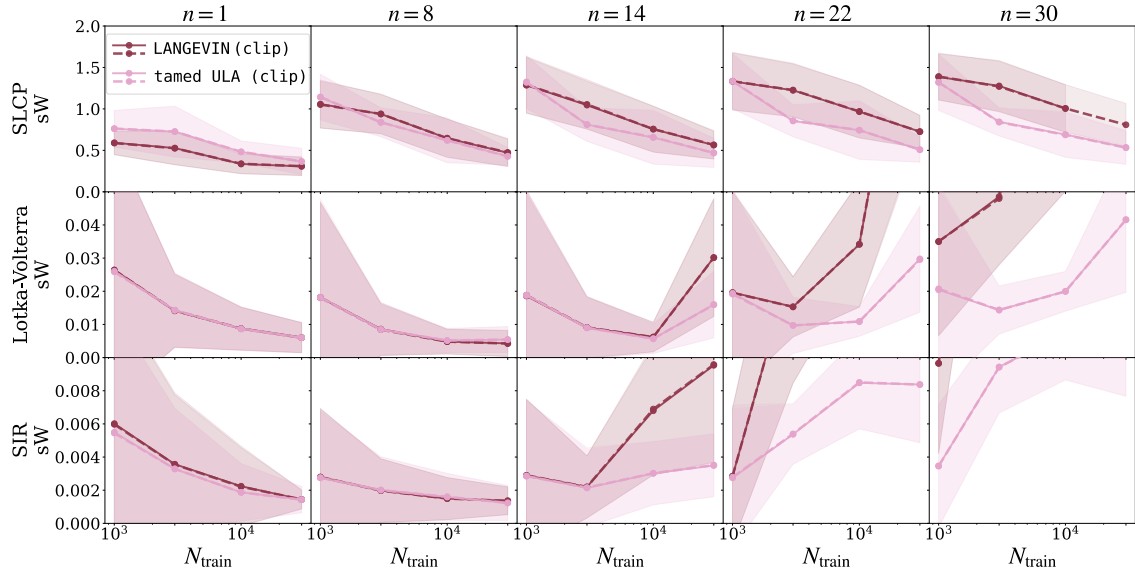

Figure 21: Comparison between the `LANGEVIN` algorithm from (Geffner et al., 2023) (used in all our experiments) and a more stable `tamed ULA` version with *tamed* step size from (Brosse et al., 2017). The plots show the sliced Wasserstein (sW) w.r.t. the true tall posterior $p(\theta \mid x^\star_{1:n})$ as a function of $N_{\text{train}} \in \{10^3, 3.10^3, 10^4, 3.10^4\}$ and for $n \in \{1, 8, 14, 22, 30\}$.

## L   Extensions: classifier-free guidance and partial factorization

The implementation of both of these extensions can be found in our Code repository: `https://github.com/JuliaLinhart/diffusions-for-sbi`. Their performance was investigated on the three benchmarks from Section 4.2: `Lotka-Volterra`, `SLCP` and `SIR`.

### L.1   Classifier-free guidance

It is possible to implicitly learn the prior score via the classifier-free guidance (CFG) approach (Ho & Salimans, 2021), which essentially consists in randomly dropping the context variables when training the posterior score model (e.g. 20% of the time). This is useful in cases where the diffused prior score cannot be computed analytically. Figure 22 displays the sliced Wasserstein (sW) distance as a function of $N_{\text{train}}$ and compares the results obtained for `GAUSS` with the learned vs. the analytical prior score (`GAUSS (CFG)` vs. `GAUSS`). We also report the results obtained for the Maximum Mean Discrepancy (MMD) and Classifier-Two-Sample Test (C2ST) accuracy in Figure 23b. The results are highly accurate for the Log-Normal priors of `Lotka-Volterra` and `SIR`, but less satisfying for the Uniform prior in `SLCP`. We think that this is caused by the discontinuities of the Uniform distribution. In summary it seems that, under some smoothness assumptions, is indeed possible to learn the prior score via the classifier-free guidance approach.

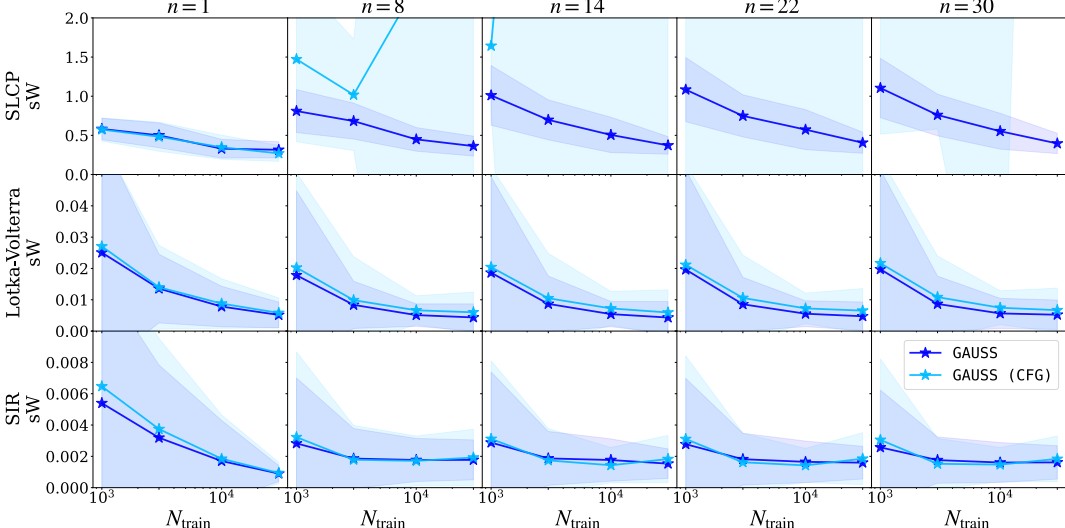

Figure 22: Results obtained for `GAUSS` with the learned vs. the analytical prior score (`GAUSS (CFG)` vs. `GAUSS`). sW w.r.t. the true tall posterior $p(\theta \mid x^{\star}_{1:n})$ as a function of $N_{\text{train}} \in \{10^3, 3.10^3, 10^4, 3.10^4\}$ and for $n \in \{1, 8, 14, 22, 30\}$.

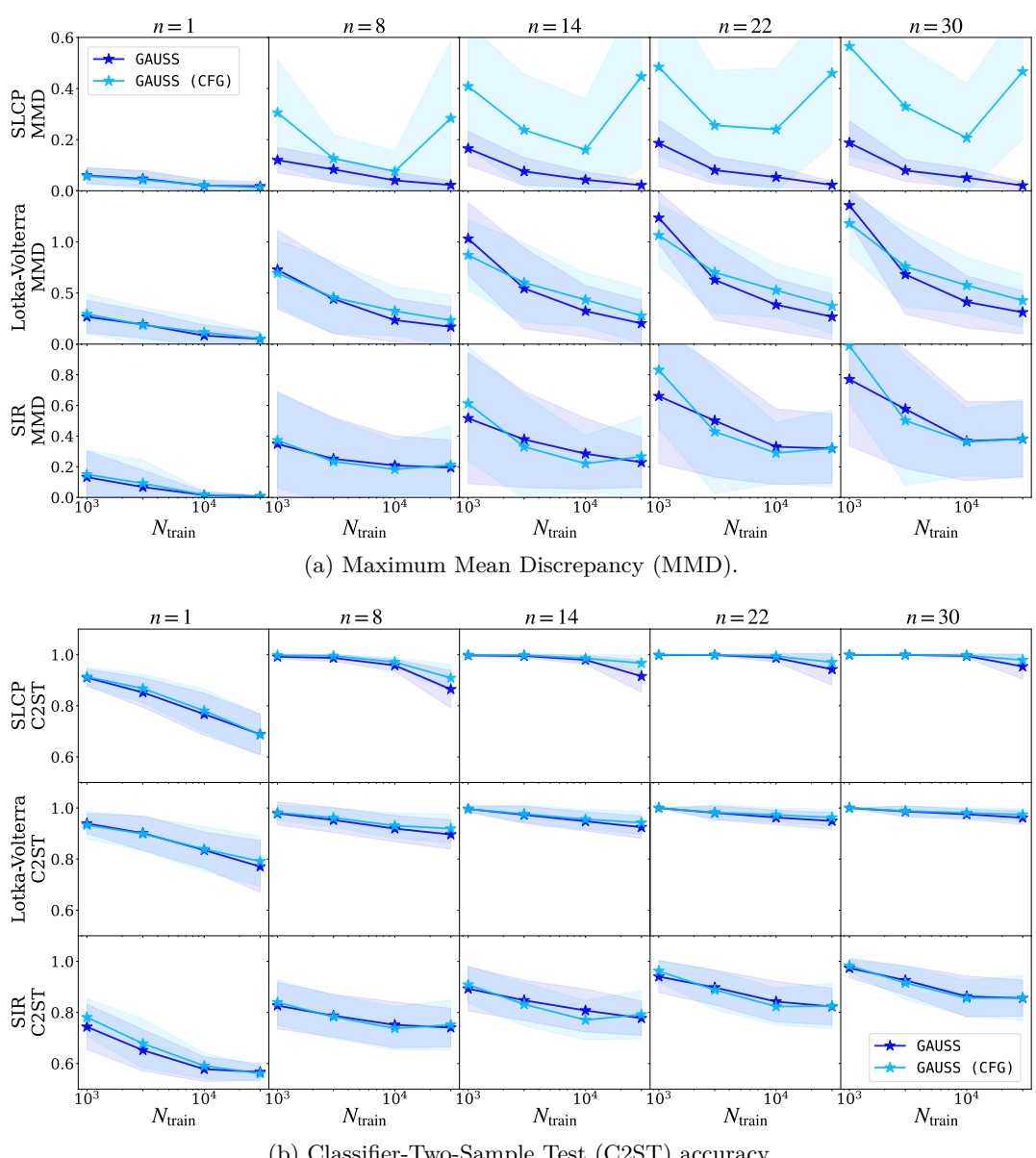

Figure 23: Results obtained for `GAUSS` with the learned vs. the analytical prior score (`GAUSS (CFG)` vs. `GAUSS`). MMD and C2ST w.r.t. the true tall posterior $p(\theta \mid x^\star_{1:n})$ as a function of $N_{\text{train}} \in \{10^3, 3.10^3, 10^4, 3.10^4\}$ and for $n \in \{1, 8, 14, 22, 30\}$.

## L.2 Partial factorization (PF-NPSE)

In the same way as in (Geffner et al., 2023), our proposed algorithm can naturally be extended to a partially factorized version. Specifically, it consists in approximating the tall posterior by factorizing it over batches of context observations (instead of a single $x$). To do so, the score model is modified to take as input context sets with variable sizes (between 1 and $n_{max}$). The given sampling algorithm (e.g. `GAUSS`, `LANGEVIN`) is then modified to split the context observations $x_1^\star, \ldots, x_n^\star$ into subsets of smaller size $k < n_{max} < n$, before passing them to the trained score model. This approach should allow for a good trade-off between the accumulation of approximation errors due to multiple evaluations of the score model ($n/n_{max}$ times) and the increased simulation budget ($\times n_{max}$). This approach should allow for a good trade-off between the accumulation of approximation errors due to multiple evaluations of the score model ($n/n_{max}$ times) and the increased simulation budget ($\times n_{max}$).

We investigated the performance of PF-NPSE on the SBI benchmarks (`Lotka-Volterra`, `SIR` and `SLCP`). For each of the three examples we trained a PF-NPSE model targeting the score models for the law of $\theta$ given $x_{1:n_{max}}$ for $n_{max} \in \{1, 3, 6, 30\}$. Figure 24 displays the sliced Wasserstein (sW), Maximum Mean Discrepancy (MMD) and the Classifier-Two-Sample Test (C2ST) accuracy for $n = 30$ observations as a function of the $N_{train}$ for samples obtained with the partially factorized `LANGEVIN` and `GAUSS` samplers and for all $n_{max}$. The extreme case $n_{max} = 1$ corresponds to the original "fully" factorized version of the samplers. $n_{max} = n = 30$ correspond to the other extreme case with no factorization, but maximum simulation budget. We can see that the optimal sW values lie in the middle of the spectrum (i.e. for $n_{max} = 3, 6$), which corresponds to what was concluded in (Geffner et al., 2023). Note that the performance of `LANGEVIN` is drastically improved for $n_{max} > 1$, while `GAUSS` all results are close. In any case, the results suggest that a practitioner will gain in choosing (a small enough) $n_{max} > 1$.

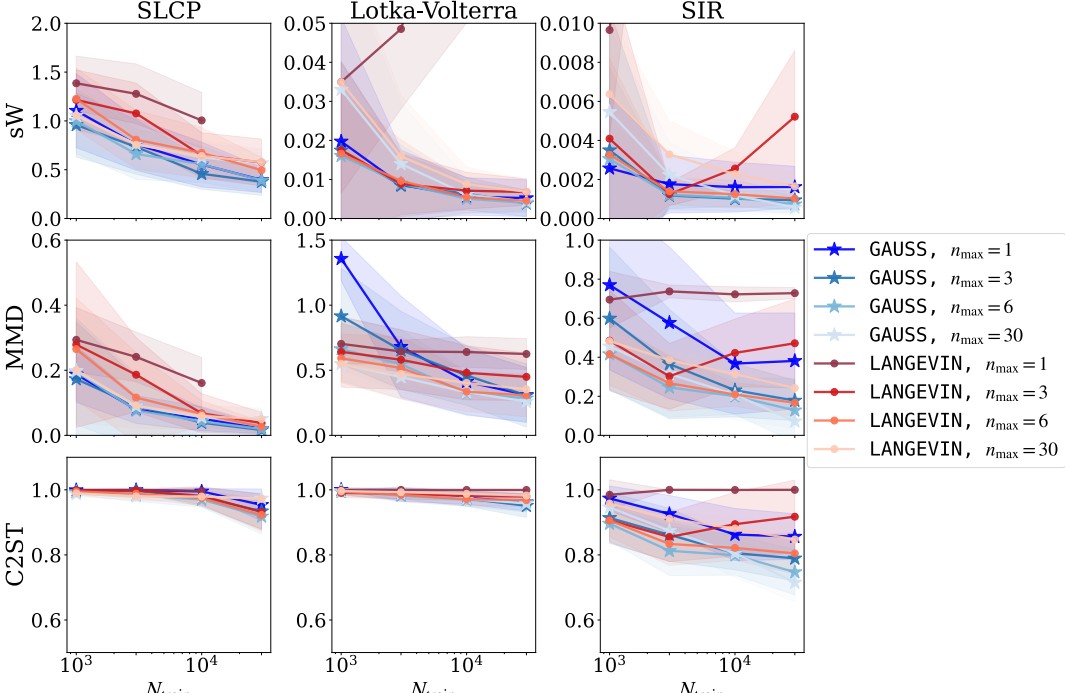

Figure 24: Results obtained with the *partially factorized* `LANGEVIN` and `GAUSS` samplers to infer the tall posterior conditioned on a total number of observations $n = 30$, for $n_{max} = 1, 3, 6, 30$. We report the sliced Wasserstein (sW), Maximum Mean Discrepancy (MMD) and the Classifier-Two-Sample Test (C2ST) accuracy w.r.t. the true tall posterior $p(\theta \mid x_{1:n}^\star)$ as a function of $N_{train} \in \{10^3, 3.10^3, 10^4, 3.10^4\}$. The extreme case $n_{max} = 1$ corresponds to the original "fully" factorized version of the samplers. $n_{max} = n = 30$ correspond to the other extreme case with no factorization, but maximum simulation budget. We can see that the optimal distance values lie in the middle of this spectrum (i.e. for $n_{max} = 3, 6$).

# M Comparison with the deterministic sampler from Geffner et al. (2023)

## M.1 Theoretical comparison

The alternative sampler proposed in Geffner et al. (2023) (Appendix D) builds a Markov chain for the tall posterior by composing the reverse kernels associated with each individual posterior. They follow the approach from Sohl-Dickstein et al. (2015) to approximately sample from a product of distributions, each associated to an independent diffusion process. Concretely, if

$$q_{t-1|t}^{(j)}(\theta_{t-1} \mid \theta_t) \approx \mathcal{N}\left(\mu_t^{(j)}(\theta_t), \ (1-\alpha_t)\mathbf{I}\right) \tag{52}$$

denotes the learned reverse kernel for $x_j$, the proposed algorithm performs a reverse step of the form

$$\theta_{t-1} \sim \mathcal{N}\left(\mu_t(\theta_t, x_{1:n}), \ \sigma_t^2 \mathbf{I}\right), \tag{53}$$

where $\mu_t$ is the average over all individual means plus a heuristic prior correction, and all observations share the same scalar variance $\sigma_t^2 = \frac{1-\alpha_t}{n-\alpha_t(n-1)}$. This transition is therefore *not* derived as the reverse process of a single diffusion whose marginals match the tall posterior, but rather defines a surrogate Markov chain intended to approximate it.

Our approach instead samples backward from the single diffusion process whose marginals satisfy

$$p_t(\theta_t \mid x_{1:n}) \propto \int \lambda(\theta_0)^{1-n} \prod_{j=1}^{n} p(\theta_0 \mid x_j) \, q_{t|0}(\theta_t \mid \theta_0) \, \mathrm{d}\theta_0, \tag{54}$$

and then approximates its reverse process through a Tweedie approximation of each backward kernel

$$q_{0|t}^{(j)}(\theta_0 \mid \theta_t) \approx \mathcal{N}\left(\mu_t^{(j)}(\theta_t), \ \Sigma_{t,j}(\theta_t)\right) \quad \text{with} \quad \Sigma_{t,j}(\theta_t) = \frac{1-\alpha_t}{\sqrt{\alpha_t}} \nabla \mu_t^{(j)}(\theta_t), \tag{55}$$

where $\mu_t^{(j)}(\theta_t) = \frac{1}{\sqrt{\alpha_t}}(\theta_t + (1-\alpha_t) \nabla_{\theta_t} \log p_t(\theta_t \mid x_j))$. This yields our approximate score

$$\nabla_{\theta_t} \log p_t(\theta \mid x_{1:n}) \approx \Lambda(\theta_t)^{-1} \left( \sum_{j=1}^{n} \Sigma_{t,j}^{-1}(\theta_t) \nabla_{\theta_t} \log p_t(\theta_t \mid x_j) + (1-n) \Sigma_{t,\lambda}^{-1}(\theta_t) \nabla_{\theta_t} \log p_t^{\lambda}(\theta_t) \right), \tag{56}$$

with $\Lambda(\theta) = \sum_{j=1}^{n} \Sigma_{t,j}^{-1}(\theta) + (1-n) \Sigma_{t,\lambda}^{-1}(\theta)$ and assumed constant covariances, as explained in Section 3.2.

We therefore have the score of the marginal density of an actual known forward diffusion process and can thus run a DDIM sampler along the corresponding reverse-time dynamics. Importantly, the backward covariance matrices in $q_{0|t}$ include important information about these dynamics (via the score) and can be different for each individual posterior. In contrast, the sampler in Geffner et al. (2023) uses a fixed transition covariance $\sigma_t^2 \mathbf{I}$, enforcing isotropic contraction that does not reflect posterior geometry. We do acknowledge that more approximations come into play when we assume the covariances to be constant (and compute them following the `GAUSS` or `JAC` algorithms). But they seem more subtle than just defining isotropic covariances as in the sampler from Geffner et al. (2023) (Appendix D) and adding a prior correction term.

For example, take the case where each posterior and the prior are Gaussian (Appendix D). Our approximation recovers the exact tall-posterior precision (and score). The sampler proposed in Geffner et al. (2023) does not, unless all covariances are isotropic.

In short, both methods make Gaussian approximations when computing the backward transitions, but not at the same level. Ours is structurally closer to the true reverse diffusion, enabling DDIM sampling, correct posterior contraction, and exactness in the Gaussian limit. Note that both methods can be equivalent if each individual posterior and the prior are Gaussian with isotropic covariance matrix (or the parameter space is one-dimensional).

## M.2 Empirical Comparison

We compare `GAUSS`, `JAC`, and `LANGEVIN` with the deterministic sampler from Geffner et al. (2023) (Appendix D), referred to as `DET_GEF`, on the toy models from Section 4.1 and the benchmark tasks from Section 4.2.[11] The implementation for this new sampler and instructions to reproduce the following experiments can be found in our Code repository: `https://github.com/JuliaLinhart/diffusions-for-sbi`.

**Runtime.** Table 8 extends the speed-up comparison from Table 2 for the `Gaussian` toy example. It shows that `DET_GEF` is consistently the fastest sampler in our experiments, followed by `GAUSS` and `JAC`. For the remaining experiments, we again consider the equivalent time setting with 400 and 1000 steps for `JAC` / `LANGEVIN` and `DET_GEF` / `GAUSS` respectively.

| $m$ | Speed up `GAUSS` | Speed up `JAC` | Speed up `DET_GEF` |
|---|---|---|---|
| 2 | $0.39 \pm 0.04$ | $0.45 \pm 0.00$ | $0.22 \pm 0.00$ |
| 4 | $0.37 \pm 0.01$ | $0.45 \pm 0.00$ | $0.22 \pm 0.00$ |
| 8 | $0.37 \pm 0.01$ | $0.46 \pm 0.00$ | $0.22 \pm 0.00$ |
| 10 | $0.37 \pm 0.01$ | $0.45 \pm 0.00$ | $0.22 \pm 0.00$ |
| 16 | $0.37 \pm 0.01$ | $0.49 \pm 0.01$ | $0.22 \pm 0.00$ |
| 32 | $0.37 \pm 0.01$ | $0.52 \pm 0.01$ | $0.21 \pm 0.00$ |

Table 8: Ratio of the runtime for `GAUSS`, `JAC`, and `DET_GEF` w.r.t. `LANGEVIN` for the `Gaussian` example in different dimensions $m$. Averaged over the number of steps $T \in \{50, 150, 400, 1000\}$, different noise levels $\epsilon \in \{0, 10^{-3}, 10^{-2}, 10^{-1}\}$ and the number of observations $n \in [1, 100]$. Mean and std over 5 different seeds.

**Accuracy and stability.** Figure 25 shows the empirical results obtained for the `Gaussian` toy examples from Section 4.1. In the `Gaussian` example, `GAUSS` achieves the best performance across noise levels, as the Gaussian approximation is exact in this case. `JAC` is also exact, but less stable when noise increases. `DET_GEF` remains close but shows a consistent gap, since it relies on a surrogate chain with fixed isotropic variance and is therefore not exact in this example where the covariance is not isotropic. As $n$ increases, `DET_GEF` and `LANGEVIN` improve and become closer in performance, consistent with posterior concentration: the tall posterior becomes increasingly unimodal with small variance (Bernstein–von Mises), making `DET_GEF` a better approximation and `LANGEVIN` easier to converge.

For `GMM`, `LANGEVIN`, `JAC`, and `DET_GEF` outperform `GAUSS` in low-noise regimes. In particular as $n$ grows, all methods become highly accurate, except `GAUSS` whose approximation gap remains visible. However, `GAUSS` remains the most stable method in the most noisy regime ($\epsilon = 0.1$), where `LANGEVIN` and `DET_GEF` degrade more strongly, and `JAC` diverges strongly, with sW values outside the plot limits.

We now move on to the sbi-benchmark tasks from Section 4.2, where the score is learned and imperfect, and compositional evaluation across observations causes error accumulation as $n$ increases. Results are shown in Figure 26. In this regime, `LANGEVIN` and `DET_GEF` diverge as $N_{\mathrm{train}}$ grows: the learned score becomes more confident while retaining small systematic bias, which is amplified by posterior contraction. `DET_GEF` is particularly sensitive since its approximation relies directly on the composed backward mean, which depends on the learned score. In contrast, `GAUSS` more closely approximates the reverse diffusion dynamics of the tall posterior and remains significantly more stable, consistent with its robustness in the noisy `GMM` toy regime. `JAC` with clipping is promising but still biased.

Overall, these additional experiments confirm our main conclusions: `LANGEVIN` is the computationally most expensive and least stable method. `DET_GEF` provides a useful fast deterministic (i.e. that does not rely on Langevin dynamics) baseline, but `GAUSS`, and stable variants of `JAC`, offer the most reliable sampling behavior in realistic learned-score tall-data benchmarks.

---

[11]Note that the results for `GAUSS`, `JAC`, and `LANGEVIN` may slightly differ from those reported in the main text, as the experiments were rerun from scratch and minor variations can arise from different random seeds and GPU configurations.

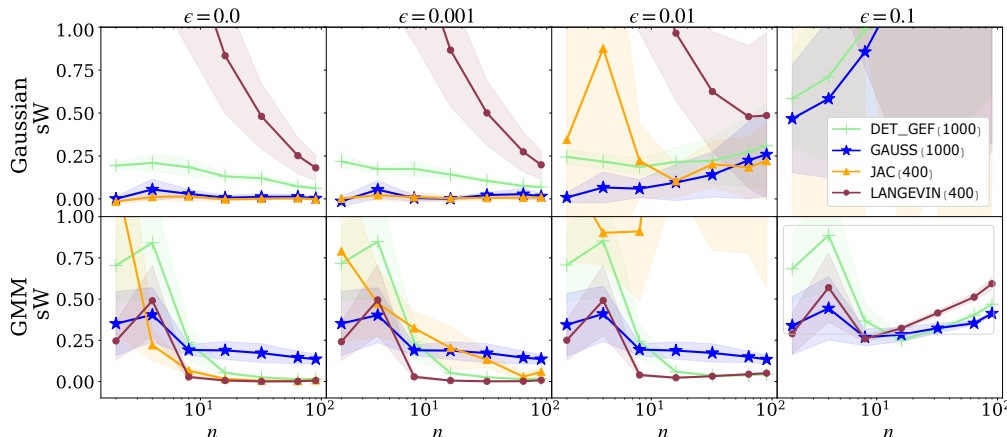

Figure 25: Sliced Wasserstein (sW) distance as a function of $n$ and for increasing noise levels $\epsilon$. Results are shown for both Gaussian toy examples with $m = 10$. Mean and std over 5 different seeds.

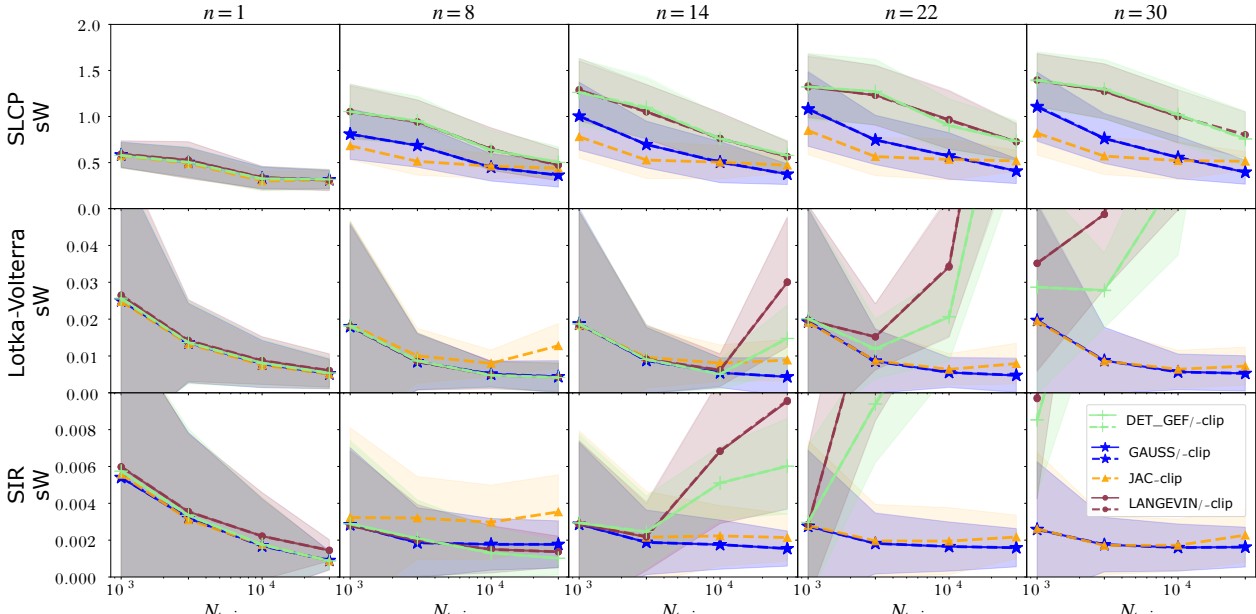

Figure 26: Sliced Wasserstein (sW) distance as a function of $N_{\text{train}}$ and for increasing $n$, for the benchmark tasks SLCP, Lotka-Volterra, and SIR. Mean and std over 25 seeds.

**Takeaway:**

- **GAUSS** is the most stable sampler, with the best results in highly noisy regimes or for learned scores (sbibm), even if its approximation appears to be less accurate than **JAC** or **DET_GEF** in the non-Gaussian (GMM) setting (for exact non-noisy scores).

- **JAC** is the most accurate method in the noise-free setting across both toy tasks, but remains unstable in practice. Clipping improves stability but introduces bias. A more principled stabilisation (beyond clipping) could yield further improvements and could be a promising research direction.

- **DET_GEF** is fastest and competitive with our proposals, even outperforming **GAUSS** on GMM in the low-noise regime as $n$ grows. However, it is still less stable than **GAUSS** as noise increases or in the learned score regime (sbibm), with similar behavior to **LANGEVIN**.

### M.3 Clarifying Langevin behavior across settings

**Noise-free toy regime.** Figure 25 shows an interesting trend: `LANGEVIN` improves as $n$ increases. This could be explained by the fact that, as $n$ grows, the tall posterior becomes sharply concentrated and locally Gaussian (Bernstein–von Mises), making the sampling dynamics easier to handle with an appropriate step size, whereas for small $n$ the broader posterior would require more careful mixing (larger step sizes or more inner steps).

**Noisy toy and learned-score regime.** Figure 26 in contrast, shows that performance degrades with increasing $n$ (especially at large $N_{\text{train}}$). This is probably because small score approximation errors accumulate across composed observations and are amplified by posterior contraction, leading to biased concentrated estimates and divergence. The fact that `DET_GEF` exhibits similar instability indicates that this behavior is not merely a Langevin tuning issue, but rather an intrinsic sensitivity of samplers whose updates rely directly on the composed learned scores, making them highly vulnerable to accumulated bias in concentrated regimes. In contrast, `GAUSS` and `JAC` approximate the tall reverse diffusion dynamics more explicitly through diffusion-consistent backward-kernel constructions and associated covariance structure, which provides additional stabilization and yields more robust behavior under imperfect learned scores.

