# OpenReview forum: "Diffusion posterior sampling for simulation-based inference in tall data settings"
_TMLR — Accepted by TMLR_

### Review · Reviewer_So1v · 2025-11-24

**Summary Of Contributions:**

Authors consider a problem of identifying parameters of the system $\theta$ from multiple observations $x_1,\dots, x_n$ in form of a posterior distribution $p(\theta|x_1,\dots, x_n)$. It is assumed that: (i)$p(x|\theta)$ is intractable; (ii) it is possible/easy to generate $x$ from $\theta$ by simulation. The approach proposed by authors is to:
1. approximate $p(\theta|x)$ by score matching
2. approximate $p(\theta|x_1,\dots, x_n)$ using prior distribution $p(x)$, $p(\theta|x_i), i=1,\dots,n$ and data generated from $p(\theta|x)$

As a result, sampling from approximated $p(\theta|x_1,\dots, x_n)$ can be done by backward diffusion process.

Authors demonstrate that their approach is cheaper and more accurate than Factorised Neural Posterior Score Estimation, a method that extends Neural Posterior Score Estimation to the tall data setting.

**Audience:**

Yes

**Audience Explanation:**

Given that the work is on the intersection of generative modelling, AI4Science and numerical methods, many readers of TMLR can be interested in the contribution.

**Claims And Evidence:**

Yes

**Claims Explanation:**

Authors made several claims:
1. Introduction of a novel method that is similar to F-NPSE requires only learning $p(\theta|x)$ but that works in a tall data setting
2. Introduced method does not require Langevin dynamics
3. Empirical results showing that new method is superior to F-NPSE (better accuracy and stability)
4. Evaluation is done on multiple standard simulation-based inference benchmarks including challenging real-world problems.

In my view, the main claims are well-supported by the results of the authors.

**Requested Changes:**

The paper is well-written, with clearly stated motivation and detailed discussion of related work. I have only a few minor questions that appear below:
1. The derivation of the method (Sections 3.1 and 3.2) is motivated by the assumption that the approximation of $p(\theta|x_1,\dots, x_n)$ should be suitable for sampling with a backward diffusion process. Is it possible to say something about the accuracy of this approximation? Possibly, given some convenient assumptions.
2. Lemma 3.1 contains an assumption that a certain matrix is positive definite. In appendix A.1 authors provide some discussion when this is the case. This condition is possible to check numerically. Did the authors try that?
3. Algorithm 2 requires an approximate covariance matrix. One may expect that with large number of parameters such an approximation will become less reliable. Can the authors please comment on that?
4. In both Algorithm 1 and Algorithm 2 dense matrices are formed and stored. Does it mean that the methods are not applicable to parameter spaces of very large dimensions, e.g., $m = 10^{6}$?
5. I suggest clarifying the meaning of $m$, $d$ in Section 4.2.
6. In Appendix J. 2. there are references to undefined Section and equations. This appendix also mentions "this thesis’ introduction" which is likely a typo.
7. Figure 5 is too small. Can the authors provide a larger version of this paper in the Appendix or in the main text of the paper?
8. Similarly, Figure 10 in the appendix is too small. Is it possible to provide a larger version of it?
9. In Section 4.3. authors refer to $g$ as "gain". I suggest removing that or to provide a more detailed discussion of parameters in the main text.

---

> ### Author Response · Authors · 2025-12-11
> **Response to Reviewer So1v**
>
> ## 1. Accuracy of the approximation
>
> We thank the reviewer for this very relevant question. The other reviewers asked similar questions and we refer the reviewer to the answers given to those.
> - What makes our approximation more subtle than the one proposed in the appendix from Geffner et al. (Question 1, Reviewer p3EZ).
> - Is there an irreducible bias introduced by the approximations made that would not be there with a perfectly tuned Langevin algorithm? (Question 2, Reviewer p3EZ)
> - How good is the approximation made using the Tweedie framework, what assumptions are required? (Question 1, Reviewer Tubi)
> - Can the method be improved with a more elaborate non-Gaussian approximation? (Question 4, Reviewer p3EZ)
>
> ## 2. Numerical check for positive definite matrices
>
> This is an important comment. During the initial experimental phase of this project, this positive definite character was specifically what that made the numerical implementation very hard and was the reason why our first version was very unstable. These first experiments ended up making us look for a better/simpler formula to implement, resulting Algorithms 1 and 2 (JAC and GAUSS).
>
> The GAUSS algorithm has no issues with positive definiteness, as the covariances are true covariances from a Gaussian distribution. For JAC however, taking the gradient of the score network has no reason to result in a "true covariance matrix" with guaranteed positive definiteness. This is why JAC appears less stable than GAUSS. We used clipping to improve stability.
>
> Importantly, it is critical to have a way to ensure positive definiteness without introducing too much bias into the sampling procedure. On that note, we would like to refer the reviewer to Gloeckler et al. (2024); Appendix B.3: they propose an improvement of our method by ensuring the positive definiteness of Lambda using its eigenvalue decomposition. Results showed JAC do be way more stable after this small adjustment.
>
> As discussed in our paper, this points towards the promising direction of being able to use JAC as the correct approximation (no Gaussian assumption for the posterior as in GAUSS), and without suffering instability or bias introduced by clipping (as seen in our experimental results).
>
> ## 3-4. Covariance matrices in large parameter spaces (e.g., $m=10^6$)
>
> - **Reliability of the approximation in Algorithm 2:** the approximation in GAUSS is definitely not ideal and if JAC can be made more stable it is a better algorithm than GAUSS (see the response to the previous question), that might scale better to large parameter-spaces. Indeed, it seems obvious that the GAUSS approximation will be worse as the number of parameters grows, due to the challenge of capturing higher order correlations or more complicated structures (e.g. multiple modes, ...).
>
> - **Storing dense matrices:** we do agree that storing dense matrices is not ideal and might make the method inapplicable in large parameter spaces, or at least computationally very expensive! Maybe "learning" the correction term directly (as suggested by reviewer p3EZ) would be a solution to that.
>
> While it would be interesting to see if and how our method scales to large parameter spaces, we would like to point out, however, that this scenario rarely happens in SBI (parameter dimension is typically of order 1-10 and sometimes 10-100).
>
> ## 5. Meaning of $(m,d)$ in Section 4.2.
>
> We thank the reviewer for his comment. Although we do not fully understand what he means. Is it reminding the reader that $m$ is the parameter dimension and $d$ the dimension of the data/simulations? Or is it the meaning of the values of $m$ and $d$? In the first case, we think that this is rather clear from the previous experiments. In the second case, I don't know how relevant it is given the limited length of the paper.
>
> ## 6-7-8-9. Typos, etc.
>
> - We thank the reviewer for pointing out the typos, we will correct them in the final version of our paper.
>
> - We will also increase the size for the requested Figures. Does the main text length need to stay the same?
>
> - We agree that it might be confusion and will remove "gain", replacing it with just "g" in the final version of our paper.

---

> > ### Comment · Reviewer_So1v · 2025-12-17
> >
> > I would like to thank the authors for clarifications regarding spd matrices and accuracy of the method. In the rebuttal authors addressed my main concerns.
> >
> > The authors asked for several clarifications on minor points that I raised. I provide answers below:
> > 1. Meaning of  $(m, d)$ in Section 4.2.
> >
> >     In principle I agree with the authors that they completely define probability distribution and the problem on page 1 in the Introduction. Namely, for probability distribution $p(x, \theta)$ where $x\in\mathbb{R}^{d}$, $\theta\in\mathbb{R}^{m}$ the problem is to infer parameters $\theta^{\star}$ corresponding to $n$ observations $x_{1}, \dots, x_{n}$. However, this is the only place where the dimensions of $\theta$ and $x$ are specified explicitly. Implicitly, the dimension of the space of parameters is mentioned in several places, e.g., “we need to calculate a $m\times m$ matrix” (page 7).
> >
> >     In Section 4.1, the authors seem to use distribution with $d = m$, and $d$ does not appear in the section explicitly. In Table 2 (and page 9) authors also refer to “Gaussian example in different dimensions $m$”, so now the reader may assume that $m$ is a dimension of $x$. In my view it all adds up to the confusion and when the reader reaches Section 4.2 they need to recall that definition from page 1.
> >
> >     So my suggestion is to remind the reader the dimensions of observations and parameters.
> >
> > 2. Size of Figures.
> >
> >     Even with maximal zoom allowed by my pdf reader I can barely see the contour plots of probability distributions for pairs of variables that authors provide in Figure 5. My suggestion is to rework the Figure 5 and figures alike available in the appendices.
> >
> > I do not consider these points substantial, so the decision whether to follow the suggestions above is fully at the discretion of the authors.

---

> > > ### Author Response · Authors · 2026-01-09
> > > **Response**
> > >
> > > We thank Reviewer So1v for the clarifications and will address these points in the final version of the paper.

---

### Review · Reviewer_Tubi · 2025-11-25

**Summary Of Contributions:**

The paper presents an SBI method for tall data settings based on the scores of single-observation posteriors. This is made possible by an exact decomposition of the score function for multi-observation posteriors. The per-observation scores can be estimated purely from joint samples, which are available in the SBI setting. While the introduced decomposition of the score is not fully tractable, the authors present two different ways to approximate it. This gives (approximate) scores for a known diffusion process, allowing deterministic samplers such as DDIM to be applied.

**Additional Comments:**

While I have a good understanding of SBI, I am not an expert in diffusion.

**Audience:**

Yes

**Audience Explanation:**

SBI and SBI in tall data settings is a relevant problem in many scientific domains. It is clearly not solved yet and this work makes an interesting contribution to the field.

**Broader Impact Concerns:**

This is fundamental research with no immediate impact concerns.

**Claims And Evidence:**

Yes

**Claims Explanation:**

The paper is well written overall, with a clear motivation by pointing out issues with F-NPSE. The exact decomposition of the tall data posterior score seems correct. Several approximations made to then estimate this seem reasonable, some questions on this below. The results show that this method cab work in principle and often better than F-NPSE.

### Questions

- The correction term in the tall data score, $L_\lambda$, is approximated using the Tweedie framework. Can the authors elaborate on what kinds of assumptions this rests on and under which circumstances this is a good or bad approximation?
- Why does LANGEVIN improve when conditioned on taller data? See e.g. Figure 2, Gaussian.
- For problems in 4.1, does the method not learn the score but just computes it analytically except for $L_\lambda$ and the difficulty is then to handle the noise?
- Why not compare methods with the same time stepping? The "equivalent time" setting is interesting, but seeing methods with the same number of diffusion steps would also be relevant.
- Why does Langevin performance worsen with increased $N_{train}$ (Figure 3,  LV and SIR)?
- For GAUSS, is the time to estimate the $\Sigma_{0,j}$'s included in Tables 1 and 2?
- Can you better explain what you are doing with the clipping? The explanation in 4.2 is a bit limited.

**Requested Changes:**

I will wait for answers to my questions before I request changes.

---

> ### Author Response · Authors · 2025-12-11
> **Response to Reviewer Tubi**
>
> ## 1. Elucidating the Tweedie approximation
>
> The Tweedie identity relates the conditional expectation under the clean variable to the score of its noisy version and was used for diffusion-based inverse problems by Boys et al. (2023). For a diffusion step $\theta_t \sim \mathcal N (\sqrt{\alpha_t}\theta_0, (1-\alpha_t)I$, Tweedie gives:
>
> $E[\theta_0|\theta_t] = \frac{1}{\sqrt{\alpha_t}}(\theta_t + (1-\alpha_t)\nabla \log p_t(\theta_t)).$
>
> We approximate the backward kernels $q_{0|t}$ using Gaussian distributions whose means and covariances are built from these Tweedie moments. The validity of this approximation rests on two conditions:
> - **Small diffusion increments for local closeness:** when $\alpha_t \approx 1$, $\theta_t$ and $\theta_0$ only differ by a small Gaussian perturbation and local expansions of the posterior are accurate.
> - **Smoothness of the target distribution:** this allows a second-order Taylor expansion of $\log q_{0|t}$ around $\theta_t$, which after exponentiation becomes a locally Gaussian approximation of $q_{0|t}$. This is further supported in our tall-data setting: as $n$ grows, the posterior rapidly concentrates and becomes locally Gaussian (Bernstein-von-Mises).
>
> In summary, the proposed approximation is "good" because simple and effective. Importantly, in the Gaussian case, our approximation is exact, but we acknowledge that it might degrade or collapse for highly multimodal or heavy-tailed distributions.
>
> ## 2. Improvement of Langevin for taller data (Figure 2, Gaussian)
>
> We thank the reviewer for this interesting question. The observed behavior is peculiar on first sight and goes against what we observe in the experiments in section 4.2 (Figure 3), where the score model is learned and the performance worsens with increasing $n$. In order understand what's happening, let's isolate the case without noise ($\epsilon=0$, first column in Figure 2), which reflects a perfectly learned score model (analytical scores are used). The Gaussian example is the case where our approximation is exact, explaining the good performance of our methods for all $n$. Langevin improves as n increases. This is also true in the GMM example, where similar behavior can be detected for JAC and even GAUSS (less but still).
>
> So why is that? As $n$ grows, the posterior becomes Gaussian and the variance becomes smaller (Berstein-von-Mises), which might make it easier for the samplers to converge. In other words, the tall posterior becomes "more simple" and therefore easier to approximate. In Figure 3, this is not the case as the learned score is imperfect: we observe an accumulation of errors as $n$ grows. This can be explained (i) by the multiple network evaluations in the compositional score formula, (ii) poor tuning of the Langevin step-size w.r.t. $n$ (see reponse to question 2 of Reviewer p3EZ).
>
> ## 3. Analytical score + noise injection
>
> We thank the reviewer for clarifying this. Yes, in 4.1 all the quantities are known analytically. We simulate the case where the score is learned (and thus not perfect) by adding noise in the form of a randomly initialized neural network. By increasing the noise factor we simulate the case of a badly learned score and analyze the robustness of the samplers to this situation.
>
> ## 4. Equivalent time setting vs. same number of steps
>
> We agree that comparing for same number of steps is relevant and would like to point out that we do so in Table 1. For clarity, space and time, we chose to perform subsequent experiments in the equivalent time setting only, as we found the message more clear: with less compute we perform better (because we are faster and can thus use more steps and be more precise). However we do agree that it would be interesting to see how this affects the results from Section 4.2: does the performance gap to Langevin remain? Looking at Table 1, we think that the answer is yes.
>
> ## 5. Worse performance for Langevin with more training samples (Figure 3, LV and SIR)
>
> We thank the reviewer for this insightful question. It was also asked by reviewer p3EZ (Question 3) and we would like to refer to the response we gave him. However, let's clarify why this phenomenon is only observed for LV and SIR, and not SLCP: We think that it's because SLCP is a hard example (multimodal) and the performance of the samplers is not good to to begin with. So a small bias would not impact the already bad performance metric value. SIR and LV on the other hand have simple Gaussian posteriors and as the variance decreases a small bias can very much affect the performance: overconfidence becomes critical as $n$ increases.
>
>
> ## 6. For GAUSS, is the time to estimate the 's included in Tables 1 and 2?
>
> Yes!
>
> ## 7. Clipping
>
> We agree that the explanation provided for clipping is limited. It is a very simple procedure, where we ensure that the samples generated at each time step remain in the support of a standard Gaussian by truncating the samples to a predefined range (here [-3,3]).

---

### Review · Reviewer_p3EZ · 2025-11-28

**Summary Of Contributions:**

This paper addresses the challenge SBI with "tall data", where the goal is to infer parameters given multiple independent observations. The authors comment on the limitations on the LAngevin sampling algorithm used by F-NPSE, and propose an alternative sampling algorithm that does not rely on Langevin dynamics (which at the same time involves selecting a few parameters, like number of steps per noise level, and step size). They derive this approximate sampling algorithm by approximating certain intractable terms in the "true tall-data" score with Gaussian distributions.

**Additional Comments:**

One more question. How much of these results do you think comes from poor tuning of the Langevin sampler? Having to tune it is, of course, a weakness of the method. That being said, if tuned properly, it should work better than the proposed methods, or is that not correct? I am mostly thinking about this from the perspective of the bias introduced by the approximations, which seems to be irreducible.

**Audience:**

Yes

**Audience Explanation:**

I think people working in the field of SBI, specifically with problems involving large sets of observations, could find this work useful and interesting.

**Broader Impact Concerns:**

No.

**Claims And Evidence:**

Yes

**Claims Explanation:**

The authors provide an empirical evaluation showing that the method they propose, which does not rely on Langevin dynamics, yields good performance. While the paper has good evidence, I have some questions / concerns that I'd like to have addressed.

**Requested Changes:**

A few comments / questions / concerns.

**Comparison with Deterministic Baselines**

I think it would be interesting if the authors compare against the non-Langevin sampler from Geffner et al. (2023), presented in their appendix. That sampling algorithm also circumvents Langevin dynamics, and appears to do it in a way that's not completely foreign to the directions pursued in this work. It appears that both works use Gaussian assumptions to render the sampling tractable, though they appear to apply these approximations for different terms: the proposed method in this work approximates the backward kernel ($q_{0|t}$), whereas Geffner et al. approximate the backward transitions $q(t-1 | t)$. I think it would be valuable to compare these approaches, from both an empirical perspective (i.e. include it in the empirical evaluation), and a theoretical perspective, trying to shed light on how the different approximations used differ and whether they may be equivalent in some toy scenarios maybe?

**Asymptotic Exactness and Structural Bias**

A strong theoretical property of the Langevin sampler is its asymptotic exactness; with a step size tending to zero and a sufficient number of steps, it draws from the exact distribution defined by the score field. The method proposed here, however, relies on two distinct approximations: first, the approximation of the intractable backward kernels as Gaussians, and second, the approximation of the covariance matrices as constant (in the GAUSS algorithm) to avoid Jacobian computations. Consequently, regardless of the sampling budget or the precision of the score network, this method cannot theoretically recover the exact posterior structure for non-Gaussian distributions. While the authors correctly identify that the asymptotic behavior of Langevin is difficult to realize in practice (and I fully agree with this), the proposed method trades this for an irreducible structural bias. Am I missing something about the proposed method, regarding whether it can be made exact with sufficient sampling compute?

**Analysis of Baseline Performance and Hyperparameter Tuning**

In Figure 3, the Langevin baseline performs worse as the amount of training data ($N_{train}$) increases. I find this counter-intuitive, as providing more data to the score network should result in a more accurate approximation of the gradients. Can the authors comment on why the performance gets worse with more training data?

**Approximating the Intractable Correction Term**

The derivation of the tall data score hinges on the "correction term" $L_\lambda$, which involves an intractable integral over the product of backward kernels. The authors currently address this by forcing the kernels to be Gaussian so the integral becomes analytically tractable. However, this imposes the structural bias mentioned previously. I would be interested to hear the authors' thoughts on whether this term could instead be approximated using a different method (e.g. Normalizing Flows, or a DeepSet-style network). If one could train a network to approximate this term directly (conditioned on the set of single-observation scores), it might be possible to avoid the Gaussian assumption entirely. Do you think this is possible? I understand that non Gaussian approximations will lead to issues when computing the resulting integrals, so this may not be possible. Just wondering if you had any thoughts on this.

---

> ### Author Response · Authors · 2025-12-11
> **Response to Reviewer p3EZ**
>
> ## 1. Comparison with Deterministic Baselines
>
> We thank the reviewer for this excellent suggestion. The sampler proposed in **Geffner et al. (2023, Appendix D)** is closely related, and comparing the two illuminates the role of Gaussian approximations in both approaches. We unfortunately did not have time to run the empirical comparison, but we would be happy to provide results as a follow-up or include them in the final version. The clearest experiment would be the Gaussian toy example, where our approximation is exact.
>
> **The alternative sampler proposed in Geffner et al.** construct a surrogate Markov chain by composing individual Gaussian reverse transitions $q^j_{t-1|t}\approx \mathcal N (\mu^j_t, (1-\alpha_t)I)$ (same as the ones used in DDIM) and updating with
>
> $\theta_{t-1} \sim \mathcal N(\mu_t,\sigma_t^2 I),$
>
> where $\mu_t$ is the average over all individual means + a heuristic prior correction, and $\sigma_t^2$ is a fixed isotropic variance. This chain **does not correspond to the reverse of a single diffusion for the tall posterior.**
>
> **Our approach** instead models the reverse process of the *single* diffusion process of the tall posterior and uses the Tweedie framework to construct a Gaussian approximation for the *intractable* backward kernels $q^j_{0|t}$ with same mean $\mu^j_t$ but **non-isotropic covariance**
>
> $\Sigma^j_t = \frac{(1-\alpha_t)}{\sqrt{\alpha_t}}\nabla\mu_t^j.$
>
> Crucially, the covariance depends on $\mu_t^j$ which is defined via the score, which includes **important information about the reverse dynamics for each individual posterior**, whereas Geffner et al. enforce isotropic contraction, which suppresses posterior geometry.
>
> In short, both methods use Gaussian approximations, but ours is closer to the true reverse diffusion, enabling DDIM sampling and exactness in the Gaussian limit - where each posterior and the prior are Gaussian (Appendix D1 of our submission). **They are equivalent if each individual posterior and the prior are Gaussian with isotropic covariance matrix (or the parameter space is 1D).**
>
> ## 2. Asymptotic Exactness and Structural Bias
>
> We thank the reviewer for pointing out the missing clarity in asymptotic behavior of the samplers. Langevin dynamics is asymptotically exact when given the *true score*. Here we compute an approximate score of $p_t$ (via GAUSS or JAC), resulting in a different diffusion bridge
> $\pi_t$. No method using this approximate score-including Langevin-can converge to $p_t$ for $t>0$.
>
> However, this **does not lead to an irreducible bias in the final posterior samples**. Indeed, because the approximation only affects the bridge and not the endpoint, the method remains asymptotically correct at $t=0$ where $\pi_0 = p_0$ (by construction). Thus the sampler follows the correct reverse dynamics for $\pi_t$, and the final posterior samples are correct.
>
> Of course, the choice of "bridging" distributions impacts the efficiency of the sampling procedure. **Our choice of bridging distributions is closer to the true reverse diffusion process, allowing stable deterministic DDIM sampling.**
>
> N.B: In practice, the dominant bias arises from the *learned* score: summing evaluations across observations accumulates approximation error, and this—as discussed in the experiments.
>
> ## 3. Analysis of Baseline Performance and Hyperparameter Tuning
>
> We thank the reviewer for this insightful question. It is true that this example is somehow counter intuitive.
>
> While increasing $N_\mathsf{train}$ improves the score model, sampling becomes much harder as $n$ grows: the posterior contracts sharply, making Langevin highly sensitive to step size (theory points out that it must scale with the posterior variance). Here we use a fixed step size that quickly becomes too large, causing overshooting and divergence.
>
> At the same time, a more confident score + a sharper posterior means small biases can produce non-overlapping posteriors, explaining the exploding distance metrics.
>
> In conclusion, the degradation reflects the fact that Langevin is harder to tune in concentrated regimes. Our sampler, relying on deterministic reverse diffusion, remains stable in these regimes.
>
> ## 4. Approximating the Intractable Correction Term
>
> Learning the correction term directly is an interesting direction. However, it is not straightforward: the term is the score of an implicit marginal, not a density, and must remain consistent across diffusion time. A learned non-Gaussian correction would likely require augmented datasets or expensive inner Monte-Carlo loops—precisely what we aim to avoid. We see this as promising future work.
>
> ## 5. Additional Comments
> Better tuning would certainly improve the Langevin baseline, but even with the hyper-parameters recommended by Geffner et al. (who performed extensive tuning), we observed instability as $n$ increases. This reflects the intrinsic difficulty of Langevin in concentrated posterior regimes, not insufficient tuning on our side.

---

> > ### Comment · Reviewer_p3EZ · 2026-01-06
> > **Response**
> >
> > Thanks for your response. I think adding the Gaussian approximation baseline from Geffner et al. would make sense. Also, while I agree that tuning Langevin dynamics may be hard, I'd say spending some more time tuning the baselines to try to get it to perform reasonably well would be a nice addition too. This does not reduce the current paper's merit in any way, I agree with the fact that tuning Langevin dynamics properly is not easy, and this paper proposes an approximate method to address that.

---

> > > ### Author Response · Authors · 2026-01-09
> > > **Response**
> > >
> > > We thank Reviewer p3EZ for his response. We agree with the relevance of the comparison to the Gaussian approximation baseline from Geffner et al. and will include corresponding results in the final version of our paper. As for the additional tuning of the Langevin algorithm, we will do our best, but cannot guarantee that the results will be conclusive and relevant enough to include them in the paper.

---

> > > > ### Comment · Reviewer_p3EZ · 2026-01-09
> > > > **Response**
> > > >
> > > > One simple option would be making the Langevin step size smaller and increasing the number of steps per noise level accordingly. This will increase the computational cost, and you can report that as well, as another way to show the benefits of your method in terms of efficiency, if the difference is large. By making the step size smaller it should hopefully not be too hard reaching a point where the sampler behaves in a stable manner, regardless of cost.

---

> > > > > ### Author Response · Authors · 2026-01-09
> > > > > **Response**
> > > > >
> > > > > Yes, thank you, I agree! I was going to say, stepping outside of the "equivalent time setting" would probably be enough. And this would be another way to show the efficiency of our method, besides our results in Table 1. But if it's not enough, we would have to try for more sophisticated solutions (e.g. adaptive step sizes), which we will not have the time for. Let's hope and see!

---

### Author Response · Authors · 2025-12-11
**General Response to all Reviewers**

We sincerely thank all reviewers for their very insightful and constructive feedback. The questions raised have helped us refine several theoretical aspects of the work, as well as better articulate empirical behaviors—particularly those that may at first seem counter-intuitive. We are confident that the revisions and clarifications resulting from this rebuttal will strengthen the paper and its usefulness to the community.

We invite all reviewers to consult the full set of responses below, as many answers intersect across reviews and together provide a coherent picture of the theoretical and practical implications of our approach. Recurring questions for all reviewers that we found to be most relevant concern:
- the **nature and accuracy of our approximations** (Gaussian + Tweedie) and how these compare to the deterministic sampler from (Geffner et al., 2023; Appendix D) $\rightarrow$ (Q1-2-4, Reviewer p3EZ; Q1 Reviewer Tubi; Q1 Reviewer So1v)

- **how both $n$ and $N_\mathrm{train}$ affect sampling performance**, especially for Langevin dynamics $\rightarrow$ (Q3, Reviewer p3EZ; Q2 Reviewer Tubi)

- whether **more careful tuning of Langevin hyper-parameters** could significantly improve its behavior $\rightarrow$ (Q5, Reviewer p3EZ)

Because of the strict character limits on individual responses, several answers had to be significantly more concise than we would have preferred. We are very happy to provide additional mathematical details, extended derivations, or further clarifications if needed.

We also apologize for not being able to run the requested empirical comparisons within the rebuttal window. They are genuinely interesting, and we would be happy to include the corresponding results in the camera-ready version.

Finally, we also thank the reviewers for noting typos, unclear phrasing, and figure-size issues; all of these will be corrected in the final version.

---

### Decision · Action_Editor_jiN2 · 2026-01-19

**Recommendation:** Accept as is

**Additional Comments:**

The reviewers did not have any substantial concerns about the paper and appreciated the detailed clarifications provided by the authors.

The authors are urged to include the changes promised to the reviewers in the camera-ready version of the paper. It would also be good to make the score function notation consistent throughout the paper by always specifying the variable for the grad operator.

**Audience:**

Yes

**Audience Explanation:**

This paper will be of interest both to researchers working on Simulation Based Inference and to practitioners using it as a tool.

**Claims And Evidence:**

Yes

**Claims Explanation:**

The paper proposes a way to make Simulation Based Inference more efficient and reliable in the tall data setting, where multiple independent observations are available per parameter configuration. A previously proposed method provides a flexible approach to this setting by first learning the score functions learned from (single observation, parameter configuration) pairs and then composing the functions at inference time to handle multiple observations. The method however requires Langevin sampling at inference time, making it computationally expensive and challenging to tune. The authors propose an alternative approach that retains the score function compositionality but eliminates the need for Langevin sampling. This is achieved by approximating the diffusion process modelling the multi-observation posterior and performing inference by invoking existing diffusion sampling algorithms, such as DDIM.

There was agreement among the reviewers that the claims in the paper are well-supported.